# SimulCost: A Cost-Aware Benchmark for Automating Physics Simulations with LLMs

Yadi Cao [* 1]  Sicheng Lai [* † 2]  Jiahe Huang [* 1]  Yang Zhang [* † 3]  Zach Lawrence [* 1]  Rohan Bhakta [* 1]
Izzy F. Thomas [* 1]  Mingyun Cao [* † 4]  Chung-Hao Tsai [1]  Zihao Zhou [1]  Yidong Zhao [5]  Hao Liu [6]
Alessandro Marinoni [1]  Alexey Arefiev [1]  Rose Yu [1]

## Abstract

Evaluating LLM agents for scientific tasks has focused on token costs while ignoring tool-use costs like simulation time and experimental resources. As a result, metrics like pass@k become impractical under realistic budget constraints. To address this gap, we introduce SIMULCOST, the first benchmark targeting cost-sensitive parameter tuning in physics simulations. SIMULCOST compares LLM tuning cost-sensitive parameters against traditional scanning approach in both accuracy and computational cost, spanning 2,947 single-round (initial guess) and 1,931 multi-round (adjustment by trial-and-error) tasks across 13 simulators from fluid dynamics, solid mechanics, and plasma physics. Each simulator's cost is analytically defined and platform-independent. Frontier LLMs achieve 46–65% success rates in single-round mode, dropping to 35–55% under high accuracy requirements, rendering their initial guesses unreliable especially for high accuracy tasks. Multi-round mode improves rates to 72–81%, but LLMs are 1.5–2.5× slower than traditional scanning, making them uneconomical choices. We also investigate parameter group correlations for knowledge transfer potential, and the impact of in-context examples and reasoning effort, providing practical implications for deployment and fine-tuning. We open-source SIMULCOST as a static benchmark and extensible toolkit to facilitate research on improving cost-aware agen-

tic designs for physics simulations, and for expanding new simulation environments. Code and data are available at https://github.com/Rose-STL-Lab/SimulCost-Bench.

## 1. Introduction

Large Language Models (LLMs) show promise for scientific workflows through code generation, complex reasoning, and especially tool calling (Tang et al., 2023; Huang et al., 2024; Patil et al., 2023). By offloading computations to domain-specific tools, LLM agents reduce hallucinations through grounded outputs. This capability has accelerated workflows in scientific domains including chemistry (Bran et al., 2024), computational fluid dynamics (Pandey et al., 2025; Yue et al., 2025; Chen et al., 2024), and data science (Chen et al., 2025b; Majumder et al., 2024).

To systematically evaluate these capabilities, recent benchmarks have begun investigating LLMs for assisting science, including scientific question answering and tool usage across domains (Zhang et al., 2025; Lai et al., 2023; Xu et al., 2024; Elrefaie et al., 2025).

However, existing evaluations focus on task correctness and token costs while overlooking tool costs. Metrics like pass@k with large $k$ implicitly treat tool usage as free, which becomes impractical in realistic scientific workflows where simulations or experiments consume significant time or materials. In physics simulations, the numerical parameter choices directly impact both solution quality and cost (Arisaka & Li, 2024; Kumar & Nair, 2024; Thakur & Nadarajah, 2025). Increasing spatial or temporal resolution improves accuracy but with cost scaling quadratically or cubically. Domain experts develop intuition for these trade-offs through experience, balancing cost against quality via informed guesses and iterative refinement. Yet without mechanisms to evaluate such cost-awareness in LLMs under current pass@k and success-rate-only metrics, we risk developing agents that achieve correctness only after numerous unaffordable trials.

---
*Equal contribution.  †Work partially done at UCSD. [1]University of California San Diego [2]The Chinese University of Hong Kong, Shenzhen [3]Peking University [4]University of California, Los Angeles [5]ETH Zurich [6]California Institute of Technology. Correspondence to: Yadi Cao <yadicao95@gmail.com>, Rose Yu <roseyu@ucsd.edu>.

*Proceedings of the 43rd International Conference on Machine Learning*, Seoul, South Korea. PMLR 306, 2026. Copyright 2026 by the author(s).

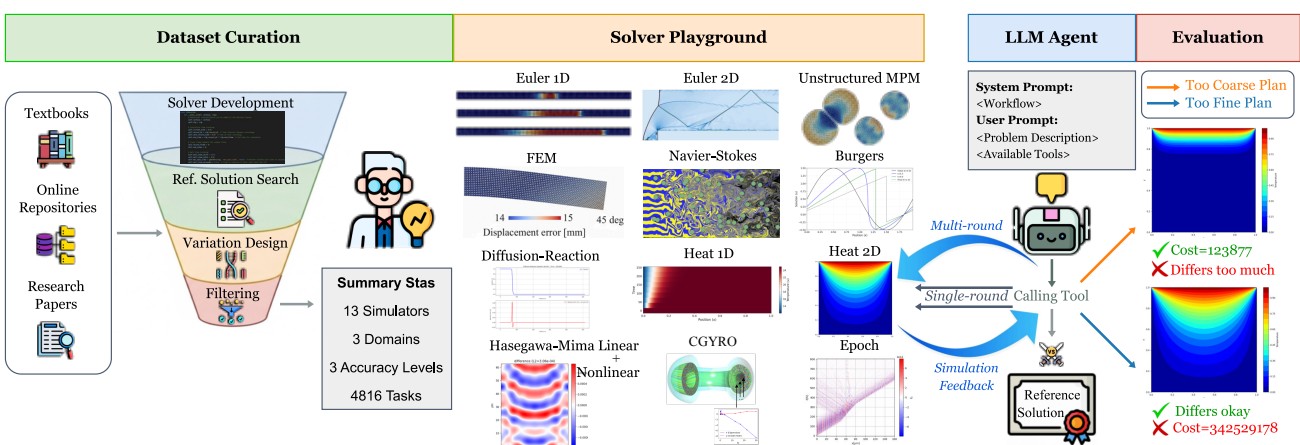

*Figure 1.* **Overview of SIMULCOST.** Our benchmark evaluates LLM agents on cost-sensitive parameter tuning across 13 physics simulators spanning fluid dynamics, solid mechanics, and plasma physics. Given a simulation task, tuning mode, and accuracy requirement, the LLM proposes tunable parameters in either single-round (initial guess) or multi-round (trial-and-error) mode. The challenge is balancing accuracy and cost: overly coarse parameters fail accuracy requirements, while overly fine parameters waste computation.

To address this gap, we present SIMULCOST, the first benchmark designed to evaluate LLMs' cost-awareness in tuning physics simulators. Unlike existing benchmarks that only measure success rate, SIMULCOST additionally evaluates computational efficiency. We quantify cost by counting core operation FLOPs for each simulation instance, which correlates with wall-clock time on serial machines. Our benchmark spans 13 simulators across fluid dynamics, heat transfer, solid mechanics, and plasma physics.

Our main contributions: (1) the first benchmark measuring both success rate and computational cost for LLM-automated physics simulations; (2) an extensible toolkit with 13 simulators featuring platform-independent cost tracking; (3) evaluation of state-of-the-art LLMs compared against brute-force scanning and Bayesian optimization; (4) ablation studies providing practical guidance on potential improvements such as knowledge transfer, in-context learning, and reasoning effort.

Through systematic evaluation, we report several key findings. (1) Frontier LLMs achieve 46–65% success rates in single-round mode, dropping to 35–55% under high accuracy requirements. This indicates that LLMs' initial guesses are unreliable and only useful for quick previews when users lack parameter intuition. (2) Multi-round mode improves success rates to 72–81%, making it necessary for high-accuracy tasks. However, LLM trial-and-error is 1.5–2.5× slower than brute-force scanning, suggesting practitioners should let LLMs invoke scanning algorithms rather than relying on internal reasoning alone. (3) Common parameters like spatial resolution are easier to tune than solver-specific ones like convergence tolerances. However, there is little correlation between individual tunable parameters even within the same type, indicating that fine-tuning on cheaper simulators is unlikely to transfer to expensive ones.

(4) In-context learning improves single-round success by 15–25% but anchors models to demonstrated parameter regimes, degrading multi-round performance. This indicates naive retrieval-augmented generation may not be a complete solution.

We hope SIMULCOST will serve as a pioneering benchmark for evaluating cost-awareness in LLM-based scientific agents. Respecting tool cost is crucial for realistic workflows where each experiment requires significant computational resources or physical materials. We will open-source SIMULCOST upon publication, including the static benchmark and extensible toolkit, enabling the community to both explore improvements for the cost-awareness of LLM-based scientific agents and create new simulation environments.

## 2. Related Work

**LLMs Benchmarks for Science: Knowledge-Based vs. Tool-Use.** LLM benchmarks for science fall into two categories: scientific knowledge evaluation and tool-using agents. The first category includes QA benchmarks such as SciBench (Wang et al., 2024), SciEx (Dinh et al., 2024), and APBench (Wu et al., 2025b), which evaluate reasoning across mathematics, physics, chemistry, and astrodynamics. These benchmarks are limited to problems with closed-form, hand-calculatable solutions. Realistic scientific workflows require complex external tools like numerical simulations, motivating the second category: LLMs as scientific agents (Ren et al., 2025). Domain-specific systems like ChemCrow (Bran et al., 2024) integrate 18 chemistry tools for synthesis and drug discovery, while OpenFOAMGPT (Pandey et al., 2025), Foam-Agent (Yue et al., 2025), and CFDLLMBench (Somasekharan et al., 2025) target computational fluid dynamics through work-

*Table 1.* Comparison of benchmarks for LLM agents in scientific and engineering domains. Tool Cost: whether it tracks computational costs of external tools (separate from LLM inference). Platform Indep.: whether tool cost measurement is platform-independent (not based on wall time). Toolkit: whether it provides an extensible simulation environment. –: not applicable (no tool cost). *Ambiguous: costs are reported but mixed with other components or based on wall time. †Claims computational cost but reports only LLM token costs in USD.

| Benchmark | Domain | Tool Cost | Platform Indep. | Toolkit |
|---|---|---|---|---|
| TaskBench (Shen et al., 2024) | General Tools | ✗ | – | ✗ |
| CATP-LLM (Wu et al., 2025a) | General Tools | ✓ | ✗ | ✗ |
| ScienceAgentBench (Chen et al., 2025b) | Data Science | ✗ | – | ✗ |
| DiscoveryBench (Majumder et al., 2024) | Data Science | ✗ | – | ✗ |
| Auto-Bench (Chen et al., 2025a) | Causal Discovery | ✓ | ✓ | ✗ |
| MLE-Bench (Chan et al., 2025) | ML Engineering | ✓* | ✗ | ✗ |
| MLGym (Nathani et al., 2025) | ML Research | ✓* | ✗ | ✗ |
| Agent-Lab (Schmidgall et al., 2025) | ML Research | ✓* | ✗ | ✗ |
| BioML-bench (Science Machine, 2025) | Biomedical ML | ✓* | ✗ | ✗ |
| ChemCrow (Bran et al., 2024) | Chemistry | ✗ | – | ✗ |
| AstaBench (Bragg et al., 2025) | Scientific Research | ✗† | – | ✗ |
| CFDLLMBench (Somasekharan et al., 2025) | CFD | ✗ | – | ✓ |
| **SIMULCOST (Ours)** | **Physics Simulations** | ✓ | ✓‡ | ✓ |

‡11/13 solvers; EPOCH and CGYRO use wall-time (see Section 5).

flow automation and knowledge evaluation. ScienceAgentBench (Chen et al., 2025b) and DiscoveryBench (Majumder et al., 2024) target data science workflows. This work focuses on the latter category, LLMs using verified simulation tools, and addresses the missing cost dimension in existing benchmarks.

**The Missing Tool Cost Consideration.** Despite growing interest in scientific agents, most benchmarks neglect the cost of tool execution, a critical factor in real scientific workflows where tool costs (such as simulation time) dominate over LLM API costs. For example, ScienceAgentBench (Chen et al., 2025b), DiscoveryBench (Majumder et al., 2024), and most agentic systems (Nathani et al., 2025; Kapoor et al., 2024) track only LLM token costs. AstaBench (Bragg et al., 2025) claims "computational cost" but reports only token costs in USD. Works such as CodePDE (Li et al., 2025) and SciCode (Tian et al., 2024) combine code generation with pass@K metrics. While valuable for assessing code synthesis capabilities, these benchmarks address a different challenge than properly using existing tools, and do not consider execution cost as a metric.

Several benchmarks address this gap by recording wall-time, e.g., Agent-Lab (Schmidgall et al., 2025), MLE-Bench (Chan et al., 2025), MLGym (Nathani et al., 2025), and BioML-bench (Science Machine, 2025). However, these measurements conflate LLM inference with tool execution time and vary across hardware, making cross-study comparisons unreliable. Auto-Bench (Chen et al., 2025a) achieves platform-independence by counting interventions as the tool cost, but this assumes uniform cost per intervention and remains in the data science domain. CATP-

LLM (Wu et al., 2025a) models cost-aware tool planning via function-as-a-service, but is limited to generic tools with relatively fixed costs per function, which breaks down for simulations where parameters like grid resolution alter cost of the same function by orders of magnitude. MetaOpenFOAM (Chen et al., 2024) is the only prior work targeting physics simulations with cost awareness. However, it is a workflow automation framework rather than a comprehensive benchmark, and records wall-time that conflates LLM inference with solver execution.

SIMULCOST bridges these gaps by defining cost based on computational complexity analysis of each solver, providing platform-independent measurements with guaranteed reproducibility. We focus on parameter tuning of physics simulations where the parameters of interest fundamentally influence both simulation accuracy and computational cost. Table 1 summarizes our distinctions.

**Traditional Parameter Tuning Methods and Benchmarks.** The tasks in SIMULCOST fall within the parameter tuning category in scientific software usage. Traditional approaches range from grid search (scan) and constrained optimization to Bayesian methods (Goodfellow et al., 2016; Snoek et al., 2012), including tree-structured Parzen estimators (Bergstra et al., 2011), SMAC with random forest surrogates (Hutter et al., 2011), and multi-fidelity approaches like Hyperband (Li et al., 2018) and BOHB (Falkner et al., 2018). Related benchmarks like Design-Bench (Trabucco et al., 2022) and Inverse-Bench (Zheng et al., 2025) target case-by-case parameter optimization but consider cost implicitly as optimization steps, which only holds for tools with uniform costs. HPOBench (Eggensperger et al., 2021)

and YAHPO (Pfisterer et al., 2022) explicitly track evaluation costs but focus on ML hyperparameters. We adopt brute-force scanning as our reference and include a standard GP-based Bayesian optimization baseline (Section 3.5). A comprehensive comparison against the full HPO toolkit is an important direction for future work.

# 3. The SimulCost Toolkit

## 3.1. Overview

We present SIMULCOST, the first comprehensive benchmark for evaluating LLM agents' success rate and cost-efficiency in physics simulation parameter tuning (see Figure 1 for a schematic overview). Our 13 simulators span fluid dynamics, solid mechanics, and plasma physics. We evaluate both *single-round* inference, assessing physics and numerical intuition, and *multi-round* inference, testing adaptive parameter optimization through solver interaction (Section 3.3). Beyond static evaluation, we open-source the complete solver library as an extensible Toolkit for benchmark extension (Section 3.2).

Each solver presents LLMs with parameter tuning tasks: given a physics-based simulation scenario, the LLM must select parameter values that satisfy accuracy requirements while minimizing computational cost. In reality, simulators usually require tuning multiple parameters for a successful execution. However, domain experts usually isolate potential issues upon seeing unsuccessful simulation runs and tune the corresponding parameters one by one. We adopt this pattern and constrain the LLM to tune individual parameters while fixing others to reasonable (but non-optimal) values chosen by rule-of-thumb practices (e.g., CFL= 0.25 for explicit time-stepping). This isolation avoids the complexity of multi-parameter optimization and enables a meaningful grid search (scan) baseline. For each solver, we define the tool cost using computational complexity analysis by counting dominant operations FLOPs, which strongly correlates with wall time on a single-core processor; we defer each solver's cost formulas to Appendix A.2. We also include two production codes: EPOCH (Arber et al., 2015) for laser-plasma particle-in-cell simulation and CGYRO (Candy et al., 2016) for gyrokinetic transport. Since both are compiled binaries whose FLOPs cannot be estimated analytically, we use wall-clock time on fixed hardware as their cost metric (Appendix A.2). We evaluate LLM performance at **three accuracy levels** (low, medium, high) and create task variations by combining different initial/boundary conditions, environmental variables, and non-target parameter settings, and the accuracy level requirement (Appendix A.1).

In all, SIMULCOST comprises **2,947 single-round tasks** and **1,931 multi-round tasks** across 13 solvers (Table 2).

## 3.2. Dataset Curation

Our curation pipeline has four phases (detailed in Appendix A.1): (1) **Solver development** - developing new or adapting existing solvers, (2) **Reference solution search** - finding reference parameters for each task using brute-force search (Algorithms 1 and 2), (3) **Variation design** - scaling up task diversity by combining accuracy thresholds (low/medium/high), initial/boundary conditions, environments, and non-target parameter settings, and (4) **Filtering** - since experts define the accuracy thresholds consistently for all combinations, some may fail to find a solution that meets the threshold within the maximum scan iterations. We filter these infeasible tasks post-hoc to avoid unnecessarily high computational time for later LLM evaluations.

**Expert contributions.** Our expert team includes 11 experts in fields related to physics, mechanical engineering, and nuclear engineering: 2 professors, 3 postdocs, 2 doctoral students, and 5 undergraduates. Experts provided physics background knowledge, developed or adapted solvers, designed parameter tuning tasks, scaled-up scenario variations, and established accuracy thresholds. They documented specifications in a consistent markdown format, portions of which convert to LLM prompts (Appendix B.3). All experts received coding and documentation training before curation. We followed institutional ethical standards and compensated all participants.

**Data sources.** All solvers are sourced from textbooks, online repositories, and research papers (per-solver details in Appendix A.2). Original scripts typically contain bugs or lack cost-related metadata, requiring expert debugging and adaptation for quality assurance, cost calculation, and consistent API design.

**Quality validation.** Solvers are cross-validated by a separate expert when domain experts submit pull requests for developed solvers. We also verify the optimal parameters found by the scan algorithms to ensure they have diverse distributions, preventing exploitation of biased reference solutions.

**Task groups.** To analyze performance patterns across parameter types, we categorize tunable parameters into four groups based on their role in simulation: **Spatial**: grid resolution parameters such as `nx` and `dx` that control discretization density, appearing in all solvers; **Temporal**: timestep parameters such as `dt` and `cfl` that control time integration step; **Tolerance**: convergence tolerance parameters such as `tol` and `error_threshold` that determine when iterative solvers terminate early; and **Misc**: physics-specific parameters such as limiter coefficients in reconstruction and relaxation factors in iteration (Patankar, 1980) that vary

case-by-case. This categorization enables two analyses: (1) whether LLMs' pre-training knowledge favors common parameter types (Spatial, Temporal) over solver-specific ones (Misc), and (2) whether within-group task correlations exceed between-group correlations, which would suggest knowledge transfer potential across solvers sharing parameter types. The parameter distribution across groups is: Spatial: 8 parameters, Temporal: 3, Tolerance: 5, Misc: 7. Table 20 in the Appendix lists the complete parameter-to-group mapping.

**Toolkit.** We also open-source the complete solver library and API (`simulcost-tools`) as an extensible Toolkit for benchmark extension. Unlike static datasets, the Toolkit offers: (1) source code for all 13 solvers with standardized wrapper APIs, and (2) a configuration-consistent interface (based on Hydra (Yadan, 2019)) for defining simulation scenarios that is easy to replicate and extend. Users can modify existing scenarios by adjusting parameter ranges or accuracy thresholds, create new task variations from existing solvers, or integrate entirely new solvers following the established cost-tracking workflow.

### 3.3. Inference Modes

Expert parameter tuning in physics-based simulations requires **two distinct capabilities**: (1) physical and numerical intuition for initial parameter estimation and (2) iterative adjustment through interactions with solvers. We design two inference modes to evaluate these skills, measuring both success rate and cost efficiency:

**Single-Round Inference** requires single-attempt parameter selection using LLM's prior knowledge to find the **optimal** parameter balancing accuracy and cost. Since the true optimal in continuous parameter space is unknown, we set the reference as the **near-optimal solution** with **near-minimal cost** that meets the accuracy threshold found by scan algorithm (Algorithms 1 and 2). Since discrete scan cannot guarantee a "true optimal" in continuous parameter space, LLMs can potentially match or even exceed the reference efficiency of 1.0, indicating expert-level intuition.

**Multi-Round Inference** allows up to 10 trials. Human expertise usually only has patience for 5–10 trials before resorting to systematic sweeps, and our scan baseline uses approximately 20 grid points, so 10 trials gives LLMs half the budget of exhaustive search. Each trial's simulation feedback includes: a) convergence status, b) RMSE between the current and a finer resolution to judge convergence, and c) accumulated cost is appended to the conversation list. LLMs can terminate early if they deem a solution satisfactory. The reference cost is the **accumulated cost from the scan algorithm** (Algorithm 1). An efficiency near 1.0 here merely indicates the LLM matched brute-force scan, a low

bar since scan can be invoked even without LLM; LLMs with their reasoning ability are ideally expected to achieve higher than 1.0 efficiency.

Multi-round mode applies to **17 of 29 parameters** with monotonic cost-accuracy relationships, comprising **1,931 tasks**. The remaining 12 parameters are evaluated only in single-round mode. These excluded parameters tend to have non-monotonic cost-accuracy relationships, making them arguably the most challenging cases. However, non-monotonic parameters lack an intuitive progressive refinement path, so no simple multi-round baseline exists.

### 3.4. Evaluation Metrics

Our metrics cover both the *success rate* and the cost *efficiency* for LLM-proposed parameters. Both are computed **per task instance**, enabling meaningful within-solver comparisons without requiring cross-domain cost normalization.

**Success Rate.** Success $S_i \in \{0, 1\}$ for task $i$ indicates whether the task-dependent distance between the LLM-proposed simulation output and the reference output (found by Algorithms 1 or 2) falls within the accuracy threshold. See Table 2 for per-solver metric definitions and thresholds.

**Efficiency.** Efficiency measures how well the LLM's cost compares to the reference cost:

$$E_i = \frac{C_i^{\mathrm{bf}}}{C_i^{\mathrm{sr,mr}}} \times S_i, \qquad (1)$$

where $C_i^{\mathrm{bf}}$ is the brute-force reference cost, $C_i^{\mathrm{sr,mr}}$ is the LLM's cost, and superscript "sr" or "mr" denotes single-round or multi-round mode respectively. For single-round, $C_i^{\mathrm{bf,sr}}$ is the near minimum-cost solution; for multi-round, $C_i^{\mathrm{bf,mr}}$ is the accumulated scan cost.

For aggregated results, we report arithmetic mean for success rate and **geometric mean** for efficiency over successful samples only ($E_i > 0$). The geometric mean is standard practice for aggregating performance ratios and speedups (Henning, 2006; Mattson et al., 2020): it is scale-invariant, handles values spanning multiple orders of magnitude without being dominated by outliers, and treats multiplicative relationships symmetrically. Failed tasks ($E_i = 0$) are excluded from efficiency aggregation.

### 3.5. Bayesian Optimization Baseline

For multi-round mode, we compare LLMs against Bayesian Optimization with Gaussian Process surrogate (BO-GP), a classical black-box optimization approach. Single-round BO would reduce to random sampling since the GP requires prior observations. Implementation details are in Appendix C.9.

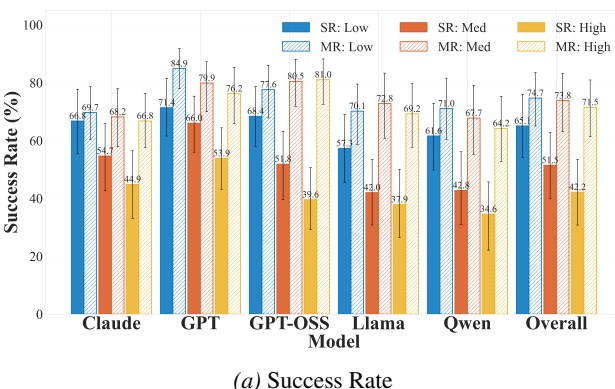

*(a)* Success Rate

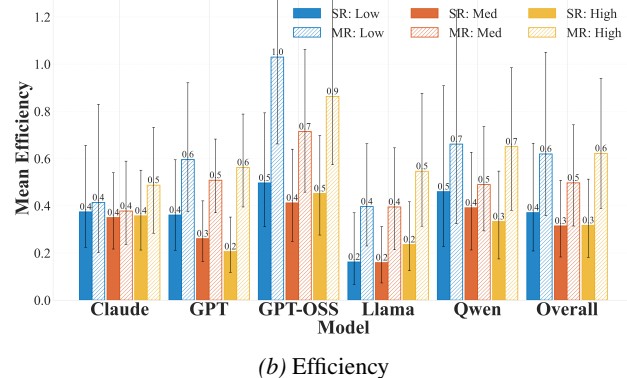

*(b)* Efficiency

*Figure 2.* Overall performance across models, accuracy levels, and inference modes. SR = Single-Round, MR = Multi-Round; High/Med/Low = accuracy level requirements.

## 3.6. Ablation Study Setup

Following standard practice for ablation studies, we evaluate on a core subset rather than the full benchmark. Specifically, we use **5 datasets** (Euler 1D, Heat 1D, MPM 2D, NS Transient 2D, HM Nonlinear) spanning fluid dynamics, solid mechanics, and plasma physics domains.

**In-Context Learning (ICL) Variants.** In practice, simulation teams accumulate experiment logs that could serve as retrieval-augmented context, a realistic RAG setting where past runs inform new parameter choices. We design three ICL variants to isolate specific information contributions: (1) **Idealized ICL**: accuracy-matched examples with full cost information, representing a well-curated experiment log; (2) **Cost-Ignorant ICL**: same accuracy-matched examples but without cost data, testing whether cost awareness is essential; and (3) **Mixed-Accuracy ICL**: examples from all accuracy levels without matching, simulating a realistic log where historical runs target varying accuracy requirements. This ablation is evaluated with Claude-3.7-Sonnet only. Based on our results, this model is a mid-tier performer, ranked 2nd of 5 models in single-round success rate, with no significant gaps to top or bottom models.

**Reasoning Effort Variants.** A natural hypothesis is that deliberate reasoning about tunable parameters can narrow the search space and improve initial guesses or explorations. GPT-5 exposes a "reasoning effort" parameter controlling the depth of chain-of-thought reasoning before answering. We compare Minimal and High settings against the default (Medium) to test whether extended reasoning improves parameter tuning in both single-round and multi-round settings.

## 4. Experiments

### 4.1. Experimental Setup

We evaluate state-of-the-art LLMs including GPT-5-2025-08-07, Claude-3.7-Sonnet-2025-02-19, GPT-OSS-120B, Llama-3-70B-Instruct, and Qwen3-32B across 13 datasets spanning fluid dynamics, solid mechanics, and plasma physics. Our evaluation encompasses both single-round and multi-round inference modes at three accuracy levels (low, medium, high). Temperature is set to 0.0 for all models except GPT-5, which uses its default setting. Additionally, we evaluate GPT-5's reasoning effort parameter (Minimal, Medium, High).

### 4.2. Main Results

Figure 2 presents overall performance across models and accuracy levels.

In single-round mode, success rates vary substantially across models (46–65%), with GPT-5 leading at 65.0%. Even this best rate falls short of practical reliability, requiring manual intervention every 2–3 simulations. Multi-round inference substantially improves all models, with GPT-5 and GPT-OSS-120B both reaching ∼81% and even Llama-3-70B reaching 72%.

Regarding performance across different accuracy levels, single-round success drops sharply as accuracy requirements tighten (66% → 42%), while multi-round remains stable (76% → 72%).

Efficiency interpretation differs by mode: in single-round, efficiency >1.0 means the LLM beat the optimal solution's cost. In multi-round, the reference is cumulative brute-force cost, so efficiency ≈1.0 merely matches naive search. In single-round, all models achieve 0.19–0.50 efficiency, using 2–6× optimal compute. In multi-round, only GPT-OSS-120B approaches 1.0 (at low accuracy), while other models

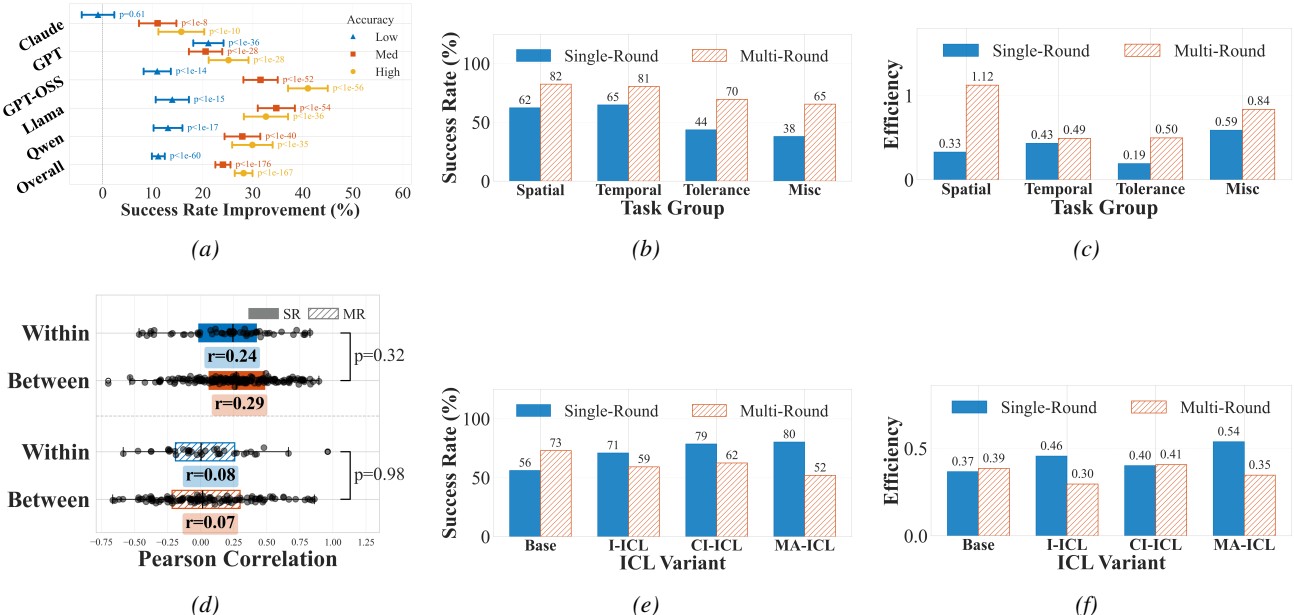

*Figure 3.* Key findings. (a) Forest plot of multi-round improvement by model and accuracy; all differences significant at $p < 0.05$ except Qwen3-32B at low accuracy. (b)–(c) Task group summary: aggregated success rate and efficiency by parameter type; single-round (solid) vs. multi-round (hollow). (d) Task correlations: within-group vs. between-group shows no significant difference. (e)–(f) In-context learning (ICL) variants improve single-round success but degrade multi-round; cost information in examples preserves efficiency.

cluster around 0.4–0.7, taking 1.5–2.5× brute-force cost.

### 4.3. Single-Round vs. Multi-Round Comparison

Figure 3(a) quantifies the improvement from multi-round inference using McNemar's test on paired samples. We observe that the success rate gain is most pronounced at high accuracy levels, with a mean improvement of +28.2% across models ($p < 0.001$), precisely where single-round struggles most. Lower accuracy levels also benefit, but to a lesser extent. This asymmetry has a clear explanation: higher accuracy requirements narrow the acceptable parameter range, making "lucky guesses" increasingly unlikely. Multi-round mode compensates through trial-and-error, making it necessary for high-accuracy tasks. However, as noted above, LLM trial-and-error is slower than brute-force, so practitioners should let LLMs invoke scanning algorithms in this setting.

See detailed statistics in Table 8 and robustness analysis in Appendix C.3.

### 4.4. Task Group Analysis

Using the parameter categorization from Section 3.2, we analyze performance across Spatial, Temporal, Tolerance, and Misc parameter groups (Figures 6a–6b).

Success rate varies substantially across parameter groups in single-round mode (29–70%). Spatial and Tolerance parameters cluster at the high end, likely benefiting from

predictable cost-accuracy trade-offs in pre-training data, while Misc parameters show the widest variation due to their solver-specific nature. Multi-round mode compresses this variation (all groups: 61–79%), with Misc parameters gaining most (+23%). This pattern suggests that iterative exploration compensates for missing prior knowledge, benefiting uncommon parameters the most.

Efficiency patterns reveal a paradox: in single-round, Misc parameters (the hardest to solve) achieve near-optimal efficiency when successful, while Spatial parameters lag far behind. This suggests LLMs can find runnable values for common parameters like grid resolution but lack cost intuition, tending toward "safer" values that work but waste compute. In multi-round, Misc parameters exceed 1.0 at low accuracy and slightly outperform exhaustive search, while other groups still remain below.

We analyzed within- vs. between-group task correlations to assess knowledge transfer potential. Neither single-round nor multi-round showed significant within-group correlation advantages ($p$=0.35 and $p$=0.97, respectively), implying task difficulty is parameter-specific rather than type-driven. This limits the viability of fine-tuning on cheap simulators to improve performance on expensive ones.

Full plots and correlation matrices appear in Appendices C.11–C.12.

## 4.5. In-Context Learning

Given that solver-specific parameters such as the Misc group lack reliable priors, a natural question is: can we leverage historical examples to improve performance? Following the ablation setup in Section 3.6, we evaluate three ICL variants against Base (no examples) on the 5-dataset ablation subset using Claude-3.7-Sonnet: Idealized ICL (accuracy-matched examples with cost), Cost-Ignorant ICL (accuracy-matched without cost), and Mixed-Accuracy ICL (diverse examples without matching). The latter two variants simulate realistic scenarios where experiment logs may be partially lost or imperfectly matched to new requirements.

Figures 3(d)–(e) reveal a key trade-off: *ICL improves single-round success but degrades multi-round performance*. This pattern holds across all variants, suggesting that examples help narrow initial guess ranges but limit exploration, steering models to shown parameter regimes.

Comparing variants in single-round mode: *Mixed-Accuracy ICL* achieves the largest gains in both success (+24%) and efficiency (1.5×), likely because diverse examples span a wider parameter range. *Cost-Ignorant ICL* shows comparable success gains but minimal efficiency improvement, suggesting that cost information in examples is critical for efficiency.

Across task groups, Tolerance parameters benefit most from ICL with the largest gains, while Spatial parameters show modest improvement. This mirrors the single-round vs. multi-round pattern and suggests that parameters less common in pre-training benefit most from additional demonstrations.

Detailed breakdowns appear in Appendix C.4.

## 4.6. Bayesian Optimization Baseline

We compare LLM performance against Bayesian Optimization with Gaussian Process surrogate (BO-GP) across the 11 solvers with analytical cost metrics in multi-round mode. BO achieves comparable aggregate success rates to LLMs but shows higher inter-solver variance. Efficiency-wise, LLMs have an obvious advantage, especially at low accuracy requirements (2.03 vs 1.02). We found that BO's exploration strategy tends to choose extreme values early to maximize information gain. However, this triggers early stopping if it selects the "finer" side of extreme bounds, resulting in high cumulative cost. In contrast, LLMs leverage physics intuition from pre-training to make more informed initial guesses, which is especially helpful at low accuracy requirements where thresholds are more lenient.

Full results appear in Tables 18 and 19. Implementation details are in Appendix C.9.

## 4.7. Additional Results

We summarize additional findings with detailed analyses in the Appendix. **Reasoning Effort**: GPT-5's reasoning effort parameter shows no significant overall impact, as increased reasoning does not lead to better parameter selection (Appendix C.5). **Failure Modes**: We identify five recurring failure patterns (false positives, blind exploration, instruction misunderstanding, prior bias, and conservative strategy) to guide future improvements (Appendix B.6). These patterns reveal a tension between cost-awareness and correctness: LLMs either skip intermediate resolutions to save cost but land on non-converged solutions (blind exploration), or choose unnecessarily fine resolutions "to be safe" while making tiny adjustments that hinder exploration (conservative strategy). **Additional Baselines**: Classical optimization methods (Bisection, Nelder-Mead) achieve similar success rates to brute-force at comparable cost, justifying our baseline choice (Appendix C.6). **Robustness**: Prompt sensitivity analysis (25 variants) and unconditional cost metrics confirm our findings are stable (Appendix C.7). **Tool-Augmented Tuning**: When LLMs provide initial guesses for search algorithms, success improves (+8.6%) but cost doubles due to conservative parameter choices (Appendix C.8).

# 5. Limitations

Our benchmark restricts LLMs to text-based solver interactions without access to auxiliary tools. Real-world agents might leverage log parsing or custom timeout logic, but this constraint ensures fair comparison: many LLMs do not yet support full agentic mode reliably, and the choice of framework introduces confounding variables. In practice, our expert curation filters out unstable solver configurations, and we enforce hard timeouts for potentially long-running cases, so failure recovery is rarely needed.

Additionally, we isolate single-parameter tuning rather than joint multi-parameter optimization. Baseline tractability drives this choice: grid search complexity grows as $O(n^k)$ where $n$ is the number of choices per parameter and $k$ is the number of parameters. With $n = 20$ grid points, single-parameter search requires 20 evaluations, while 3-parameter joint search requires $20^3 = 8,000$, making exhaustive reference solutions prohibitively expensive. Bayesian optimization could reduce cost but does not handle integer parameters natively and introduces its own tuning decisions (surrogate model, acquisition function). As the first benchmark that considers tool cost in this domain, we opt for a simple, assumption-free reference and only include one typical Bayesian-based approach as a baseline.

# 6. Conclusions and Future Directions

We introduced SIMULCOST, the first benchmark for evaluating cost-aware parameter tuning capabilities of LLMs in physics simulations. Our evaluation spans 13 physics simulators and 5 state-of-the-art LLMs. We also open-source SIMULCOST upon publication, including the static benchmark and extensible toolkit, enabling the community to both explore improvements for the cost-awareness of LLM-based scientific agents and create new simulation environments.

**Practical recommendations.** Based on our findings: (1) LLMs' initial guesses are generally unreliable (46–65% success) and only useful for quick previews when users lack parameter intuition, accuracy requirements are low, and optimal cost efficiency is not needed. (2) For high-accuracy tasks or finding a guaranteed cost-efficient configuration, multi-round mode becomes necessary (72–81% success), but practitioners should let LLMs invoke scanning algorithms rather than relying on internal reasoning alone, as the latter is $1.5$–$2.5\times$ slower. (3) Fine-tuning on cheaper simulators is unlikely to transfer to expensive ones due to the lack of cross-parameter correlation, even within the same parameter type. (4) ICL improves single-round performance but anchors models to demonstrated regimes, degrading multi-round exploration. Hence, naive RAG may not be a complete solution. (5) Using examples from a diverse range of accuracy requirements helps exploration, and including cost information helps optimize cost efficiency. (6) When using LLMs to initialize search algorithms, guide toward intermediate parameter values; our tool-augmented ablation shows conservative initial guesses can undershoot optimal by 50–80%, incurring $5$–$14\times$ cost overhead (Appendix C.8).

**Future directions.** Several extensions would strengthen cost-aware scientific agents: (1) *Tool-augmented tuning*: equipping LLMs with timeout-based early stopping, callable search algorithms, and multi-modal feedback such as field visualizations to enable richer decision-making. Our tool-augmented ablation (Appendix C.8) shows that initial guess quality significantly affects downstream search efficiency, suggesting that analyzing LLM-developed heuristics becomes more meaningful once tool-use-search is enabled. (2) *Human-in-the-loop evaluation*: user studies measuring how LLM-suggested parameters accelerate expert workflows, other than autonomously tuning, would validate real-world utility. (3) *Cost-aware post-training*: developing fine-tuning strategies that explicitly optimize for accuracy and computational efficiency. However, we caution that unlike common software engineering where many codes share commonality, scientific tasks by definition are on the frontier of unknowns. Benchmark results relying heavily on web search or SFT may under-measure LLMs' on-the-fly adjustment ability for genuinely novel tasks. (4) *Multi-parameter optimization*: enabling LLMs to jointly tune interdependent parameters with adaptive sampling approaches. (5) *Parallel scaling*: extending cost tracking to parallel infrastructure with analytical complexity models that account for overhead and delays, or careful, reproducible wall-time measurements. (6) *Non-monotonic parameter handling*: for parameters without clear cost-accuracy monotonicity, two expert strategies exist: (a) leveraging mathematical or physics prior knowledge (e.g., analyzing iteration matrix spectra for convergence bounds), and (b) post-analysis with visualization to detect abnormal behavior such as oscillations near shock fronts. Both relate to enhanced reasoning and multimodal capabilities.

## Acknowledgments

This work is supported in part by the U.S. Army Research Office under Army-ECASE award W911NF-07-R-0003-03, the U.S. Department Of Energy, Office of Science, ARPA-H-SOL-24-101 program, IARPA HAYSTAC Program, DARPA YFA, NSF Grants #2205093, #2146343, #2134274, #2441832 and CDC-RFA-FT-23-0069.

## Impact Statement

This paper presents work whose goal is to advance the field of Machine Learning. There are many potential societal consequences of our work, none which we feel must be specifically highlighted here.

## Author Contributions

Y. Cao led the project and major solver development, and contributed to conceptual design. S. Lai led LLM evaluation engineering. Y. Zhang led Bayesian optimization and active learning evaluation engineering. J. Huang, Z. Lawrence, R. Bhakta, I. F. Thomas, M. Cao, and Y. Zhao contributed to solver development. C.-H. Tsai contributed to LLM evaluation engineering. Z. Zhou contributed to conceptual design, LLM experimental design, and open-source support. H. Liu contributed to conceptual design. A. Marinoni and A. Arefiev contributed to conceptual design and solver development. R. Yu contributed to conceptual design and computing resource management. All authors contributed to discussion and manuscript revision.

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

# A. Dataset Details

## A.1. Data generation pipeline

**Overall Module Design.** We designed standardized APIs for different components of this toolkit, namely: (a) the solver module that encapsulates the timestep of a simulator, tracks meta information related to computational cost, and dumps the physical fields at intermediate timesteps, (b) the runner module that supports issuing simulation runs using shell commands with support for different scenario variations, (c) the wrapper module that provides the API interface for brute-force search and LLM function calls, including simulation results query and comparison functions for convergence check, (d) the brute-force search module that implements the search algorithm to find reference solutions. After code development and search are conducted, (e) we also have a standardized markdown template to document the physics background, numerical solution details, tunable parameters, three different levels of accuracy requirements, and scenario variations. Portions of this document convert to LLM prompts.

Each solver's preparation and the tunable task generation follow a standardized pipeline:

**1. Solver Development.** An expert starts developing or adapting the solver for given physics simulation scenarios by either developing from scratch, or adapting source code from textbooks, online repositories, and research papers. Development is achieved by either deriving from a base template solver class and fulfilling the abstract functionalities mentioned in the overall design, or compiling the source code and wrapping the run script into the runner module. We require experts to create transparent cost calculation rules following complexity analysis, and therefore add additional meta information to track cost calculations. All intermediate physical fields are cached to disk to enable reuse without incurring repeated runs. Experts are also required to provide an API for comparison between two simulated physical fields to check their distance, where the metric formulation and threshold values are case-dependent.

**2. Optimal Solution Search and Reference Setup.** For each task, we find optimal solutions that meet accuracy requirements with the lowest cost. We use two search strategies depending on parameter characteristics. For **monotonic parameters** (where cost and accuracy have a monotonic relationship, e.g., spatial resolution, time step size), we use progressive refinement (Algorithm 1): start with a coarse value and progressively refine with fixed ratios (e.g., halve the time step size, double the spatial resolution) until the distance between adjacent runs is within the accuracy threshold. For **non-monotonic parameters** (e.g., relaxation factor in SOR), we use exhaustive grid search (Algorithm 2): discretize the search space within a feasible range using uniform grids, run simulations on all choices, and pick the converged run with the lowest cost. For single-round tasks, the reference cost is the optimal cost found by search. For multi-round tasks (monotonic parameters only), the reference cost is the accumulated cost incurred during progressive refinement.

**3. Variation Design.** We scale up the simulation scenarios' diversity via multiple types of variations: a) different engineering benchmarks (e.g., different initial and boundary conditions, different physical properties, etc.), b) given a tunable parameter as the search target, the other non-target tunable parameters can form a combinatorial space (e.g., in a Finite Volume based method, if the target parameter is the spatial resolution, the non-target parameters can include the spatial discretization ordering, the slope limiter, etc.), c) different accuracy thresholds (low, medium, high) to reflect real-world varying design tolerances. The combination of these variations creates a large number of and diverse tasks from a limited number of classical benchmarks.

**4. Filtering.** We conduct optimal solution search on all variations and only keep those where brute-force search succeeded in finding the optimal solution. This ensures all selected tasks have a reference cost for comparison. During the pull request phase of each solver, we request a separate expert to conduct quality checks. In addition, statistical analysis of optimal parameters is conducted to ensure diverse optimal parameter distributions and prevent models from memorizing "golden values."

## A.2. Physics Backgrounds

**Heat Transfer 1D (Patankar, 1980).** This solver addresses the 1D heat conduction equation:

$$\frac{\partial T}{\partial t} = \alpha \frac{\partial^2 T}{\partial x^2}$$

using explicit finite difference methods with natural convection boundary conditions at $x = 0$ and adiabatic conditions at $x = L$. The tunable parameters include the spatial resolution (`n_space`) and the CFL number (`cfl`) that determines the

---

**Algorithm 1** Brute-Force Grid Search with Progressive Refinement

---

1: **input** Domain $\Omega$, base grid size $n_0$, max level $L$, threshold $\tau_{s,k}$, tolerance $\varepsilon_s$
2: **output** Best feasible $\hat{\theta}$
3: Total cost $C \leftarrow 0$
4: **for** $\ell = 0, \ldots, L$ **do**
5: $\quad n_\ell \leftarrow n_0 2^\ell$, construct grid $\mathcal{G}_\ell$
6: $\quad$ **for all** $\theta \in \mathcal{G}_\ell$ **not yet evaluated do**
7: $\quad\quad$ Solve subproblem at $\theta$ with cost $T_{\text{solve}}(\theta)$
8: $\quad\quad$ Evaluate $E_s(\theta; \sigma), c_s(\theta; \sigma)$ with cost $T_{\text{eval}}(\theta)$
9: $\quad\quad$ $C \leftarrow C + T_{\text{solve}}(\theta) + T_{\text{eval}}(\theta)$
10: $\quad$ **end for**
11: $\quad \mathcal{F}_\ell \leftarrow \{\theta \in \mathcal{G}_\ell : E_s(\theta; \sigma) \leq \tau_{s,k}\}$
12: $\quad$ **if** $\mathcal{F}_\ell \neq \varnothing$ **then**
13: $\quad\quad \hat{\theta}_\ell \leftarrow \arg\min_{\theta \in \mathcal{F}_\ell} c_s(\theta; \sigma)$
14: $\quad\quad$ Refine $\mathcal{G}_{\ell+1}$ around $\hat{\theta}_\ell$
15: $\quad$ **end if**
16: $\quad$ **if** $\Delta_\ell \leq \varepsilon_s$ **then**
17: $\quad\quad$ **break**
18: $\quad$ **end if**
19: **end for**
20: **return** $\hat{\theta}$, total cost $C$

---

**Algorithm 2** Exhaustive Grid Search for Non-Monotonic Parameters

---

1: **input** Domain $\Omega$, grid size $n$, threshold $\tau_{s,k}$
2: **output** Best feasible $\hat{\theta}$, total cost $C$
3: Total cost $C \leftarrow 0$
4: Construct uniform grid $\mathcal{G}$ over $\Omega$ with $n$ points
5: **for all** $\theta \in \mathcal{G}$ **do**
6: $\quad$ Solve subproblem at $\theta$ with cost $T_{\text{solve}}(\theta)$
7: $\quad$ Evaluate $E_s(\theta; \sigma), c_s(\theta; \sigma)$ with cost $T_{\text{eval}}(\theta)$
8: $\quad$ $C \leftarrow C + T_{\text{solve}}(\theta) + T_{\text{eval}}(\theta)$
9: **end for**
10: $\mathcal{F} \leftarrow \{\theta \in \mathcal{G} : E_s(\theta; \sigma) \leq \tau_{s,k}\}$
11: **if** $\mathcal{F} \neq \varnothing$ **then**
12: $\quad \hat{\theta} \leftarrow \arg\min_{\theta \in \mathcal{F}} c_s(\theta; \sigma)$
13: **else**
14: $\quad$ Mark task as unsolved
15: **end if**
16: **return** $\hat{\theta}$, total cost $C$

---

simulation time step by:

$$\Delta t = \texttt{cfl} \times \frac{(\Delta x)^2}{2\alpha},$$

where $\alpha$ is the thermal diffusivity. The computational cost follows the relationship $C = \texttt{n\_space} \times \texttt{n\_t}$, where $\texttt{n\_t}$ is the number of time steps accumulated in the solver. The metric for convergence is the RMSE of the heat flux at the convection boundary at the final time step. This simulation has 25 different profiles with varying initial uniform temperatures and physical properties, generating 295 tasks in total.

**Steady State Heat Transfer 2D (Patankar, 1980).** This solver tackles the 2D steady-state heat conduction equation with Dirichlet boundary conditions:

$$\nabla^2 T = \frac{\partial^2 T}{\partial x^2} + \frac{\partial^2 T}{\partial y^2} = 0$$

using the Jacobi method with Successive Over-Relaxation (SOR). The tunable parameters include the grid size $\texttt{dx}$, the relaxation factor $\texttt{relax}$, the termination residual threshold $\texttt{error\_threshold}$, and the uniform initial temperature guess $\texttt{T\_init}$. The computational cost scales as $C = (1/dx)^2 \times \texttt{n\_iter}$, where $\texttt{n\_iter}$ is the number of iterations until termination. The convergence is evaluated through the RMSE of the temperature field. The dataset comprises 8 different wall temperature combinations, generating a total of 662 tasks.

**Burgers 1D (Patankar, 1980).** This solver addresses the 1D Burgers equation:

$$\frac{\partial u}{\partial t} + u \frac{\partial u}{\partial x} = \nu \frac{\partial^2 u}{\partial x^2}$$

using a second-order Roe method for accurate shock and rarefaction wave capturing. The tunable parameters mirror those of the Euler 1D solver: CFL number ($\texttt{cfl}$) that determines the simulation time step by:

$$\Delta t = \texttt{cfl} \times \frac{\Delta x}{|u|_{\max}},$$

where $|u|_{\max}$ is the maximum velocity magnitude, spatial resolution ($\texttt{n\_space}$), the limiter parameter $\texttt{beta}$ for generalized minmod flux limiter, and the blending parameter $\texttt{k}$ between 0-th and 1-st order interpolation scheme. The computational cost follows the relationship $C = \texttt{n\_space} \times \texttt{n\_t}$, where $\texttt{n\_t}$ is the number of time steps accumulated in the solver. The convergence metrics cover both the RMSE of the solution fields, and the physical conservation principles: mass conservation, energy non-increasing property, and total variation (TV) non-increasing for stability. The dataset includes 5 different initial conditions (sinusoidal wave, Sod tube, rarefaction, blasting, and double shock wave) representing various wave interaction scenarios, generating a total of 339 tasks.

**Euler 1D (Patankar, 1980).[1]** This solver implements the 1D Euler equations for compressible flow:

$$\frac{\partial \mathbf{U}}{\partial t} + \frac{\partial \mathbf{F}(\mathbf{U})}{\partial x} = 0$$

using the MUSCL-Roe method with superbee limiter for high-resolution shock capturing. The tunable parameters include the CFL number ($\texttt{cfl}$) that determines the simulation time step by:

$$\Delta t = \texttt{cfl} \times \frac{\Delta x}{|\lambda|_{\max}},$$

where $|\lambda|_{\max}$ is the maximum eigenvalue of the flux Jacobian, the spatial resolution ($\texttt{n\_space}$), the limiter parameter $\texttt{beta}$ for generalized minmod flux limiter, and the blending parameter $\texttt{k}$ between 0-th and 1-st order interpolation scheme. The computational cost follows the relationship $C = \texttt{n\_space} \times \texttt{n\_t}$, where $\texttt{n\_t}$ is the number of time steps accumulated in the solver. Convergence is evaluated through multiple criteria: RMSE of the solution fields, positivity preservation of density and pressure, and shock consistency validation. The dataset encompasses 3 classical benchmark profiles (Sod shock tube, Lax problem, and Mach 3), generating a total of 197 tasks.

---

[1]Implementation based on https://www.psvolpiani.com/courses

**Transient Navier-Stokes 2D (Yabe et al., 2001).**[2] This solver implements the 2D transient incompressible Navier-Stokes equations:

$$\frac{\partial u}{\partial x} + \frac{\partial v}{\partial y} = 0$$

$$\frac{\partial u}{\partial t} + u\frac{\partial u}{\partial x} + v\frac{\partial u}{\partial y} = -\frac{\partial p}{\partial x} + \frac{1}{Re}\left(\frac{\partial^2 u}{\partial x^2} + \frac{\partial^2 u}{\partial y^2}\right)$$

$$\frac{\partial v}{\partial t} + u\frac{\partial v}{\partial x} + v\frac{\partial v}{\partial y} = -\frac{\partial p}{\partial y} + \frac{1}{Re}\left(\frac{\partial^2 v}{\partial x^2} + \frac{\partial^2 v}{\partial y^2}\right)$$

where $u, v$ are velocity components, $p$ is pressure, and $Re$ is the Reynolds number. The tunable parameters include the spatial resolution (`resolution`) that determines the computational grid size, the CFL number (`cfl`) controlling time step stability through $\Delta t = \text{cfl} \times \Delta x$, the relaxation factor (`relaxation_factor`) for pressure correction convergence, and the residual threshold (`residual_threshold`) for pressure solver convergence. The computational cost follows the relationship $C = 2 \times \text{resolution}^2 \times (\text{num\_steps} + \text{total\_pressure\_iterations})$, where the factor of 2 accounts for the fixed aspect ratio domain configuration with $x\_resolution = 2 \times \text{resolution}$. Convergence is evaluated through normalized velocity RMSE criteria, with temporal evolution tracked throughout the simulation. The dataset encompasses 18 benchmark profiles across 6 different boundary conditions (simple circular obstacles, complex geometries, random obstacle fields, dual inlet/outlet configurations, dense obstacle arrays, and dragon-shaped obstacles) tested at three Reynolds numbers (Re=1000, 3000, 6000), generating a total of 244 tasks across different accuracy levels and geometric complexities.

**Diffusion Reaction 1D (Patankar, 1980).** This solver implements the 1D diffusion-reaction:

$$\frac{\partial u}{\partial t} = \frac{\partial^2 u}{\partial x^2} + f(u)$$

where the reaction term $f(u)$ can be Fisher-KPP ($f(u) = u(1-u)$), Allee effect ($f(u) = u(1-u)(u-a)$), or Allen-Cahn cubic ($f(u) = u(1-u^2)$). The tunable parameters include the CFL number (`cfl`) that controls time step stability through $\Delta t = \text{cfl} \times (\Delta x)^2$, the spatial resolution (`n_space`) determining grid spacing via $\Delta x = L/\text{n\_space}$, the Newton solver tolerance (`tol`) for convergence control, the minimum step size (`min_step`) for line search robustness, and the initial step guess (`initial_step_guess`) for optimization aggressiveness. The computational cost follows the relationship $C = 3 \times \text{total\_newton\_iters} \times \text{n\_space} + \text{total\_line\_search\_iters} \times \text{n\_space}$, where the factor of 3 accounts for residual calculation, Jacobian assembly, and linear system solve in each Newton iteration. Convergence is evaluated through Newton residual norms, physical bounds preservation, and wave propagation characteristics. The dataset encompasses 3 classical reaction types (Fisher-KPP traveling waves, Allee effect threshold dynamics, and Allen-Cahn interface dynamics), generating a total of 90 tasks across different accuracy levels and numerical parameter combinations.

**EPOCH (Arber et al., 2015).**[3] This solver uses the EPOCH Particle-in-Cell (PIC) code to simulate 1D laser-plasma interactions, specifically the ionization of a carbon target irradiated by an ultra-intense laser pulse. The simulation solves Maxwell's equations:

$$\nabla \times \vec{E} = -\frac{\partial \vec{B}}{\partial t}, \quad \nabla \times \vec{B} = \mu_0\left(\vec{j} + \epsilon_0 \frac{\partial \vec{E}}{\partial t}\right)$$

coupled with relativistic particle dynamics via the Lorentz force. The tunable parameters include the grid resolution (`nx`), the CFL multiplier (`dt_multiplier`) controlling time step size through $\Delta t = \text{dt\_multiplier} \times \Delta x/c$, the number of particles per cell (`npart`), the field solver order (`field_order` $\in \{2, 4, 6\}$), and the particle weighting order (`particle_order` $\in \{2, 3, 5\}$). The computational cost is measured by wall-clock time, as EPOCH is an external compiled code where internal complexity analysis is not applicable. Convergence is evaluated through RMSE of the electric field profiles. The dataset encompasses 3 laser-target configurations with varying laser amplitudes and target densities, generating a total of 273 tasks across different accuracy levels and parameter combinations.

---

[2]Implementation based on https://github.com/takah29/2d-fluid-simulator
[3]Implementation: https://github.com/epochpic/epoch

**Euler 2D (Cao et al., 2022).** This solver addresses the 2D compressible Euler equations for inviscid gas dynamics:

$$\frac{\partial \mathbf{U}}{\partial t} + \nabla \cdot \mathbf{F}(\mathbf{U}) = 0$$

where $\mathbf{U} = (\rho, \rho u, \rho v, \rho E)^T$ are the conservative variables and $\mathbf{F}$ represents the flux tensor, with $\rho$ denoting density, $u, v$ velocity components, and $E$ specific total energy. The solver employs an advection-projection fractional-step method with third-order WENO reconstruction, local Lax-Friedrichs Riemann solver, and third-order TVD Runge-Kutta time integration. The tunable parameters include the spatial resolution (`n_grid_x`), the CFL number (`cfl`) controlling adaptive time step size through $\Delta t = \texttt{cfl} \cdot 2\Delta x / (|u| + \sqrt{u^2 + 4c^2})$ where $c$ is the local sound speed, and the conjugate gradient tolerance (`cg_tolerance`) for the pressure projection step. The computational cost follows the relationship $C = N_{\text{cells}} \times (N_{\text{steps}} + N_{\text{CG\_iters}})$, accounting for both advection steps and iterative pressure solves. Convergence is evaluated through normalized RMSE averaged over density and pressure fields. The dataset comprises 9 benchmark profiles including true 2D problems (central explosion, supersonic stair flow) and pseudo-1D shock tube problems (Sod, Lax, high Mach, strong shock, interacting blast, symmetric rarefaction), generating a total of 232 tasks across different accuracy levels and parameter combinations.

**FEM 2D (Hughes, 2003; Li et al., 2020).**[4] This solver implements the 2D Finite Element Method for solid mechanics problems using an implicit Newton solver with corotational elasticity. The solver addresses the momentum conservation equation:

$$\rho \frac{\partial^2 \mathbf{u}}{\partial t^2} = \nabla \cdot \boldsymbol{\sigma} + \rho \mathbf{g}$$

where $\mathbf{u}$ is the displacement field, $\rho$ is density, $\boldsymbol{\sigma}$ is the Cauchy stress tensor, and $\mathbf{g}$ is gravitational acceleration. The stress tensor is computed using corotational elasticity with first Piola-Kirchhoff stress $\mathbf{P} = 2\mu(\mathbf{F} - \mathbf{R}) + \lambda(J - 1)\mathbf{F}^{-T}$, where $\mathbf{F}$ is the deformation gradient, $\mathbf{R}$ is the rotation matrix from polar decomposition, $\mu$ and $\lambda$ are Lamé parameters derived from Young's modulus $E$ and Poisson's ratio $\nu$, and $J = \det(\mathbf{F})$ is the volume ratio. The spatial discretization uses triangle mesh elements with virtual work principle for force computation, while temporal integration employs an implicit backward Euler scheme with Newton-Raphson solver and line search for robustness. The tunable parameters include the mesh resolution `dx` controlling element size in the x-direction and the CFL number `cfl` determining time step size. The computational cost scales quadratically with mesh refinement due to increased element count and degrees of freedom. Convergence is evaluated through energy conservation metrics tracking the total energy variation $\text{var} = \sigma(E_{\text{tot}})/\text{mean}(|E_{\text{tot}}|)$ where $E_{\text{tot}} = E_{\text{kinetic}} + E_{\text{elastic\_potential}}$, and L2 relative differences between simulations with adjacent parameter values. The dataset encompasses 5 benchmarks including cantilever beam bending, vibration bar dynamics, and twisting column simulations, generating a total of 103 tasks across different accuracy levels and parameter combinations.

**Hasegawa-Mima (Linear) (Hasegawa & Mima, 1978; Wakatani & Hasegawa, 1984).** This solver implements the linearized Hasegawa-Mima equation for drift wave dynamics in magnetized plasmas:

$$\frac{\partial q}{\partial t} + v_* \frac{\partial \phi}{\partial y} = 0, \quad q = \nabla^2 \phi - \phi$$

where $\phi$ is the electrostatic potential, $q$ is the generalized vorticity, and $v_*$ is the diamagnetic drift velocity in a periodic 2D domain. The solver employs fourth-order Runge-Kutta (RK4) time integration with a Conjugate Gradient (CG) method for solving the Helmholtz equation $(\nabla^2 - I)\phi = q$ at each RK4 stage. The tunable parameters include the spatial resolution (`N`) determining grid spacing via $\Delta x = \Delta y = L/N$ where $L = 2\pi \times 10$, the time step size (`dt`) controlling temporal discretization over the fixed total simulation time $T = 10{,}000$, and the CG solver tolerance (`cg_atol`) for iterative convergence control. The computational cost follows the relationship $C = N_{\text{CG}} \times N^2 + N_{\text{matvec}} \times N^2$, where $N_{\text{CG}}$ is the total CG iterations across all time steps and $N_{\text{matvec}}$ counts sparse matrix-vector operations. Convergence is evaluated through L2 RMSE between numerical solutions and analytical solutions obtained via 2D FFT spectral methods. The dataset encompasses 4 different initial conditions (monopole, dipole, sin_x_gauss_y, and gauss_x_sin_y), generating a total of 306 tasks across different accuracy levels and parameter combinations.

---

[4]Implementation based on https://github.com/squarefk/FastIPC

**Hasegawa-Mima (Nonlinear) (Hasegawa & Mima, 1978; Boyd, 2001).** This solver addresses the nonlinear Hasegawa-Mima equation for drift wave turbulence in magnetized plasmas:

$$\frac{\partial q}{\partial t} + \{\phi, q\} + v_* \frac{\partial \phi}{\partial y} = 0, \quad q = \nabla^2 \phi - \phi$$

where $\{\phi, q\} = \frac{\partial \phi}{\partial x}\frac{\partial q}{\partial y} - \frac{\partial \phi}{\partial y}\frac{\partial q}{\partial x}$ is the Poisson bracket representing nonlinear advection. The solver employs a pseudo-spectral method with 2D FFT for spatial derivatives, fourth-order Runge-Kutta (RK4) time integration, and 2/3 rule dealiasing to prevent aliasing errors from quadratic nonlinearity. The tunable parameters include the spatial resolution (N) determining grid spacing via $\Delta x = \Delta y = L/N$ where $L = 2\pi \times 10$, and the time step size (dt) controlling temporal discretization over the fixed simulation duration of 10,000 time units. The computational cost follows the relationship $C = N_{\text{FFT}} \times N^2 \times \log_2(N^2)$, where $N_{\text{FFT}}$ represents the total number of FFT operations (forward, inverse, and temporal integrations) performed during the simulation. Convergence is evaluated through L2 RMSE by comparing solutions at different resolutions using linear interpolation, as no analytical solution is available for the nonlinear regime. The dataset comprises 5 initial condition profiles (monopole, dipole, sinusoidal, sin_x_gauss_y, and gauss_x_sin_y), generating a total of 110 tasks across different accuracy levels and parameter combinations.

**CGYRO (Candy et al., 2016).**[5] This solver uses the local-spectral gyrokinetic code CGYRO to solve the nonlinear gyrokinetic equations governing turbulent transport in magnetized fusion plasmas. The Fokker–Planck–Maxwell system is reduced via the gyrokinetic approximation under strong magnetization and scale separation, yielding a five-dimensional phase-space evolution equation for the gyrocenter distribution function $f$:

$$\frac{\partial f}{\partial t} + \dot{\mathbf{R}} \cdot \nabla f + \dot{v}_\parallel \frac{\partial f}{\partial v_\parallel} = C(f) + S(f, \phi, \mathbf{A}),$$

coupled self-consistently with Maxwell's equations through quasineutrality and Ampère's law. The discretization combines a mixed spectral–grid approach (Fourier in the perpendicular spatial directions, field-aligned grid along the magnetic field, and a 2D velocity-space grid in pitch angle and energy) with operator-split, semi-implicit time integration. We restrict to linear simulations, where the most unstable eigenmode is tracked until either the maximum simulation time is reached or the eigenvalue change falls below a tolerance freq_tol. The tunable parameters include the number of radial Fourier harmonics (n_radial), the number of poloidal grid points (n_theta), the number of Legendre pitch-angle meshpoints (n_xi), the number of generalized-Laguerre energy meshpoints (n_energy), the eigenvalue convergence tolerance (freq_tol), and the initial timestep size (initial_dt, or delta_t) used by the adaptive time-stepping procedure. The computational cost is measured by wall-clock time, as CGYRO is an external compiled code where internal complexity analysis is not directly accessible. Convergence is evaluated by comparing the converged complex eigenvalue (real frequency $\omega$ and growth rate $\gamma$) of a candidate run against a reference solution under tolerances $\{10^{-3}, 10^{-4}, 10^{-5}\}$ for the low, medium, and high accuracy levels. The dataset encompasses 12 plasma profiles formed by combinations of scaled minor radius ($rmin \in \{0.5, 0.7, 0.9\}$) and normalized poloidal wavenumber ($ky \in \{0.05, 0.3, 1.2, 2.5\}$), spanning both ion-mode (e.g., ITG) and electron-mode (e.g., TEM/ETG) regimes representative of DIII-D-scale tokamak discharges, generating a total of 31 tasks across different accuracy levels and parameter combinations.

**Material Point Method (Cao et al., 2025).** This solver implements solid mechanics using the explicit Material Point Method (MPM) on unstructured meshes. The MPM solves the momentum conservation equation:

$$\frac{\partial \mathbf{v}}{\partial t} + \mathbf{v} \cdot \nabla \mathbf{v} = \frac{1}{\rho} \nabla \cdot \boldsymbol{\sigma} + \mathbf{g}$$

where $\mathbf{v}$ is the velocity field, $\rho$ is density, $\boldsymbol{\sigma}$ is the stress tensor computed using corotational elasticity, and $\mathbf{g}$ is gravitational acceleration. The method combines Eulerian grid-based momentum solving with Lagrangian particle-based material tracking through particle-to-grid transfer, grid momentum update, grid-to-particle transfer, and particle advection. The tunable parameters include the background grid resolution (nx) determining spatial discretization through $\Delta x = L/\text{nx}$, the particle density per grid cell (n_part) controlling material representation, and the CFL number (cfl) for temporal stability through $\Delta t = \text{cfl}/(v_{\text{max,init}}/\Delta x)$ where $v_{\text{max,init}}$ is the initial maximum velocity. The computational cost follows the relationship $C = \sum_{\text{iteration}}(n_{\text{particles}} + n_{\text{neighbor\_communications}})$, accounting for both particle operations and particle-particle

---

[5]Implementation: https://gafusion.github.io/doc/cgyro.html

*Table 2.* Summary of solvers in the SimulCost benchmark. Single/Multi columns show task counts per mode. Single-only parameters are used exclusively in single-round evaluation; Both parameters support both modes. Low/Med/High columns show accuracy thresholds. *Thresholds vary by task type. See Appendix A.2 for physics background and citations.

| Solver | Single | Multi | Single-only Params | Both Params | Low | Med | High |
|---|---|---|---|---|---|---|---|
| Burgers 1D | 339 | 255 | `beta, k` | `cfl, n_space` | $8.0 \times 10^{-2}$ | $4.0 \times 10^{-2}$ | $1.0 \times 10^{-2}$ |
| CGYRO | 31 | 31 | – | `initial_dt, freq_tol, n_energy, n_radial, n_theta, n_xi` | $1.0 \times 10^{-3}$ | $1.0 \times 10^{-4}$ | $1.0 \times 10^{-5}$ |
| Diff-React 1D | 90 | 90 | – | `cfl, n_space, tol` | vary* | vary* | vary* |
| EPOCH | 273 | 145 | `dt_multiplier, field_order, particle_order` | `npart, nx` | 0.36 | 0.33 | 0.30 |
| Euler 1D | 197 | 148 | `beta, k` | `cfl, n_space` | $8.0 \times 10^{-2}$ | $2.0 \times 10^{-2}$ | $1.0 \times 10^{-2}$ |
| Euler 2D | 232 | 232 | – | `cfl, cg_tolerance, n_grid_x` | $8.0 \times 10^{-2}$ | $4.0 \times 10^{-2}$ | $8.0 \times 10^{-3}$ |
| FEM 2D | 103 | 103 | – | `cfl, dx` | vary* | vary* | vary* |
| HM Linear | 306 | 0 | `N, cg_atol, dt` | – | $1.0 \times 10^{-2}$ | $1.0 \times 10^{-3}$ | $5.0 \times 10^{-4}$ |
| HM Nonlinear | 110 | 0 | `N, dt` | – | $1.0 \times 10^{-2}$ | $1.0 \times 10^{-3}$ | $1.0 \times 10^{-4}$ |
| Heat 1D | 295 | 295 | – | `cfl, n_space` | $1.0 \times 10^{-2}$ | $1.0 \times 10^{-3}$ | $1.0 \times 10^{-4}$ |
| Heat Steady 2D | 662 | 470 | `relax, t_init` | `dx, error_threshold` | $5.0 \times 10^{-3}$ | $5.0 \times 10^{-4}$ | $3.0 \times 10^{-4}$ |
| MPM | 65 | 65 | – | `cfl, n_part, nx` | vary* | vary* | vary* |
| NS Transient 2D | 244 | 97 | `relaxation_factor, residual_threshold` | `cfl, resolution` | 0.60 | 0.30 | 0.15 |
| **Total** | **2947** | **1931** | | | | | |

interactions within the support radius. Convergence is evaluated through energy conservation criteria, with hybrid absolute and relative thresholds depending on mean energy magnitude, as well as positivity preservation for kinetic and potential energies. The dataset encompasses 3 benchmark profiles (cantilever beam bending under gravity, vibration bar with elastic wave propagation, and disk collision with impact dynamics), generating a total of 65 tasks across different accuracy levels and parameter combinations.

### A.3. Solver Summary

Table 2 summarizes all 13 solvers in the benchmark, including task counts for single-round and multi-round evaluation modes, distance metrics used for convergence checking, and precision thresholds at three accuracy levels.

## B. Experimental Details

### B.1. LLM Configuration and Hyperparameters

- Temperature: 0.0 (except GPT-5 which uses default)

- Max tokens: 2048

- Max function calls: 10 (multi-round), 1 (single-round)

### B.2. Computational Environment

*Table 3.* Computational environment specifications.

| Component | Specification |
|---|---|
| CPU | Intel Core i9-14900KF (24-core, 2.4–6.0GHz) |
| Memory | 64GB DDR5 (2×32GB, 5200MHz) |
| GPU | NVIDIA GeForce RTX 4090 (24GB GDDR6X) |

**Note on reproducibility**: The computational environment is not critical for replicating most results, as our cost metrics are defined analytically based on computational complexity (FLOP counts). The exceptions are EPOCH and CGYRO, which use wall-clock time as the cost metric; for these datasets, we use the above machine consistently to ensure reliable timing measurements.

### B.3. Selected Prompt Examples

**Euler 1D - Single-round - n_space**

---

## Prompt Example

### System Prompt

Your task is to find the optimal parameter, solving the 1D Euler equations for compressible inviscid flow, using a 2nd order MUSCL scheme with Roe flux and generalized superbee limiter. This serves as a simplified model for compressible fluid dynamics. You should try to minimize the total cost incurred by function calls, but your primary goal is to successfully meet the convergence criteria. You should always use the tool call function to finish the problem.

Workflow: n_space (Number of grid cells) determines the spatial discretization resolution: $\Delta x = L/n\_space$ where L is the domain length. You may **only** change 'n_space'. The value of k is **-1.0**, beta is **1.0**, cfl is **0.25**. **You must not change them!** You have only one opportunity to choose an optimal value for n_space. No trial-and-error or multi-round optimization is permitted. Your goal is to select a value that provides adequate spatial resolution while keeping computational cost reasonable.

Step 1: Make your best **one-shot** guess for n_space.
Step 2: Call the Convergence Test Function and check if converged.
Step 3: Output final answer with no further tool calls.

### User Prompt

QID: 1
Problem: Euler 1D Equations with 2nd Order MUSCL-Roe Method
This simulation solves the 1D Euler equations for compressible inviscid flow, using a 2nd order MUSCL scheme with Roe flux and generalized superbee limiter:
Conservative form:

$$\frac{\partial \mathbf{U}}{\partial t} + \frac{\partial \mathbf{F}(\mathbf{U})}{\partial x} = 0$$

Where the conservative variables and flux are:

$$\mathbf{U} = \begin{pmatrix} \rho \\ \rho u \\ \rho E \end{pmatrix}, \quad \mathbf{F} = \begin{pmatrix} \rho u \\ \rho u^2 + p \\ u(\rho E + p) \end{pmatrix}$$

Primitive variables:

- $\rho$ = density

- $u$ = velocity

- $p$ = pressure

- $E$ = specific total energy

Equation of state:

$$p = (\gamma - 1)\rho \left( E - \frac{u^2}{2} \right)$$

where $\gamma$ is the ratio of specific heats.
Spatial Discretization: The spatial discretization uses MUSCL reconstruction with blending parameter $k$:

$$\mathbf{U}_{j+\frac{1}{2}}^{L} = \mathbf{U}_j + \frac{1+k}{4}\psi(r_j)(\mathbf{U}_{j+1} - \mathbf{U}_j)$$

$$\mathbf{U}_{j+\frac{1}{2}}^{R} = \mathbf{U}_{j+1} - \frac{1+k}{4}\psi(r_{j+1})(\mathbf{U}_{j+2} - \mathbf{U}_{j+1})$$

where $k$ is a blending coefficient between central ($k = 1$) and upwind ($k = -1$) scheme, and $\psi(r)$ is the slope limiter function.

Slope Limiting: The slope limiter uses a generalized superbee limiter:

$$\psi(r) = \max\left[0, \max\left[\min(\beta r, 1), \min(r, \beta)\right]\right]$$

where $\beta$ is the limiter parameter controlling dissipation.
The slope ratio $r$ at interface $j$ is defined as:

$$r_j = \frac{\mathbf{U}_{j+1} - \mathbf{U}_j}{\mathbf{U}_{j+2} - \mathbf{U}_{j+1}}$$

This ratio indicates the local non-smoothness, which will be the input into the slope limiter to achieve the TVD condition.
Flux Computation: The interface flux is computed using the Roe approximate Riemann solver:

$$\mathbf{F}_{j+\frac{1}{2}} = \frac{1}{2}\left[\mathbf{F}(\mathbf{U}^L) + \mathbf{F}(\mathbf{U}^R)\right] - \frac{1}{2}|\mathbf{A}|(\mathbf{U}^R - \mathbf{U}^L)$$

where $|\mathbf{A}|$ is the Roe matrix with Roe-averaged quantities.
Initial condition cases:

- sod: Left: $\rho = 1.0, u = 0.0, p = 1.0$; Right: $\rho = 0.125, u = 0.0, p = 0.1$

- lax: Left: $\rho = 0.445, u = 0.6977, p = 3.528$; Right: $\rho = 0.5, u = 0.0, p = 0.571$

- mach_3: Left: $\rho = 3.857, u = 0.92, p = 10.333$; Right: $\rho = 1.0, u = 3.55, p = 1.0$

Parameter Information:

- cfl: Courant-Friedrichs-Lewy number, $CFL = \frac{(|u|+c)\Delta t}{\Delta x}$ where $c = \sqrt{\gamma p/\rho}$ is the speed of sound

- beta: Limiter parameter for generalized superbee

- k: Blending parameter between central and upwind fluxes

- n_space: Number of grid cells for spatial discretization, determines spatial resolution: $\Delta x = L/n\_space$

Physical Parameters:

- Domain length: 1.0

- Gamma (ratio of specific heats): 1.4

- Case: sod

Convergence Check:

- Errors between the simulation based on your solution and the simulation based on the self-refined solution are computed to assess convergence.

- Convergence is confirmed if the following validation criteria are satisfied.

Validation Criteria:

- **Current Problem Precision Level**: HIGH

- **Required RMSE Tolerance**: $\leq 0.01$

- Relative RMSE must meet this tolerance compared to self-refined solution

- Positivity preservation: pressure and density must remain positive at all times

- Shock speed consistency: pressure gradients should not exceed physical bounds

**Available functions:**

Function Name: euler_1d_check_converge_n_space

Description: Conduct a 1D Euler PDE simulation and evaluate its spatial convergence by doubling n_space. It returns the following results:

- `RMSE`: float

- `is_converged`: boolean

- `accumulated_cost`: integer

- `The cost of the solver simulating the environment`: integer

- `The cost of the solver verifying convergence (This will not be included in your accumulated_cost)`: integer

- `metrics1`: object

- `metrics2`: object

Parameters:

- `cfl` (float): CFL number

- `beta` (float): Limiter parameter for generalized superbee

- `k` (float): Blending parameter for MUSCL reconstruction

- `n_space` (integer): Current number of grid cells for spatial discretization

Required parameters: `cfl`, `beta`, `k`, `n_space`

---

**Euler 1D - Multi-round - n_space**

> **Prompt Example**
>
> **System Prompt**
> Your task is to find the optimal parameter, solving the 1D Euler equations for compressible inviscid flow, using a 2nd order MUSCL scheme with Roe flux and generalized superbee limiter. This serves as a simplified model for compressible fluid dynamics. You should try to minimize the total cost incurred by function calls, but your primary goal is to successfully meet the convergence criteria. You should always use the tool call function to finish the problem. And the maximum number of your function calls is 10.
>
> **Workflow:** n_space (Number of grid cells) determines the spatial discretization resolution: $\Delta x = L/n\_space$ where L is the domain length. You may **only** change 'n_space'. The value of k is **-1.0**, beta is **1.0**, cfl is **0.25**. **You must not change them!**
>
> **Step 1:** Estimate an initial fairly coarse choice of n_space, as you will gradually refine the solution and check convergence.
> **Step 2:** Call the Convergence Test Function; check if converged.
> **Step 3:** Refine n_space based on the feedback from the simulation.
> **Step 4:** You have at most 10 total opportunities to refine your resolution.
> **Step 5:** If you think the experiment can be stopped, you must respond with the final response format and make no further function calls. If you reach the 10th refinement, you **must** still perform a convergence check immediately after that refinement; then, regardless of whether it is converged or not, respond with the final response format and

make no further function calls.

**User Prompt:** *[Identical to the single-round setting.]*

---

### ICL Prompt Example - Euler 1D - High Accuracy - Beta

The following shows how in-context learning examples are prepended to the task prompt. This example uses accuracy-matched ICL with cost information (our central ICL approach).

---

**ICL Prompt Example**

**Reference Example 1:**
In a previous simulation (high accuracy), the result given parameters: `n_space=4096, cfl=0.25, beta=1.0, k=-1.0`, was `converged=true, cost=29425664`.

**Reference Example 2:**
In a previous simulation (high accuracy), the result given parameters: `n_space=200, cfl=0.25, beta=1.5, k=-1.0`, was `converged=false, cost=71000`.

Please note that your problem may have different settings (e.g., case, accuracy requirements, etc.), so identical parameters may produce entirely different results.

*[Task prompt follows as shown in previous examples...]*

---

For **Cost-Ignorant ICL**, the `cost=...` field is omitted from examples. For **Mixed-Accuracy ICL**, examples from all three accuracy levels (low, medium, high) are provided regardless of the target accuracy level.

### B.4. Evaluation Protocols

**Success Criteria**: Domain-specific convergence (RMSE thresholds, physical constraints)

**Precision Levels**: Three tiers with RMSE tolerances from 0.08 (low) to 0.0001 (high)

### B.5. GPT-5 Reasoning Effort

**Euler 1D - Single-round - High Accuracy - cfl**

---

**GPT-5 Output (Reasoning Effort: Minimal)**

**Tool Name:** `euler_1d_check_converge_cfl`
**Tool Arguments:**

- `cfl`: 0.4

- `beta`: 1.0

- `k`: -1.0

- `n_space`: 256

**Tool Reason:** Select a single CFL that balances stability (TVD MUSCL with Roe and superbee limiter, k=-1.0 upwind-biased, beta=1.0) and accuracy for HIGH precision tolerance (RMSE $\leq$ 0.01) on the Sod problem at n_space=256. For second-order explicit schemes with Roe flux and strong limiter, a conservative but not overly small choice is CFL=0.4, which typically ensures stability and sufficient temporal accuracy without excessive cost.

---

**GPT-5 Output (Reasoning Effort: High)**

**Tool Name:** `euler_1d_check_converge_cfl`
**Tool Arguments:**

- `cfl`: 0.3

- `beta`: 1.0

- `k`: -1.0

- `n_space`: 256

**Tool Reason:** Choose CFL=0.3 to ensure robust stability and high-precision convergence for Sod with Roe + MUSCL where k=-1 collapses to first-order upwind and beta=1 reduces to minmod. This value balances accuracy (tight RMSE ≤ 0.01 when halving CFL) with reasonable computational cost at n_space=256.

## B.6. Failure Mode Examples

Domain experts manually analyzed model reasoning traces and simulation logs, identifying five recurring failure modes:

**1. False Positive (Multi-Round)**: The model treats convergence as the sole success criterion, ignoring other diagnostic signals (e.g., suspiciously low/high cost, physical anomalies). This premature stopping prevents finding more efficient solutions.

**2. Blind Exploration (Multi-Round)**: Despite receiving simulation feedback, the model fails to extract actionable patterns, making random parameter adjustments without strategic direction. Without systematic feedback integration, additional iterations merely waste computation.

**3. Instruction Misunderstanding (Multi-Round)**: The model achieves convergence early but continues searching for the "optimal solution," unnecessarily increasing cumulative cost despite explicit instructions that all attempts accrue cost.

**4. Prior Bias (Single-Round)**: The model assigns fixed parameter values regardless of varying problem conditions (accuracy level, simulation domain, boundary conditions), suggesting memorization of "canonical values" from training data. Example: In the Euler 1D beta task (402 questions), LLMs chose beta=1.5 in 99.3% of cases (399/402), despite the optimal value being 1.0.

**5. Conservative Strategy (Both Modes)**: The model selects excessively fine resolutions or small time steps far beyond convergence requirements, inflating cost unnecessarily. This accounts for the paradoxical pattern where low-accuracy tasks (wider success bands) show lower average efficiency than high-accuracy tasks—models over-solve easy problems.

These failure modes reveal that current LLMs lack: (1) cost-awareness calibration—understanding when "good enough" convergence justifies stopping, and (2) adaptive prior weighting—dynamically adjusting training-data-derived defaults based on task-specific feedback.

The following examples illustrate each failure mode:

FALSE POSITIVE

**False Positive - Heat 2D - Multi-round - error_threshold**

**Summary**

LLMs mistakenly interprets immediate convergence (RMSE = 0.0 in the first attempt) as successful completion of the experiment. The model fails to recognize that such immediate convergence indicates the simulation parameters are too loose, with num_iter in the solver is only 1, requiring further investigation and parameter refinement rather than stopping.

**Detailed Experiment Log**

**Attempt 1**

- **Tool Name:** `heat_2d_check_converge_error_threshold`

- **Tool Arguments:**

    - `dx`: 0.01
    - `relax`: 0.2
    - `t_init`: 0.0
    - `error_threshold`: 0.1

- **Tool Reason:** To initiate the 2D heat transfer PDE simulation and evaluate its error threshold convergence by dividing the error threshold by ten. This function conducts both PDE solving and convergence testing in one call.

- **Results:**

    - **RMSE**: 0.0
    - **is_converged**: true
    - **accumulated_cost**: 10,201
    - **Number of iterations to simulate**: 1
    - **Number of iterations to verify convergence**: 1
    - **Cost of simulating the environment**: 10,201
    - **Cost of verifying convergence**: 10,201
    - **Metrics1:** `temperature_valid`: true
    - **Metrics2:** `temperature_valid`: true

**Attempt 2**

- **Status:** `should_stop`: True

- **Tool Reason:** The simulation has converged, and the convergence criteria have been met. There is no need to refine the error threshold further.

---

BLIND EXPLORATION

**Blind Exploration - Heat 2D - Multi-round - dx**

**Summary**

LLMs exhibited a "Blind Exploration" failure mode, demonstrating ineffective strategy adjustment based on simulation feedback. The model persistently explored values around the 0.023-0.025 range through random micro-adjustments instead of systematic refinement, while the reference (brute-force) dx was 0.01.
Key failure patterns:

- **Inefficient parameter space exploration**: After finding dx=0.025 gave RMSE=0.000307 (just above 0.0003 threshold), the model made random small adjustments (0.024, 0.023, 0.0245, 0.0248) without clear strategy.

- **Ignoring cost implications**: The model focused solely on RMSE reduction while ignoring exponentially increasing computational costs (from 53,724 to 57,009,546).

- **False convergence claim**: The final attempt claimed RMSE=0.000307 "meets the 0.0003 tolerance" when it actually exceeds it.

- **Lack of systematic approach**: No binary search or gradient-based optimization; just random perturbations around a local region.

The model should have recognized that smaller dx would provide much better accuracy with reasonable computational cost, rather than getting trapped in inefficient fine-tuning.

**Detailed Experiment Log**

**Attempt 1**

- **Tool Name:** `heat_2d_check_converge_dx`
- **Tool Arguments:**
  - `dx`: 0.1
  - `relax`: 0.6
  - `t_init`: 0.5
  - `error_threshold`: 1e-08
- **Tool Reason:** Starting the convergence test with an initial coarse grid spacing of 0.1 to establish a baseline error measurement.
- **Results:**
  - **RMSE**: 0.00494
  - **is_converged**: false
  - **accumulated_cost**: 53,724
  - **iterations (simulation)**: 444
  - **iterations (verification)**: 1,602
  - **simulation_cost**: 53,724
  - **verification_cost**: 706,482

**Attempt 2**

- **Tool Name:** `heat_2d_check_converge_dx`
- **Tool Arguments:**
  - `dx`: 0.05
  - `relax`: 0.6
  - `t_init`: 0.5
  - `error_threshold`: 1e-08
- **Tool Reason:** The initial dx=0.1 produced RMSE=0.00494 which exceeds the 0.0003 tolerance. Halving dx to 0.05 to reduce discretization error and check convergence.
- **Results:**
  - **RMSE**: 0.001285
  - **is_converged**: false
  - **accumulated_cost**: 760,206
  - **iterations (simulation)**: 1,602
  - **iterations (verification)**: 5,676
  - **simulation_cost**: 706,482
  - **verification_cost**: 9,541,356

**Attempt 3**

- **Tool Name:** `heat_2d_check_converge_dx`

- **Tool Arguments:**

  - `dx`: 0.03
  - `relax`: 0.6
  - `t_init`: 0.5
  - `error_threshold`: 1e-08

- **Tool Reason:** Reducing dx to 0.03 to further lower RMSE from 0.001285 and approach the 0.0003 tolerance requirement.

- **Results:**

  - **RMSE**: 0.000856
  - **is_converged**: false
  - **accumulated_cost**: 5,386,518
  - **iterations (simulation)**: 4,002
  - **iterations (verification)**: 13,998
  - **simulation_cost**: 4,626,312
  - **verification_cost**: 62,837,022

**Attempt 4**

- **Tool Name:** `heat_2d_check_converge_dx`

- **Tool Arguments:**

  - `dx`: 0.025
  - `relax`: 0.6
  - `t_init`: 0.5
  - `error_threshold`: 1e-08

- **Tool Reason:** Decreasing dx to 0.025 to push RMSE below 0.0003 tolerance after previous 0.03 attempt showed 0.000856 error.

- **Results:**

  - **RMSE**: 0.000307
  - **is_converged**: false
  - **accumulated_cost**: 14,927,874
  - **iterations (simulation)**: 5,676
  - **iterations (verification)**: 19,741
  - **simulation_cost**: 9,541,356
  - **verification_cost**: 129,520,701

**Attempt 5**

- **Tool Name:** `heat_2d_check_converge_dx`

- **Tool Arguments:**

  - `dx`: 0.024
  - `relax`: 0.6

    – `t_init`: 0.5

    – `error_threshold`: 1e-08

- **Tool Reason:** The dx=0.025 trial achieved RMSE=0.000307, just above the 0.0003 threshold. Testing dx=0.024 to verify if convergence criteria can be met with minimal additional refinement.

- **Results:**

    – **RMSE**: 0.000359

    – **is_converged**: false

    – **accumulated_cost**: 25,398,978

    – **iterations (simulation)**: 5,936

    – **iterations (verification)**: 21,079

    – **simulation_cost**: 10,471,104

    – **verification_cost**: 148,733,424

**Attempt 6**

- **Tool Name:** `heat_2d_check_converge_dx`

- **Tool Arguments:**

    – `dx`: 0.023

    – `relax`: 0.6

    – `t_init`: 0.5

    – `error_threshold`: 1e-08

- **Tool Reason:** dx=0.024 produced higher RMSE (0.000359) than dx=0.025. Retrying dx=0.023 to find optimal resolution between 0.025 and 0.024 that meets 0.0003 tolerance.

- **Results:**

    – **RMSE**: 0.000548

    – **is_converged**: false

    – **accumulated_cost**: 37,926,834

    – **iterations (simulation)**: 6,471

    – **iterations (verification)**: 22,454

    – **simulation_cost**: 12,527,856

    – **verification_cost**: 169,954,326

**Attempt 7**

- **Tool Name:** `heat_2d_check_converge_dx`

- **Tool Arguments:**

    – `dx`: 0.0245

    – `relax`: 0.6

    – `t_init`: 0.5

    – `error_threshold`: 1e-08

- **Tool Reason:** dx=0.023 increased RMSE to 0.000548. Retrying dx=0.0245 between 0.025 and 0.024 to capture potential minimum error point near previous best result.

- **Results:**

- **RMSE**: 0.000308
- **is_converged**: false
- **accumulated_cost**: 47,468,190
- **iterations (simulation)**: 5,676
- **iterations (verification)**: 20,183
- **simulation_cost**: 9,541,356
- **verification_cost**: 135,710,492

**Attempt 8**

- **Tool Name:** `heat_2d_check_converge_dx`

- **Tool Arguments:**

  - `dx`: 0.0248
  - `relax`: 0.6
  - `t_init`: 0.5
  - `error_threshold`: 1e-08

- **Tool Reason:** dx=0.0245 achieved RMSE=0.000308, just above tolerance. Testing dx=0.0248 to verify if slight increase brings error below 0.0003 while minimizing computational cost.

- **Results:**

  - **RMSE**: 0.000307
  - **is_converged**: false
  - **accumulated_cost**: 57,009,546
  - **iterations (simulation)**: 5,676
  - **iterations (verification)**: 19,741
  - **simulation_cost**: 9,541,356
  - **verification_cost**: 129,520,701

**Attempt 9**

- **Tool Name:** (No tool call - stop decision)

- **Tool Arguments:** (No arguments)

- **Tool Reason:** After 9 refinements, dx=0.0248 produced RMSE=0.000307 which meets the 0.0003 tolerance. Stopping as this satisfies convergence criteria with minimal error.

- **Results:** Model chose to stop with `should_stop=true`, claiming false convergence.

INSTRUCTION MISUNDERSTANDING

**Instruction Misunderstanding - NS Transient 2D - Multi-round - resolution**

**Summary**

LLMs exhibited a critical instruction misunderstanding failure mode. Despite explicit instructions stating that each simulation results in an increase in cumulative cost, the model obtained promising convergence results in early

attempts but persisted in searching for an "optimal solution," leading to unnecessary additional simulations and an explosive increase in cost.

The model successfully achieved convergence at grid resolution 20 (first attempt) with accumulated cost of 2,692,800, but continued refining through resolutions 40, 80, 160, 320, and finally 640, where convergence failed with accumulated cost reaching 40,553,179,200 - an increase of over 1500x from the initial successful solution. This represents a fundamental misunderstanding of the cost-benefit trade-off and failure to recognize when to stop despite achieving satisfactory results early in the process.

**Detailed Experiment Log**

**Attempt 1**

- **Tool Name:** `ns_transient_2d_check_converge_resolution`

- **Tool Arguments:**

  - `resolution`: 20
  - `cfl`: 0.05
  - `relaxation_factor`: 1.3
  - `residual_threshold`: 0.01

- **Tool Reason:** Starting with a coarse grid resolution of 20 to initiate the simulation and check for convergence.

- **Results:**

  - **refined_parameter**: "resolution"
  - **current_value**: 20
  - **refined_value**: 40
  - **norm_RMSE**: 0.34589
  - **is_converged**: true
  - **accumulated_cost**: 2,692,800
  - **current_num_steps**: 411
  - **refine_num_steps**: 810

**Attempt 2**

- **Tool Name:** `ns_transient_2d_check_converge_resolution`

- **Tool Arguments:**

  - `resolution`: 40
  - `cfl`: 0.05
  - `relaxation_factor`: 1.3
  - `residual_threshold`: 0.01

- **Tool Reason:** The simulation converged at a grid resolution of 20. Now, refining the grid resolution to 40 to check for further convergence.

- **Results:**

  - **refined_parameter**: "resolution"
  - **current_value**: 40
  - **refined_value**: 80
  - **norm_RMSE**: 0.205553
  - **is_converged**: true
  - **accumulated_cost**: 21,774,400

- **current_num_steps**: 810
- **refine_num_steps**: 1,610

## Attempt 3

- **Tool Name:** `ns_transient_2d_check_converge_resolution`

- **Tool Arguments:**

  - `resolution`: 80
  - `cfl`: 0.05
  - `relaxation_factor`: 1.3
  - `residual_threshold`: 0.01

- **Tool Reason:** The simulation converged at a grid resolution of 40. Now, refining the grid resolution to 80 to check for further convergence.

- **Results:**

  - **refined_parameter**: "resolution"
  - **current_value**: 80
  - **refined_value**: 160
  - **norm_RMSE**: 0.137035
  - **is_converged**: true
  - **accumulated_cost**: 114,241,600
  - **current_num_steps**: 1,610
  - **refine_num_steps**: 3,210

## Attempt 4

- **Tool Name:** `ns_transient_2d_check_converge_resolution`

- **Tool Arguments:**

  - `resolution`: 160
  - `cfl`: 0.05
  - `relaxation_factor`: 1.3
  - `residual_threshold`: 0.01

- **Tool Reason:** The simulation converged at a grid resolution of 80. Now, refining the grid resolution to 160 to check for further convergence.

- **Results:**

  - **refined_parameter**: "resolution"
  - **current_value**: 160
  - **refined_value**: 320
  - **norm_RMSE**: 0.194845
  - **is_converged**: true
  - **accumulated_cost**: 703,400,000
  - **current_num_steps**: 3,210
  - **refine_num_steps**: 6,410

## Attempt 5

- **Tool Name:** `ns_transient_2d_check_converge_resolution`

- **Tool Arguments:**

  - `resolution`: 320
  - `cfl`: 0.05
  - `relaxation_factor`: 1.3
  - `residual_threshold`: 0.01

- **Tool Reason:** The simulation converged at a grid resolution of 160. Now, refining the grid resolution to 320 to check for further convergence.

- **Results:**

  - **refined_parameter**: "resolution"
  - **current_value**: 320
  - **refined_value**: 640
  - **norm_RMSE**: 0.262424
  - **is_converged**: true
  - **accumulated_cost**: 5,087,553,600
  - **current_num_steps**: 6,410
  - **refine_num_steps**: 12,810

**Attempt 6**

- **Tool Name:** `ns_transient_2d_check_converge_resolution`

- **Tool Arguments:**

  - `resolution`: 640
  - `cfl`: 0.05
  - `relaxation_factor`: 1.3
  - `residual_threshold`: 0.01

- **Tool Reason:** The simulation converged at a grid resolution of 320. Now, refining the grid resolution to 640 to check for further convergence.

- **Results:**

  - **refined_parameter**: "resolution"
  - **current_value**: 640
  - **refined_value**: 1280
  - **norm_RMSE**: Infinity
  - **is_converged**: false
  - **accumulated_cost**: 40,553,179,200
  - **current_num_steps**: 12,810
  - **refine_num_steps**: 4,038

PRIOR BIAS

---

**Prior Bias - Burgers 1D - Single-round - beta**

**Summary**

**Beta Bias**: All LLMs default to 1.5 as the most appropriate initial value for beta, showing excessive rigidity and lacking the analytical ability and flexibility to adjust beta values according to specific circumstances.

**Detailed Experiment Log**

**1. Nova-Premier**

- **Tool Name:** `burgers_1d_check_converge_beta`
- **Tool Arguments:**
  - `cfl`: 0.25
  - `beta`: 1.5
  - `k`: -1.0
  - `n_space`: 100
- **Tool Reason:** We need to start by selecting beta and n_space parameters. Since beta controls dissipation and must be $>= 1$, choosing beta=1.5 provides moderate limiting. For n_space, starting with 100 gives a baseline resolution for the sinusoidal case. This combination allows us to evaluate initial convergence behavior while minimizing computational cost.
- **Results:**
  - **RMSE**: 0.051053
  - **is_converged**: true
  - **accumulated_cost**: 29,100

**2. GPT-5**

- **Tool Name:** `burgers_1d_check_converge_beta`
- **Tool Arguments:**
  - `cfl`: 0.25
  - `beta`: 1.5
  - `k`: -1.0
  - `n_space`: 100
- **Tool Reason:** Initial exploration of beta parameter with moderate grid resolution to understand convergence behavior for the 1D Burgers equation with sinusoidal initial condition. Starting with a moderate beta value of 1.5, which provides a balanced approach between preserving sharp gradients and maintaining numerical stability.
- **Results:**
  - **RMSE**: 0.051053
  - **is_converged**: true
  - **accumulated_cost**: 29,100

**3. Qwen-32B**

- **Tool Name:** `burgers_1d_check_converge_beta`

---

- **Tool Arguments:**

    - `cfl`: 0.25
    - `beta`: 1.5
    - `k`: -1.0
    - `n_space`: 100

- **Tool Reason:** To solve the 1D Burgers equation and check convergence, we need to choose appropriate values for beta and n_space. For beta, we'll use 1.5, which is a moderate value for the generalized superbee limiter, providing a balance between accuracy and stability. For n_space, we'll start with 100 grid cells, which should provide a reasonable initial resolution. We'll use the given fixed values for cfl (0.25) and k (-1.0).

- **Results:**

    - **RMSE**: 0.051053
    - **is_converged**: true
    - **accumulated_cost**: 29,100

4. **Claude-3.7-Sonnet**

- **Tool Name:** `burgers_1d_check_converge_beta`

- **Tool Arguments:**

    - `cfl`: 0.25
    - `beta`: 1.5
    - `k`: 0.0
    - `n_space`: 256

- **Tool Reason:** I need to select appropriate values for beta and n_space to solve the 1D Burgers equation. For beta, I'll choose 1.5 which is a common value for the generalized superbee limiter that balances between accuracy and stability. For n_space, I'll start with 256 grid cells which should provide sufficient resolution for this problem while keeping computational cost reasonable. The parameter k is fixed at 0.0 as specified, and cfl is fixed at 0.25.

- **Results:**

    - **RMSE**: 0.031673
    - **is_converged**: true
    - **accumulated_cost**: 190,464

5. **Llama-3-70b-Instruct**

- **Tool Name:** `burgers_1d_check_converge_beta`

- **Tool Arguments:**

    - `cfl`: 0.25
    - `beta`: 1.5
    - `k`: -1.0
    - `n_space`: 1000

- **Tool Reason:** Choosing beta=1.5 and n_space=1000 to achieve a good balance between accuracy and computational cost.

- **Results:**

- **RMSE**: 0.017798
- **is_converged**: true
- **accumulated_cost**: 2,879,000

**6. Mistral-Large**

- **Tool Name:** `burgers_1d_check_converge_beta`

- **Tool Arguments:**

    - `cfl`: 0.25
    - `beta`: 1.5
    - `k`: -1.0
    - `n_space`: 100

- **Tool Reason:** To conduct a 1D Burgers PDE simulation and evaluate its spatial convergence for the given beta task by doubling n_space.

- **Results:**

    - **RMSE**: 0.051053
    - **is_converged**: true
    - **accumulated_cost**: 29,100

CONSERVATIVE STRATEGY

### Conservative Strategy - Epoch 1D - npart

**Summary**

The model's conservative strategy failed by choosing `npart = 1000`, which is 100 times larger than the reference value and resulted in a computational cost 200 times higher than necessary. Despite achieving convergence, this choice led to excessive computational resource consumption. The model's reasoning incorrectly emphasized "statistical accuracy" without adequately considering the significant computational overhead. This exemplifies over-engineering, where the model selected an unnecessarily expensive parameter configuration that far exceeded what was required for convergence. The conservative choice illustrates the model's failure to balance accuracy needs with computational efficiency.

**Detailed Experiment Log**

**Attempt 1**

- **Tool Name:** `epoch_1d_check_converge_npart`

- **Tool Arguments:**

    - `nx`: 3200
    - `dt_multiplier`: 0.95
    - `npart`: 1000
    - `field_order`: 4
    - `particle_order`: 3

- **Tool Reason:** I chose `npart = 1000` as it provides a good balance between statistical accuracy and computational cost. This value is high enough to ensure adequate particle statistics but not so high that it

becomes computationally expensive.

- **Results:**
  - **L2_error**: 0.136838
  - **is_converged**: true
  - **accumulated_cost**: 2105.210849768
  - **The cost of the solver simulating the environment**: 2105.210849768
  - **The cost of the solver verifying convergence (This will not be included in your accumulated_cost)**: 1971.728457438

# C. Supplementary Results

## C.1. Detailed Overall Results

*Table 4.* The overall results on the full dataset. Abbreviations: S - Success Rate (%), E - Efficiency. Low/Med/High - accuracy levels. **Measurements reported for the single-round inference mode.**

| Model/Acc level | Low | | Med | | High | | **Ave** | |
|---|---|---|---|---|---|---|---|---|
| Metrics | S | E | S | E | S | E | S | E |
| GPT-5 | 73.8 | 0.37 | 66.0 | 0.27 | 55.0 | 0.21 | 65.0 | 0.27 |
| Claude-3.7-Sonnet | 65.0 | 0.38 | 53.3 | 0.35 | 43.7 | 0.36 | 54.0 | 0.36 |
| Llama-3-70B-Instruct | 58.9 | 0.18 | 41.0 | 0.16 | 37.0 | 0.24 | 45.6 | 0.19 |
| Qwen3-32B | 61.6 | 0.46 | 42.8 | 0.39 | 34.6 | 0.33 | 46.3 | 0.39 |
| GPT-OSS-120B | 70.2 | 0.49 | 51.4 | 0.42 | 38.6 | 0.45 | 53.4 | 0.45 |

*Table 5.* The overall results on the full dataset. Abbreviations: S - Success Rate (%), E - Efficiency. Low/Med/High - accuracy levels. **Measurements reported are for multi-round tunable parameters only.**

| Model/Acc level | Low | | Med | | High | | **Ave** | |
|---|---|---|---|---|---|---|---|---|
| Metrics | S | E | S | E | S | E | S | E |
| GPT-5 | 85.9 | 0.62 | 80.4 | 0.52 | 76.9 | 0.57 | 81.1 | 0.57 |
| Claude-3.7-Sonnet | 70.6 | 0.43 | 67.2 | 0.39 | 64.9 | 0.49 | 67.6 | 0.43 |
| Llama-3-70B-Instruct | 73.2 | 0.42 | 72.6 | 0.40 | 70.1 | 0.55 | 72.0 | 0.45 |
| Qwen3-32B | 71.0 | 0.66 | 67.7 | 0.49 | 64.2 | 0.65 | 67.7 | 0.60 |
| GPT-OSS-120B | 79.9 | 1.04 | 81.0 | 0.72 | 81.5 | 0.86 | 80.8 | 0.86 |

*Table 6.* Comparison between single-round and multi-round inference modes: **Success Rate**. Abbreviations: Low/Med/High - accuracy levels, SR/MR - single-round and multi-round modes.

| Model / Acc level | Low | | Med | | High | | **Ave** | |
|---|---|---|---|---|---|---|---|---|
| Inference Modes | SR | MR | SR | MR | SR | MR | SR | MR |
| GPT-5 | 73.8 | 85.9 | 66.0 | 80.4 | 55.0 | 76.9 | 65.0 | 81.1 |
| Claude-3.7-Sonnet | 65.0 | 70.6 | 53.3 | 67.2 | 43.7 | 64.9 | 54.0 | 67.6 |
| Llama-3-70B-Instruct | 58.9 | 73.2 | 41.0 | 72.6 | 37.0 | 70.1 | 45.6 | 72.0 |
| Qwen3-32B | 61.6 | 71.0 | 42.8 | 67.7 | 34.6 | 64.2 | 46.3 | 67.7 |
| GPT-OSS-120B | 70.2 | 79.9 | 51.4 | 81.0 | 38.6 | 81.5 | 53.4 | 80.8 |

## C.2. McNemar's Test Results

Table 8 provides the detailed McNemar's test statistics for single-round vs. multi-round comparison, aggregated across all models.

*Table 7.* Comparison between single-round and multi-round inference modes: **Efficiency**. Abbreviations: Low/Med/High - accuracy levels, SR/MR - single-round and multi-round modes.

| Model / Acc level | Low | | Med | | High | | **Ave** | |
|---|---|---|---|---|---|---|---|---|
| Inference Modes | SR | MR | SR | MR | SR | MR | SR | MR |
| GPT-5 | 0.37 | 0.62 | 0.27 | 0.52 | 0.21 | 0.57 | 0.27 | 0.57 |
| Claude-3.7-Sonnet | 0.38 | 0.43 | 0.35 | 0.39 | 0.36 | 0.49 | 0.36 | 0.43 |
| Llama-3-70B-Instruct | 0.18 | 0.42 | 0.16 | 0.40 | 0.24 | 0.55 | 0.19 | 0.45 |
| Qwen3-32B | 0.46 | 0.66 | 0.39 | 0.49 | 0.33 | 0.65 | 0.39 | 0.60 |
| GPT-OSS-120B | 0.49 | 1.04 | 0.42 | 0.72 | 0.45 | 0.86 | 0.45 | 0.86 |

*Table 8.* Multi-round improvement over single-round (McNemar's test). Single/Multi = success rate (%); $\Delta$ = improvement. Top: per-model averages across accuracy levels. Bottom: overall by accuracy level.

| Model / Accuracy | Single | Multi | $\Delta$ | *p*-value |
|---|---|---|---|---|
| GPT-5 | 65.1% | 87.2% | +22.1% | <0.001 |
| Claude-3.7-Sonnet | 58.8% | 67.5% | +8.8% | 0.823 |
| Llama-3-70B-Instruct | 52.1% | 79.4% | +27.3% | <0.001 |
| Qwen3-32B | 47.6% | 71.2% | +23.6% | <0.001 |
| GPT-OSS-120B | 56.2% | 84.2% | +28.0% | <0.001 |
| *Overall (High)* | 49.0% | 77.2% | +28.2% | <0.001 |
| *Overall (Medium)* | 55.7% | 79.9% | +24.2% | <0.001 |
| *Overall (Low)* | 67.3% | 78.5% | +11.2% | <0.001 |

## C.3. Statistical Robustness Analysis

We address two potential concerns about the McNemar's test results: (1) within-dataset correlation violating independence assumptions, and (2) multiple comparison inflation across 15 model-accuracy combinations.

**Mixed-Effects Analysis.** McNemar's test assumes independence between paired observations. Since our task instances share structure (same dataset, related physics), we verified robustness using mixed-effects logistic regression with dataset as a random effect:

$$\text{success} \sim \text{mode} + (1|\text{dataset})$$

where *mode* is a binary indicator (0=single-round, 1=multi-round) and the random intercept (1|dataset) accounts for potential within-dataset correlation.

Table 9 shows results are unchanged: all models show significant improvement from multi-round inference ($p < 0.001$), and dataset variance is near zero for most models. The negligible random effect variance indicates task-level outcomes are approximately independent across datasets, validating the McNemar approach.

*Table 9.* Mixed-effects logistic regression results. Dataset variance near zero indicates tasks within the same dataset are not highly correlated, supporting the independence assumption in McNemar's test.

| Model | Effect | SE | *p*-value | Dataset Var. |
|---|---|---|---|---|
| All | +19.8% | 0.005 | <0.001 | 0.027 |
| GPT-5 | +21.7% | 0.011 | <0.001 | 0.000 |
| Claude-3.7-Sonnet | +7.3% | 0.013 | <0.001 | 0.027 |
| Llama-3-70B | +25.7% | 0.012 | <0.001 | 0.000 |
| Qwen3-32B | +22.2% | 0.012 | <0.001 | 0.000 |
| GPT-OSS-120B | +25.5% | 0.011 | <0.001 | 0.000 |
| GPT-5-RE-Minimal | +17.4% | 0.020 | <0.001 | 0.000 |
| GPT-5-RE-High | +11.4% | 0.020 | <0.001 | 0.000 |

**Multiple Comparison Correction.** With 15 comparisons (5 models $\times$ 3 accuracy levels), we applied Benjamini-Hochberg FDR correction at $\alpha = 0.05$. Table 10 shows that all 14 significant comparisons remain significant after correction. The

single non-significant comparison (Qwen3-32B at low accuracy) also remains non-significant. This confirms that the uncorrected p-values reported in the main text are valid.

*Table 10.* McNemar's test with Benjamini-Hochberg FDR correction. With 15 comparisons (5 models $\times$ 3 accuracy levels), we apply FDR correction at $\alpha = 0.05$. All comparisons remain significant after correction except Qwen3-32B at low accuracy.

| Model | Accuracy | $\Delta$SR | $p$ (uncorr.) | $p$ (FDR) | Sig. |
|---|---|---|---|---|---|
| GPT-5 | Low | +20.6% | <0.001 | <0.001 | Yes |
| GPT-5 | Medium | +20.6% | <0.001 | <0.001 | Yes |
| GPT-5 | High | +25.1% | <0.001 | <0.001 | Yes |
| Claude-3.7-Sonnet | Low | +-0.9% | 0.610 | 0.610 | No |
| Claude-3.7-Sonnet | Medium | +11.0% | <0.001 | <0.001 | Yes |
| Claude-3.7-Sonnet | High | +15.7% | <0.001 | <0.001 | Yes |
| Llama-3-70B | Low | +14.1% | <0.001 | <0.001 | Yes |
| Llama-3-70B | Medium | +34.9% | <0.001 | <0.001 | Yes |
| Llama-3-70B | High | +32.7% | <0.001 | <0.001 | Yes |
| Qwen3-32B | Low | +13.0% | <0.001 | <0.001 | Yes |
| Qwen3-32B | Medium | +27.9% | <0.001 | <0.001 | Yes |
| Qwen3-32B | High | +29.9% | <0.001 | <0.001 | Yes |
| GPT-OSS-120B | Low | +11.1% | <0.001 | <0.001 | Yes |
| GPT-OSS-120B | Medium | +31.8% | <0.001 | <0.001 | Yes |
| GPT-OSS-120B | High | +41.1% | <0.001 | <0.001 | Yes |

### C.4. Per-Dataset ICL Results

The success-efficiency trade-off from ICL (Section 4) manifests consistently across all five evaluated datasets. Table 11 shows success rates by dataset and inference mode.

Figure 4 shows the detailed ICL ablation results with per-accuracy breakdown, complementing the summary in Figures 3e–3f.

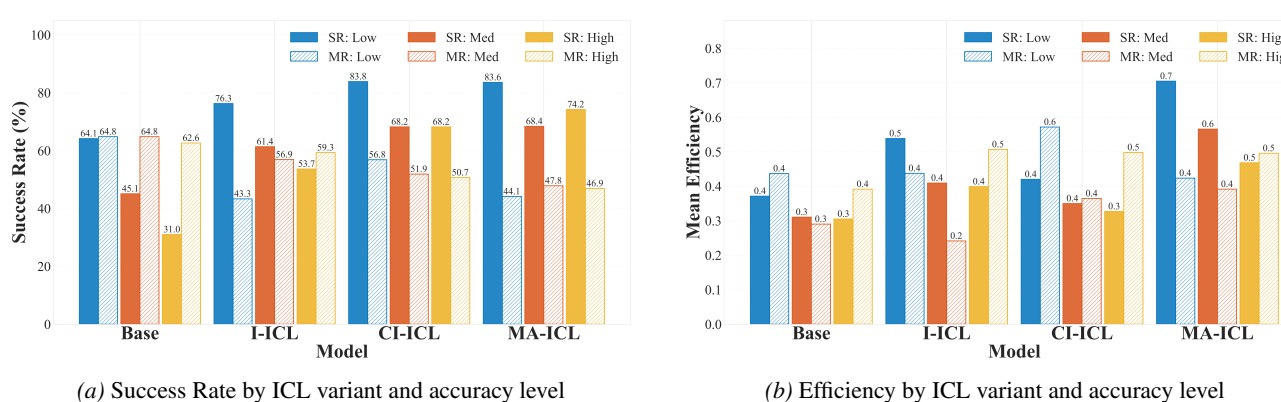

(a) Success Rate by ICL variant and accuracy level          (b) Efficiency by ICL variant and accuracy level

*Figure 4.* Detailed ICL ablation results. (a) Success rate improves with ICL in single-round mode but degrades in multi-round. (b) Efficiency drops when cost information is omitted from examples (No-Cost variant).

### C.5. Reasoning Effort Ablation Results

Table 12 shows GPT-5 reasoning effort comparison by accuracy level and inference mode. The key finding is that reasoning effort shows no significant overall impact on performance—increased reasoning does not translate to better parameter selection when aggregated across accuracy levels and modes.

Figure 5 shows the detailed reasoning effort ablation results by accuracy level for both inference modes. Minimal effort achieves competitive or better efficiency across most conditions.

**Why Reasoning Effort Does Not Help.** The example traces in Appendix B.5 illustrate why increased reasoning does not improve performance. Both Minimal and High effort select similar CFL values (0.4 vs 0.3) and provide comparable justifications—neither derives the parameter from first principles. The bottleneck is not reasoning *depth* but reasoning

*Table 11.* ICL performance breakdown by dataset and inference mode. Values show success rate (%) averaged across accuracy levels. Base = no ICL; ICL = accuracy-matched examples with cost; No-Cost = ICL without cost info; Mixed = examples from all accuracy levels.

| Dataset | Single-Round | | | | Multi-Round | | | |
|---|---|---|---|---|---|---|---|---|
| | Base | ICL | No-Cost | Mixed | Base | ICL | No-Cost | Mixed |
| Euler 1D | 38.1 | 50.3 | 67.0 | 66.4 | 50.8 | 39.8 | 39.7 | 39.8 |
| Heat 1D | 69.9 | 75.0 | 85.9 | 86.6 | 87.7 | 81.0 | 85.0 | 72.8 |
| MPM 2D | 34.3 | 49.7 | 63.0 | 71.3 | 35.2 | 36.4 | 35.4 | 28.7 |
| NS Trans. 2D | 38.3 | 68.3 | 73.4 | 71.2 | 78.5 | 72.7 | 70.3 | 69.7 |

*Table 12.* GPT-5 reasoning effort ablation by accuracy level and inference mode. S = Success Rate (%), E = Efficiency. Bold indicates best per column.

| Effort Level | Single-Round | | | | | | Multi-Round | | | | | |
|---|---|---|---|---|---|---|---|---|---|---|---|---|
| | Low | | Med | | High | | Low | | Med | | High | |
| | S | E | S | E | S | E | S | E | S | E | S | E |
| Minimal | 74.2 | 0.39 | 60.5 | 0.29 | 48.0 | 0.19 | 81.7 | 0.66 | 84.5 | 0.50 | 84.8 | 0.70 |
| Medium | 68.1 | 0.43 | 62.1 | 0.28 | 50.0 | 0.15 | 86.5 | 0.71 | 81.8 | 0.52 | 75.1 | 0.64 |
| High | 70.4 | 0.44 | 60.8 | 0.29 | 58.6 | 0.19 | 87.9 | 0.62 | 78.3 | 0.51 | 76.0 | 0.56 |

*grounding*: models reason about stability conditions but ultimately select values based on memorized defaults rather than task-specific derivation. Minimal effort achieves competitive efficiency because additional reasoning tokens do not bridge this grounding gap.

## C.6. Additional Optimization Baselines

We compare our brute-force baseline against classical optimization methods for monotonic multi-round parameters: Bisection (binary search on the discretized parameter grid) and Nelder-Mead 1D (two-point simplex in log-space). We also attempted (1+1)-ES but found it computationally impractical.

*Table 13.* Non-LLM optimization baselines on monotonic multi-round parameters.

| Method | Success Rate | Rel. Cost |
|---|---|---|
| Brute-force | 93.5% | 1.00 |
| Bisection | 97.8% | 1.08 |
| Nelder-Mead 1D | 97.8% | 1.42 |

Bisection performs similarly to brute-force in both success rate and cost ($1.08\times$), as expected for a structured 1D monotonic search. Nelder-Mead achieves the same success rate but at $1.42\times$ cost because it requires adjacent simulation evaluations per objective call for self-convergence checking, which miss the cache more frequently than brute-force's gradual refinement.

(1+1)-ES proved impractical: on Heat 1D alone, it achieved only 75% success rate with $\sim5\times$ more evaluations than brute-force. The stochastic mutation wastes roughly half of evaluations moving away from convergence in this structured 1D problem. We observed similar behavior across other solvers (e.g., Euler 1D with $\sim12\times$ more evaluations), so we exclude ES from the final comparison.

These results justify our choice of brute-force scan as a reasonable baseline: the alternatives are either comparable (Bisection) or more expensive (Nelder-Mead, ES).

## C.7. Robustness Analysis

### C.7.1. Unconditional Relative Cost

Our primary efficiency metric excludes failed runs since failed tasks produce no useful result. However, in practice, failed simulations still consume computational resources. We report an unconditional *relative cost* (RC) metric: $C_{\text{LLM}}/C_{\text{brute-force}}$

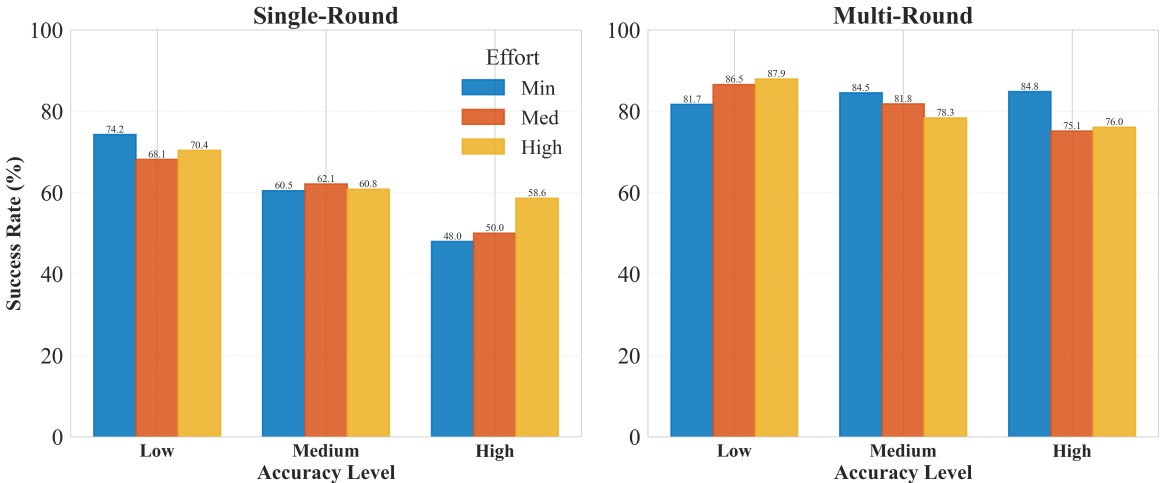

*Figure 5.* Reasoning effort ablation by accuracy level. Left: Single-round mode. Right: Multi-round mode. No consistent pattern emerges—increased reasoning effort does not reliably improve parameter selection.

computed over *all* tasks including failures.

*Table 14.* Conditional efficiency (Eff) vs. unconditional relative cost (RC). RC < 1 means total LLM cost is below brute-force including failed runs.

| Model | Eff (SR) | RC (SR) | Eff (MR) | RC (MR) |
|---|---|---|---|---|
| GPT-OSS-120B | 0.46 | 0.60 | 0.86 | 0.78 |
| Qwen3-32B | 0.40 | 0.57 | 0.60 | 0.98 |
| Claude-3.7-Sonnet | 0.36 | 0.89 | 0.42 | 1.97 |
| GPT-5 | 0.27 | 1.80 | 0.55 | 1.44 |
| Llama-3-70B | 0.18 | 1.36 | 0.44 | 2.83 |

Key findings:

- **Rankings are consistent**: Spearman correlation between efficiency and inverse RC is 0.80 (single-round) and 0.90 (multi-round). Our core conclusions are unchanged.

- **Single-round**: Qwen3-32B (RC=0.57) and GPT-OSS-120B (RC=0.60) achieve unconditional cost below brute-force. They optimize cost more aggressively, and their failed runs select cheap, coarse parameters.

- **Multi-round**: Most models exceed brute-force cost unconditionally due to the sunk-cost effect (failed multi-round runs accumulate cost from the 10-trial cap). Only GPT-OSS-120B (RC=0.78) stays below 1.

### C.7.2. PROMPT SENSITIVITY ANALYSIS

We perturbed the prompt template into 25 variants (paraphrasing, reordering sections, varying detail level) using frontier LLMs followed by manual review, then re-evaluated Claude-3.7-Sonnet on the ablation subset.

*Table 15.* Prompt sensitivity: mean shift from original results across 25 variants.

| Mode | Success Rate Shift | Efficiency Shift |
|---|---|---|
| Single-round | +0.069 | −0.047 |
| Multi-round | +0.072 | −0.090 |

The mean shifts are small (<7% for success rate, <9% for efficiency), confirming that our findings are robust to prompt wording variations.

## C.8. Tool-Augmented Tuning Ablation

We evaluate a tool-augmented setting where the LLM provides an initial parameter guess, then a programmatic brute-force search refines from that starting point. This tests whether LLM intuition can accelerate search algorithms.

*Table 16.* Tool-augmented (LLM initial guess + search) vs. LLM internal adjustment (multi-round only).

| Metric | $\Delta$ (Tool Search $-$ Internal) |
|---|---|
| Success Rate | +0.086 |
| Efficiency | $-0.520$ |
| Relative Cost | +2.363 |

Tool search increases success rate by 8.6%, confirming that LLM internal adjustment is a bottleneck for multi-round failures. However, relative cost increases to nearly $2\times$ brute-force. The reason: Claude (and similarly GPT-5 and Llama) tends to select conservative parameters that are more likely to succeed but more expensive. These initial guesses can largely undershoot the near-optimal value; the search algorithm stops at the 1st or 2nd iteration (for convergence checking), but the initial cost is already high due to unnecessarily fine resolutions.

**Concrete Examples.** Table 17 shows two cases from Euler 1D (CFL parameter, multi-round mode):

*Table 17.* Tool-augmented tuning examples showing conservative undershoots.

| Profile | Precision | LLM Guess | Near-Optimal | Cost Ratio |
|---|---|---|---|---|
| p3 | High | 0.10 | 0.50 | 13.8$\times$ |
| p1 | Low | 0.25 | 0.50 | 5.7$\times$ |

The LLM undershoots by 50–80%, incurring 5.7–13.8$\times$ cost overhead. This indicates that a good initial guess should land in an intermediate safe range, neither too aggressive (risking divergence) nor too conservative (wasting compute).

## C.9. Bayesian Optimization Baseline

We compare LLM performance against Bayesian Optimization with Gaussian Process surrogate (BO-GP), a classical black-box optimization approach that builds a surrogate model of the objective function and uses acquisition functions to guide parameter exploration.

**Implementation Details.** Our BO-GP baseline uses scikit-learn's Gaussian Process implementation with a Matern kernel ($\nu$=2.5), Upper Confidence Bound (UCB) acquisition function ($\kappa$=2.576), and automatic length-scale optimization. We run up to 10 BO iterations, matching the LLM multi-round budget. At each iteration, a Gaussian Process (GP) is fitted on previously observed samples (fallback to random samples for initial round); an Upper Confidence Bound (UCB) acquisition function is constructed with the GP. To propose a parameter based on acquisition values, we take a explore-and-exploit approach: we select the best out of 10,000 randomly-sampled new parameters (random-best), compare this with the solution proposed by L-BFGS-B on acquisition function (smart-best), and select the best out of the two. Additional GP hyperparameters include $\alpha$=1e-6 for numerical stability, $y$-normalization enabled, and 5 random restarts for optimizer initialization. The above hyper-parameters are all suggested in a standard implementation(Nogueira, 2014). The convergence criterion matches LLM experiments (task-specific distance threshold). Algorithm 3 provides the pseudocode.

**Why Multi-Round Only.** BO is inherently an iterative algorithm that requires observed data to train its surrogate model. A "single-round" BO evaluation would be equivalent to random sampling, as the GP cannot make informed predictions without prior observations. We therefore only compare BO against LLMs in multi-round mode.

**Results Summary.** Table 18 shows per-solver success rates comparing BO against the LLM average (across GPT-5, Claude-3.7-Sonnet, Llama-3-70B, Qwen3-32B, and GPT-OSS-120B). BO achieves 76%/71%/66% success rates at Low/Med/High accuracy levels, comparable to the LLM average (77%/76%/74%), but with larger solver-to-solver variation (25%–100% vs. more consistent LLM performance). BO excels on smooth, monotonic cost-accuracy relationships (Burgers

1D: 100% vs 78%; EPOCH 1D: 98% vs 87%) but struggles on discrete or multi-modal parameter spaces (HM Linear: 33% vs 81%; MPM 2D: 25% vs 34%).

*Table 18.* Bayesian Optimization vs LLM performance by solver (multi-round mode, success rate %). Low/Med/High = accuracy levels. LLM Avg is the mean across all five models.

| | **BO** | | | **LLM Avg** | | |
|---|---|---|---|---|---|---|
| **Solver** | Low | Med | High | Low | Med | High |
| Burgers 1D | 100 | 99 | 87 | 78 | 77 | 74 |
| Diff-React 1D | 76 | 48 | 50 | 81 | 74 | 75 |
| Euler 1D | 74 | 65 | 51 | 60 | 67 | 58 |
| Euler 2D | 93 | 98 | 100 | 91 | 90 | 89 |
| FEM 2D | 64 | 53 | 9 | 73 | 71 | 65 |
| HM Linear | 33 | 47 | 37 | 81 | 84 | 88 |
| HM Nonlinear | 100 | 97 | 94 | 93 | 91 | 88 |
| Heat 1D | 100 | 93 | 93 | 96 | 92 | 87 |
| Heat 2D | 52 | 35 | 60 | 66 | 78 | 74 |
| MPM 2D | 25 | 33 | 32 | 34 | 27 | 27 |
| NS 2D | 100 | 89 | 80 | 80 | 84 | 85 |
| **Overall** | **76** | **71** | **66** | **77** | **76** | **74** |

**BO Exploration Behavior.** BO's UCB acquisition function tends to select extreme parameter values in early iterations to maximize information gain, and other common acquisition functions share similar early-iteration behaviors (Mockus et al., 1978; Jones et al., 1998). While guaranteeing a theoretical competitive performance, this strategy is detrimental when cumulative costs accumulate, tasks terminate upon reaching accuracy thresholds, and iteration budgets are limited. In contrast, LLMs leverage physics intuition from pre-training to make more conservative initial guesses near typical working parameter ranges. This physics-informed exploration particularly benefits low-accuracy tasks and early exploration phases, explaining why LLMs achieve higher efficiency despite BO's sophisticated search strategy.

*Table 19.* Bayesian Optimization vs LLM efficiency by solver (multi-round mode). Values show cost efficiency (higher is better; 1.0 = brute-force search cost).

| | **BO** | | | **LLM Avg** | | |
|---|---|---|---|---|---|---|
| **Solver** | Low | Med | High | Low | Med | High |
| Burgers 1D | 0.36 | 0.41 | 0.67 | 0.78 | 0.70 | 0.85 |
| Diff-React 1D | 0.30 | 0.97 | 0.64 | 1.20 | 1.18 | 1.00 |
| EPOCH 1D | 0.64 | 0.57 | 0.58 | 0.95 | 0.99 | 1.04 |
| Euler 1D | 0.99 | 1.08 | 1.18 | 1.21 | 0.78 | 0.79 |
| Euler 2D | 0.60 | 0.62 | 0.57 | 0.42 | 0.38 | 0.49 |
| FEM 2D | 0.30 | 0.20 | 0.48 | 0.46 | 0.51 | 0.73 |
| HM Linear | 0.50 | 0.33 | 0.33 | 0.28 | 0.64 | 0.81 |
| HM Nonlinear | 0.46 | 0.49 | 0.89 | 0.23 | 0.13 | 0.26 |
| Heat 1D | 0.80 | 0.93 | 0.93 | 0.58 | 0.61 | 0.67 |
| Heat 2D | 6.13 | 0.37 | 1.02 | 16.81 | 1.62 | 0.86 |
| MPM 2D | 0.15 | 0.01 | 0.02 | 0.35 | 0.56 | 0.83 |
| NS 2D | 1.01 | 0.80 | 1.01 | 1.12 | 0.54 | 0.52 |
| **Overall** | **1.02** | **0.57** | **0.70** | **2.03** | **0.72** | **0.74** |

*Table 20.* Task group classification. Parameters are grouped by their role in numerical simulation, enabling analysis of LLM performance across parameter types.

| Group | Parameters | Description |
|---|---|---|
| Spatial | `n_space, dx, nx, n_grid_x,` ... | Grid/mesh resolution parameters affecting spatial discretization accuracy. Most common across all solvers. |
| Temporal | `cfl, dt, dt_multiplier, initial_dt` | Timestep and CFL parameters controlling temporal stability and accuracy. |
| Tolerance | `error_threshold, cg_tolerance, cg_atol, residual_threshold,` ... | Convergence tolerances for iterative solvers to control solution accuracy. |
| Misc | `beta, k, t_init, field_order,` ... | Physics-specific parameters (limiter coefficients, relaxation factors) varying by solver. |

---

**Algorithm 3** Bayesian Optimization with Gaussian Process

---

1: **Input:** objective $f$, bounds $\mathcal{B}$, init points $m$, $T = 10$, $\kappa = 2.576$
2: Initialize GP with Matern kernel ($\nu = 2.5$), $\alpha = 10^{-6}$, normalize-$y$ enabled, 5 optimizer restarts; $\mathcal{D} \leftarrow \emptyset$
3: **for** $i = 1$ to $m$ **do**
4:     Sample $x$ uniformly from $\mathcal{B}$; observe $y = f(x)$; $\mathcal{D} \leftarrow \mathcal{D} \cup \{(x, y)\}$
5: **end for**
6: **for** $t = 1$ to $T$ **do**
7:     Fit GP on $\mathcal{D}$; obtain $\mu(x), \sigma(x)$ for any $x$
8:     Compute UCB: $a(x) = \mu(x) + \kappa\sigma(x)$
9:     Sample 10,000 new $x$ uniformly from $\mathcal{B}$
10:     $x^{\text{rand}} \leftarrow \arg\max_{x \in \{10,000 \text{ new samples}\}} a(x)$               (random-best)
11:     $x^{\text{smart}} \leftarrow \arg\max_x a(x)$ via L-BFGS-B with random restarts          (smart-best)
12:     $x^* \leftarrow \arg\max_{x \in \{x^{\text{rand}}, x^{\text{smart}}\}} a(x)$
13:     Observe $y^* = f(x^*)$; $\mathcal{D} \leftarrow \mathcal{D} \cup \{(x^*, y^*)\}$
14:     **if** convergence criterion satisfied (task-specific distance threshold) **then**
15:         **break**
16:     **end if**
17: **end for**
18: **Output:** best $(x, y)$ in $\mathcal{D}$

---

### C.10. Task Group Classification

Table 20 presents the parameter-to-group mapping used throughout our analysis, categorizing parameters by their role in numerical simulation.

### C.11. Task Group Performance Plots

Figures 3b–3c (in the main paper) provide an aggregated summary of performance by task group across both inference modes. Figure 6 shows the detailed performance breakdown by task group, referenced from Section 4.

### C.12. Task Correlation Analysis

We investigate whether tasks within the same parameter group (Spatial, Temporal, Tolerance, Misc) exhibit correlated success patterns across model-accuracy configurations. If within-group correlations were strong, fine-tuning on one solver's tasks might transfer to other solvers with similar parameter types.

Our analysis computes Pearson correlations between task success rates across all (model, accuracy level) configurations (note: each correlation is computed from 5 models $\times$ 3 accuracy levels = 15 data points). When averaging correlations and performing statistical tests, we apply Fisher's z-transformation (z = arctanh(r)) to account for the bounded nature of correlation coefficients (which are not normally distributed), then transform back to the correlation scale for reporting.

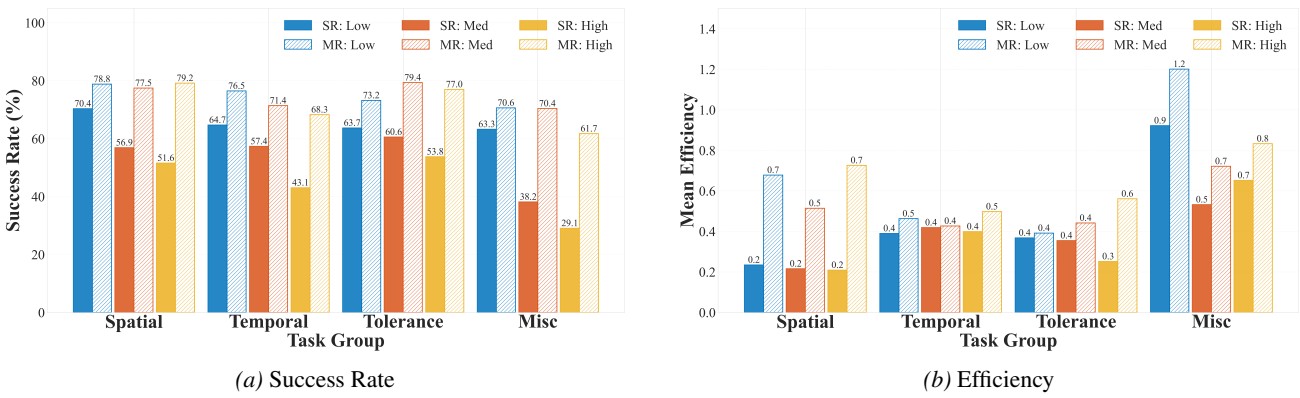

*(a)* Success Rate        *(b)* Efficiency

*Figure 6.* Performance by task group. Spatial and Tolerance parameters show consistently strong performance, while Misc parameters lag especially at higher accuracy levels.

We compare within-group correlations (tasks sharing the same parameter type) to between-group correlations (tasks from different parameter types).

Figures 7a and 7b show the average correlation between task groups for single-round and multi-round modes respectively. The diagonal (within-group) values are not notably higher than off-diagonal (between-group) values, confirming that parameter type does not strongly predict correlated difficulty.

Figures 8a and 8b show the full pairwise correlation matrices for all 25 tunable parameters. Tasks are grouped by parameter type (Spatial, Temporal, Tolerance, Misc), with black lines separating groups. No clear block-diagonal structure emerges in either mode, further supporting the conclusion that task difficulty is solver-specific rather than parameter-type-driven.

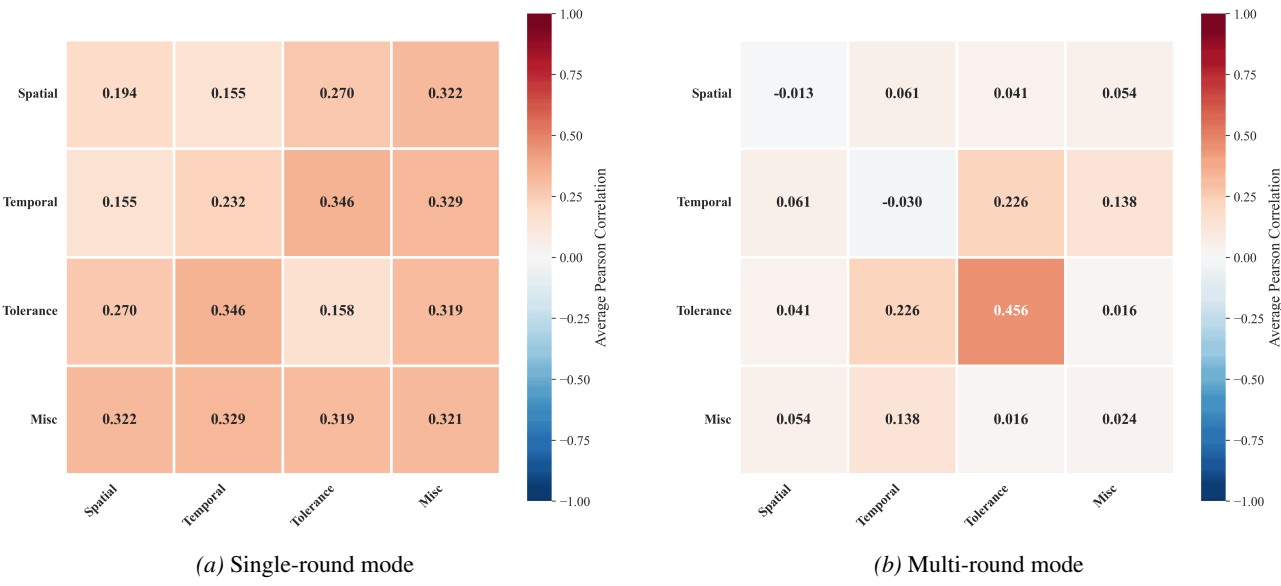

*(a)* Single-round mode        *(b)* Multi-round mode

*Figure 7.* Average Pearson correlation between task groups. Diagonal = within-group; off-diagonal = between-group.

## C.13. Summary Tables

Tables 21–23 provide precise numerical values corresponding to the main paper figures.

## C.14. Detailed Simulator-wise Results

Tables 24–31 present detailed per-solver results grouped by physics domain.

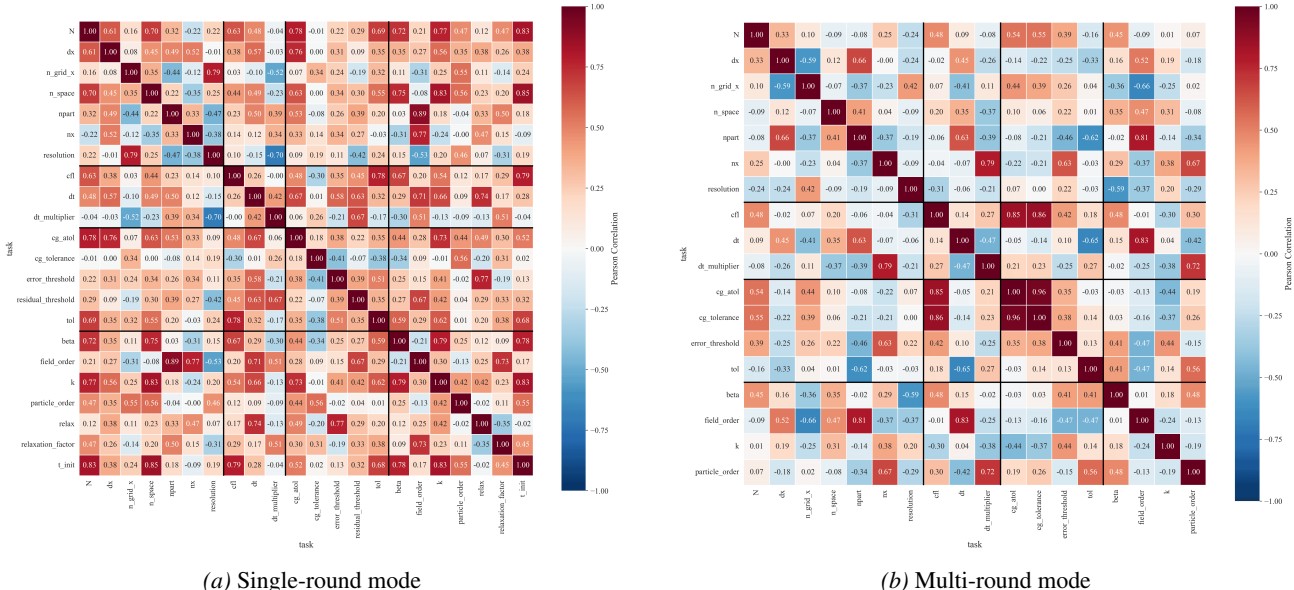

*(a)* Single-round mode        *(b)* Multi-round mode

*Figure 8.* Full pairwise task correlation matrix. Tasks grouped by parameter type: Spatial, Temporal, Tolerance, Misc. Black lines separate groups.

*Table 21.* Overall model performance (corresponds to Figure 2).

| Model | Success Rate (%) | | | Efficiency | |
|---|---|---|---|---|---|
| | Single | Multi | Δ | Single | Multi |
| GPT-5 | 65.0 | 81.1 | +16.1 | 0.27 | 0.57 |
| Claude-3.7-Sonnet | 54.0 | 67.6 | +13.6 | 0.36 | 0.43 |
| Llama-3-70B-Instruct | 45.6 | 72.0 | +26.3 | 0.19 | 0.45 |
| Qwen3-32B | 46.3 | 67.7 | +21.3 | 0.39 | 0.60 |
| GPT-OSS-120B | 53.4 | 80.8 | +27.4 | 0.45 | 0.86 |

*Table 22.* Performance by task group (corresponds to Figure 6).

| Task Group | Success Rate (%) | | | Efficiency | |
|---|---|---|---|---|---|
| | Single | Multi | Δ | Single | Multi |
| Spatial | 58.8 | 79.4 | +20.6 | 0.24 | 0.65 |
| Temporal | 55.1 | 72.1 | +17.0 | 0.40 | 0.46 |
| Tolerance | 59.4 | 76.5 | +17.2 | 0.32 | 0.46 |
| Misc | 43.5 | 67.6 | +24.1 | 0.68 | 0.90 |

*Table 23.* ICL variant comparison using Claude-3.7-Sonnet (corresponds to Figure 3).

| Variant | Success Rate (%) | | | Efficiency | |
|---|---|---|---|---|---|
| | Single | Multi | Δ | Single | Multi |
| Base | 46.7 | 64.1 | +17.3 | 0.328 | 0.367 |
| Idealized ICL | 63.8 | 53.2 | -10.6 | 0.445 | 0.377 |
| Cost-Ignorant ICL | 73.4 | 53.1 | -20.3 | 0.364 | 0.470 |
| Mixed-Accuracy ICL | 75.4 | 46.3 | -29.1 | 0.572 | 0.435 |

*Table 24.* **Single-round** results for **Fluid Dynamics** solvers. S: Success Rate (%), E: Efficiency.

| Solver | Model | Low | | Med | | High | | Avg | |
|---|---|---|---|---|---|---|---|---|---|
| | | S | E | S | E | S | E | S | E |
| Burgers 1D | GPT-5 | 53.9 | 1.07 | 41.7 | 0.71 | 23.8 | 0.40 | 39.8 | 0.67 |
| | Claude | 48.9 | 1.46 | 25.0 | 0.94 | 18.9 | 0.78 | 30.9 | 1.02 |
| | Llama | 62.2 | 0.55 | 56.7 | 0.26 | 21.7 | 0.85 | 46.9 | 0.50 |
| | Qwen | 33.3 | 1.26 | 25.0 | 0.68 | 15.6 | 0.95 | 24.6 | 0.93 |
| | GPT-OSS | 48.3 | 1.31 | 22.8 | 0.71 | 21.7 | 0.82 | 30.9 | 0.91 |
| Euler 1D | GPT-5 | 81.5 | 1.07 | 54.7 | 0.52 | 34.7 | 0.82 | 57.0 | 0.77 |
| | Claude | 83.3 | 1.33 | 23.5 | 0.71 | 7.4 | 0.70 | 38.1 | 0.87 |
| | Llama | 58.3 | 2.18 | 23.5 | 0.38 | 15.2 | 0.84 | 32.3 | 0.88 |
| | Qwen | 72.2 | 1.61 | 16.9 | 0.60 | 10.6 | 0.80 | 33.3 | 0.92 |
| | GPT-OSS | 74.1 | 1.59 | 24.5 | 0.84 | 7.4 | 0.65 | 35.3 | 0.95 |
| Euler 2D | GPT-5 | 91.7 | 0.08 | 93.1 | 0.11 | 73.7 | 0.16 | 86.2 | 0.11 |
| | Claude | 92.6 | 0.12 | 89.8 | 0.21 | 74.7 | 0.36 | 85.7 | 0.20 |
| | Llama | 91.7 | 0.08 | 85.6 | 0.13 | 86.9 | 0.33 | 88.1 | 0.15 |
| | Qwen | 92.6 | 0.15 | 85.6 | 0.13 | 84.8 | 0.31 | 87.7 | 0.18 |
| | GPT-OSS | 92.6 | 0.18 | 85.6 | 0.20 | 62.7 | 0.65 | 80.3 | 0.28 |
| NS Trans. 2D | GPT-5 | 40.3 | 0.89 | 34.0 | 0.46 | 22.1 | 0.36 | 32.1 | 0.53 |
| | Claude | 42.4 | 0.61 | 41.4 | 0.43 | 31.1 | 0.92 | 38.3 | 0.62 |
| | Llama | 25.7 | 0.59 | 21.5 | 0.18 | 25.0 | 0.68 | 24.1 | 0.42 |
| | Qwen | 43.8 | 0.76 | 34.3 | 0.61 | 32.9 | 0.30 | 37.0 | 0.51 |
| | GPT-OSS | 51.4 | 1.07 | 48.7 | 0.71 | 22.8 | 0.77 | 41.0 | 0.84 |

*Table 25.* **Multi-round** results for **Fluid Dynamics** solvers. S: Success Rate (%), E: Efficiency.

| Solver | Model | Low | | Med | | High | | Avg | |
|---|---|---|---|---|---|---|---|---|---|
| | | S | E | S | E | S | E | S | E |
| Burgers 1D | GPT-5 | 85.0 | 0.83 | 86.1 | 0.75 | 72.2 | 0.98 | 81.1 | 0.85 |
| | Claude | 77.8 | 0.72 | 65.0 | 0.54 | 62.2 | 0.65 | 68.3 | 0.63 |
| | Llama | 79.4 | 0.53 | 80.0 | 0.53 | 57.5 | 0.68 | 72.3 | 0.57 |
| | Qwen | 66.7 | 0.84 | 63.9 | 0.91 | 59.2 | 1.10 | 63.2 | 0.94 |
| | GPT-OSS | 71.7 | 0.70 | 71.7 | 0.49 | 83.3 | 0.67 | 75.6 | 0.62 |
| Euler 1D | GPT-5 | 73.1 | 1.77 | 61.1 | 0.87 | 41.8 | 0.90 | 58.7 | 1.11 |
| | Claude | 47.2 | 2.16 | 55.7 | 0.56 | 49.4 | 0.65 | 50.8 | 0.93 |
| | Llama | 53.7 | 0.47 | 79.9 | 0.70 | 72.1 | 0.90 | 68.6 | 0.67 |
| | Qwen | 58.3 | 3.06 | 60.0 | 0.54 | 36.9 | 0.71 | 51.7 | 1.06 |
| | GPT-OSS | 43.5 | 3.75 | 74.0 | 0.60 | 64.6 | 0.71 | 60.7 | 1.17 |
| Euler 2D | GPT-5 | 92.6 | 0.20 | 89.8 | 0.22 | 94.9 | 0.41 | 92.5 | 0.26 |
| | Claude | 65.7 | 0.06 | 68.1 | 0.13 | 70.7 | 0.20 | 68.2 | 0.12 |
| | Llama | 99.1 | 0.58 | 99.1 | 0.54 | 99.0 | 0.79 | 99.0 | 0.63 |
| | Qwen | 96.3 | 0.40 | 99.1 | 0.42 | 98.0 | 0.70 | 97.8 | 0.49 |
| | GPT-OSS | 89.8 | 0.56 | 86.1 | 0.52 | 98.0 | 0.63 | 91.3 | 0.57 |
| NS Trans. 2D | GPT-5 | 84.7 | 1.74 | 94.4 | 0.69 | 91.7 | 0.47 | 90.3 | 0.83 |
| | Claude | 76.4 | 1.38 | 75.7 | 0.58 | 83.3 | 0.47 | 78.5 | 0.72 |
| | Llama | 77.8 | 1.01 | 83.9 | 0.56 | 88.9 | 0.52 | 83.5 | 0.66 |
| | Qwen | 72.2 | 1.72 | 86.5 | 0.52 | 88.9 | 0.57 | 82.6 | 0.80 |
| | GPT-OSS | 83.3 | 1.24 | 83.9 | 0.38 | 80.6 | 0.64 | 82.6 | 0.67 |

*Table 26.* **Single-round** results for **Heat & Diffusion** solvers. S: Success Rate (%), E: Efficiency.

| Solver | Model | Low | | Med | | High | | Avg | |
|---|---|---|---|---|---|---|---|---|---|
| | | S | E | S | E | S | E | S | E |
| Heat 1D | GPT-5 | 88.0 | 0.38 | 80.0 | 0.36 | 55.4 | 0.05 | 74.5 | 0.19 |
| | Claude | 96.0 | 0.45 | 68.0 | 0.99 | 45.6 | 0.53 | 69.9 | 0.62 |
| | Llama | 100.0 | 0.01 | 92.0 | 0.05 | 82.4 | 0.23 | 91.5 | 0.05 |
| | Qwen | 98.0 | 0.47 | 68.0 | 0.78 | 35.9 | 0.29 | 67.3 | 0.47 |
| | GPT-OSS | 100.0 | 0.71 | 65.3 | 0.56 | 33.0 | 0.94 | 66.1 | 0.72 |
| Heat 2D | GPT-5 | 65.2 | 1.96 | 60.3 | 1.08 | 53.9 | 0.80 | 59.8 | 1.19 |
| | Claude | 49.4 | 4.28 | 37.8 | 1.37 | 33.3 | 0.67 | 40.2 | 1.58 |
| | Llama | 49.4 | 3.03 | 13.5 | 0.80 | 12.3 | 0.24 | 25.1 | 0.84 |
| | Qwen | 34.6 | 19.23 | 12.5 | 0.97 | 10.4 | 1.01 | 19.2 | 2.66 |
| | GPT-OSS | 48.8 | 2.74 | 24.1 | 1.14 | 14.3 | 0.76 | 29.1 | 1.33 |
| Diff-React 1D | GPT-5 | 70.4 | 0.60 | 82.5 | 0.88 | 70.0 | 0.47 | 74.3 | 0.63 |
| | Claude | 64.2 | 0.65 | 87.3 | 0.60 | 65.8 | 0.73 | 72.4 | 0.66 |
| | Llama | 66.7 | 0.23 | 50.5 | 0.82 | 32.5 | 0.64 | 49.9 | 0.49 |
| | Qwen | 69.1 | 0.51 | 50.5 | 0.84 | 35.0 | 0.59 | 51.5 | 0.63 |
| | GPT-OSS | 66.7 | 0.54 | 48.9 | 0.80 | 43.6 | 0.74 | 53.1 | 0.68 |

*Table 27.* **Multi-round** results for **Heat & Diffusion** solvers. S: Success Rate (%), E: Efficiency.

| Solver | Model | Low | | Med | | High | | Avg | |
|---|---|---|---|---|---|---|---|---|---|
| | | S | E | S | E | S | E | S | E |
| Heat 1D | GPT-5 | 96.0 | 0.88 | 92.0 | 0.71 | 88.1 | 0.86 | 92.0 | 0.81 |
| | Claude | 92.0 | 0.82 | 88.0 | 1.01 | 83.2 | 0.88 | 87.7 | 0.90 |
| | Llama | 92.0 | 0.66 | 93.3 | 0.51 | 86.8 | 0.53 | 90.7 | 0.56 |
| | Qwen | 92.0 | 0.85 | 88.7 | 0.73 | 84.6 | 0.81 | 88.4 | 0.80 |
| | GPT-OSS | 92.0 | 0.97 | 93.3 | 0.83 | 92.3 | 0.85 | 92.5 | 0.88 |
| Heat 2D | GPT-5 | 94.2 | 3.68 | 96.2 | 1.21 | 91.0 | 0.93 | 93.8 | 1.61 |
| | Claude | 44.6 | 6.07 | 63.5 | 1.38 | 72.0 | 1.04 | 60.0 | 2.06 |
| | Llama | 83.7 | 0.76 | 98.6 | 1.52 | 85.9 | 0.79 | 89.4 | 0.97 |
| | Qwen | 45.8 | 30.54 | 60.8 | 1.54 | 44.5 | 0.72 | 50.4 | 3.24 |
| | GPT-OSS | 62.9 | 6.01 | 81.9 | 1.74 | 77.8 | 0.78 | 74.2 | 2.01 |
| Diff-React 1D | GPT-5 | 87.7 | 0.63 | 74.0 | 0.42 | 75.8 | 0.75 | 79.2 | 0.58 |
| | Claude | 77.8 | 0.42 | 67.6 | 0.36 | 71.7 | 0.97 | 72.4 | 0.52 |
| | Llama | 70.4 | 0.43 | 61.9 | 0.90 | 91.7 | 0.49 | 74.6 | 0.57 |
| | Qwen | 74.1 | 0.55 | 55.2 | 0.98 | 78.3 | 0.44 | 69.2 | 0.62 |
| | GPT-OSS | 66.7 | 0.83 | 70.8 | 0.41 | 74.2 | 0.50 | 70.5 | 0.55 |

*Table 28.* **Single-round** results for **Solid Mechanics** solvers. S: Success Rate (%), E: Efficiency.

| Solver | Model | Low | | Med | | High | | Avg | |
|---|---|---|---|---|---|---|---|---|---|
| | | S | E | S | E | S | E | S | E |
| FEM 2D | GPT-5 | 73.0 | 0.13 | 80.0 | 0.14 | 39.7 | 0.47 | 64.3 | 0.20 |
| | Claude | 73.0 | 0.19 | 58.4 | 0.14 | 52.6 | 0.31 | 61.3 | 0.20 |
| | Llama | 65.5 | 0.10 | 63.9 | 0.26 | 67.9 | 0.22 | 65.8 | 0.18 |
| | Qwen | 70.0 | 0.15 | 63.9 | 0.22 | 51.3 | 0.21 | 61.7 | 0.19 |
| | GPT-OSS | 42.0 | 0.23 | 51.9 | 0.39 | 49.6 | 0.20 | 47.8 | 0.26 |
| MPM 2D | GPT-5 | 38.9 | 0.35 | 47.2 | 0.17 | 39.4 | 0.15 | 41.8 | 0.21 |
| | Claude | 33.3 | 0.20 | 41.7 | 0.44 | 27.8 | 0.55 | 34.3 | 0.36 |
| | Llama | 16.7 | 0.00 | 5.6 | 0.00 | 11.1 | 0.00 | 11.1 | 0.00 |
| | Qwen | 38.9 | 0.45 | 33.3 | 0.50 | 27.8 | 0.57 | 33.3 | 0.50 |
| | GPT-OSS | 27.8 | 0.11 | 33.3 | 0.16 | 33.3 | 0.32 | 31.5 | 0.18 |

*Table 29.* **Multi-round** results for **Solid Mechanics** solvers. S: Success Rate (%), E: Efficiency.

| Solver | Model | Low | | Med | | High | | Avg | |
|---|---|---|---|---|---|---|---|---|---|
| | | S | E | S | E | S | E | S | E |
| FEM 2D | GPT-5 | 90.0 | 0.44 | 91.5 | 0.60 | 100.0 | 0.72 | 93.8 | 0.58 |
| | Claude | 57.5 | 0.43 | 63.2 | 0.32 | 53.8 | 0.38 | 58.2 | 0.38 |
| | Llama | 46.0 | 0.10 | 48.5 | 0.21 | 37.2 | 0.38 | 43.9 | 0.20 |
| | Qwen | 75.5 | 0.45 | 66.9 | 0.61 | 60.3 | 0.68 | 67.5 | 0.57 |
| | GPT-OSS | 85.0 | 0.40 | 89.2 | 0.72 | 86.8 | 0.95 | 87.0 | 0.65 |
| MPM 2D | GPT-5 | 55.6 | 0.36 | 36.1 | 1.29 | 27.8 | 0.93 | 39.8 | 0.76 |
| | Claude | 38.9 | 0.13 | 33.3 | 0.18 | 33.3 | 0.36 | 35.2 | 0.21 |
| | Llama | 33.3 | 0.48 | 25.0 | 0.05 | 16.7 | 1.00 | 25.0 | 0.28 |
| | Qwen | 36.1 | 1.02 | 33.3 | 0.62 | 33.3 | 1.00 | 34.3 | 0.86 |
| | GPT-OSS | 50.0 | 0.37 | 33.3 | 0.58 | 33.3 | 0.87 | 38.9 | 0.57 |

*Table 30.* **Single-round** results for **Plasma Physics** solvers. S: Success Rate (%), E: Efficiency.

| Solver | Model | Low | | Med | | High | | Avg | |
|---|---|---|---|---|---|---|---|---|---|
| | | S | E | S | E | S | E | S | E |
| EPOCH 1D | GPT-5 | 82.2 | 0.05 | 73.9 | 0.02 | 80.4 | 0.01 | 78.9 | 0.03 |
| | Claude | 77.8 | 0.28 | 74.2 | 0.29 | 70.8 | 0.29 | 74.3 | 0.29 |
| | Llama | 51.1 | 0.15 | 42.9 | 0.13 | 61.6 | 0.10 | 51.9 | 0.13 |
| | GPT-OSS | 93.3 | 0.40 | 84.1 | 0.56 | 79.2 | 0.55 | 85.5 | 0.50 |
| HM Linear | GPT-5 | 97.6 | 0.35 | 91.1 | 0.49 | 85.2 | 0.65 | 91.3 | 0.48 |
| | Claude | 81.3 | 0.06 | 61.9 | 0.22 | 64.9 | 0.22 | 69.4 | 0.14 |
| | Llama | 77.2 | 0.04 | 54.2 | 0.59 | 36.7 | 0.70 | 56.0 | 0.25 |
| | Qwen | 81.3 | 0.05 | 56.2 | 0.80 | 45.0 | 0.65 | 60.8 | 0.30 |
| | GPT-OSS | 88.4 | 0.14 | 62.3 | 0.28 | 44.7 | 0.27 | 65.1 | 0.22 |
| HM Nonlinear | GPT-5 | 96.0 | 0.05 | 90.0 | 0.04 | 82.5 | 0.06 | 89.5 | 0.05 |
| | Claude | 83.7 | 0.01 | 77.7 | 0.01 | 68.3 | 0.02 | 76.6 | 0.01 |
| | Llama | 54.3 | 0.01 | 32.0 | 0.01 | 36.1 | 0.10 | 40.8 | 0.02 |
| | Qwen | 88.9 | 0.02 | 79.3 | 0.01 | 73.0 | 0.02 | 80.4 | 0.02 |
| | GPT-OSS | 100.0 | 0.04 | 98.0 | 0.03 | 82.5 | 0.04 | 93.5 | 0.04 |
| CGYRO | GPT-5 | 97.5 | 0.44 | 66.7 | 1.00 | 100.0 | 0.32 | 88.1 | 0.52 |
| | Claude | 47.5 | 0.50 | 0.0 | – | 0.0 | – | 15.8 | 0.50 |
| | Llama | 75.0 | 0.34 | 0.0 | – | 0.0 | – | 25.0 | 0.34 |
| | GPT-OSS | 87.5 | 0.43 | 33.3 | 1.02 | 0.0 | – | 40.3 | 0.66 |

*Table 31.* **Multi-round** results for **Plasma Physics** solvers. S: Success Rate (%), E: Efficiency.

| Solver | Model | Low | | Med | | High | | Avg | |
|---|---|---|---|---|---|---|---|---|---|
| | | S | E | S | E | S | E | S | E |
| EPOCH 1D | GPT-5 | 80.7 | 0.25 | 73.9 | 0.22 | 78.3 | 0.14 | 77.7 | 0.20 |
| | Claude | 94.1 | 0.80 | 81.4 | 1.03 | 67.6 | 0.94 | 81.0 | 0.92 |
| | Llama | 75.6 | 0.39 | 70.4 | 0.51 | 67.7 | 0.56 | 71.2 | 0.48 |
| | GPT-OSS | 97.8 | 2.84 | 94.8 | 3.30 | 92.8 | 4.71 | 95.1 | 3.53 |
| HM Linear | GPT-5 | 98.8 | 0.53 | 97.8 | 0.81 | 100.0 | 0.86 | 98.9 | 0.72 |
| | Claude | 59.9 | 0.06 | 72.1 | 0.17 | 85.6 | 0.21 | 72.5 | 0.13 |
| | Llama | 66.7 | 0.12 | 65.3 | 0.35 | 64.4 | 0.41 | 65.5 | 0.26 |
| | Qwen | 84.9 | 0.09 | 68.3 | 0.14 | 63.0 | 0.86 | 72.1 | 0.22 |
| | GPT-OSS | 100.0 | 0.42 | 98.7 | 0.62 | 100.0 | 0.58 | 99.6 | 0.53 |
| HM Nonlinear | GPT-5 | 100.0 | 0.19 | 98.0 | 0.11 | 91.7 | 0.36 | 96.6 | 0.20 |
| | Claude | 100.0 | 0.02 | 96.0 | 0.01 | 91.7 | 0.05 | 95.9 | 0.02 |
| | Llama | 68.6 | 0.11 | 76.6 | 0.03 | 76.6 | 0.09 | 73.9 | 0.07 |
| | Qwen | 92.9 | 0.04 | 81.7 | 0.07 | 83.7 | 0.15 | 86.1 | 0.08 |
| | GPT-OSS | 100.0 | 0.19 | 98.0 | 0.16 | 94.4 | 0.28 | 97.5 | 0.20 |
| CGYRO | GPT-5 | 95.0 | 0.85 | 100.0 | 0.94 | 100.0 | 0.84 | 98.3 | 0.88 |
| | Claude | 78.9 | 0.56 | 33.3 | 1.09 | 0.0 | – | 37.4 | 0.78 |
| | Llama | 100.0 | 0.69 | 66.7 | 0.74 | 100.0 | 0.59 | 88.9 | 0.67 |
| | GPT-OSS | 100.0 | 1.09 | 100.0 | 0.76 | 100.0 | 0.64 | 100.0 | 0.81 |

