# OpenReview forum: "SimulCost: A Cost-Aware Benchmark and Toolkit for Automating Physics Simulations with LLMs"
_ICML.cc/2026/Conference — ICML 2026 regular_

### Official Review · Reviewer_r3rf · 2026-03-09

**Soundness:** 3
**Presentation:** 3
**Significance:** 3
**Originality:** 3
**Overall Recommendation:** 4
**Confidence:** 3

**Summary:**

This paper introduces SIMULCOST, a novel benchmark designed to evaluate Large Language Models (LLMs) on cost-sensitive parameter tuning within physics simulation workflows. Recognizing that current scientific agent benchmarks often treat external tool calls as computationally free, SIMULCOST rigorously accounts for the computational burden (simulation time/resources) across 12 simulators spanning fluid dynamics, solid mechanics, and plasma physics. The benchmark comprises 2,916 single-round and 1,900 multi-round tasks, evaluating models based on a proposed efficiency metric that balances task success against analytical simulation costs.

While the motivation addresses a critical and under-discussed blind spot in scientific agent evaluation, the execution exhibits severe methodological flaws. The benchmark artificially simplifies simulation tuning to isolated 1D search problems, severely restricts the agent interface while drawing broad conclusions about LLM capabilities, entirely omits basic adaptive search baselines, and relies on a flawed aggregate efficiency metric that ignores the sunk computational cost of failed runs. These issues compromise the benchmark's practical utility and the validity of its core claims.

**Compliance With Llm Reviewing Policy:**

Affirmed.

**Final Justification:**

**New:**

**My final recommendation is Weak Accept.** The paper addresses an important and underexplored problem: evaluating LLM-based scientific agents under realistic simulation cost rather than treating tool calls as free. I find the benchmark direction meaningful, and the overall curation effort across multiple physics simulators is valuable. **Additionally, I appreciate the author's attitude of maintaining discussions and feedback with me.**

**My evaluation changed positively after the rebuttal and follow-up responses.** In particular, the authors provided the additional evidence that was missing before, including unconditional relative-cost reporting, non-LLM optimization baselines, and a tool-augmented setting. These results substantially improve the empirical support of the paper and make the main conclusions more convincing. While limitations remain, especially the restriction to 1D tuning and the still-limited tool-augmented setting, I now view these as **scope limitations** rather than fatal flaws. Overall, the added experiments increased my confidence in the work, so **I raise my score from 3 to 4.**

Best wishes,

Your Reviewer r3rf.

—————————————————————UPDATE—————————————————————————————

Old:

The paper addresses an important and genuinely underexplored problem: evaluating LLM-based scientific agents under realistic simulation cost rather than treating tool calls as free. I view this motivation as strong, and I also appreciate the substantial effort behind curating tasks across a diverse set of physics simulators. The benchmark framing is interesting and potentially useful to the community.

The rebuttal was thoughtful and clarified several design choices, especially the rationale for separating success rate from efficiency, the intended scope of the 1D tuning setting, and the authors’ view of why stronger optimization baselines would likely reinforce rather than overturn the paper’s current conclusions. These explanations improved my understanding of the benchmark and made the authors’ perspective clearer.

However, the central concerns in my original review still depend on evidence that has not yet been provided in the current discussion. In particular, the rebuttal does not include the actual unconditional cost results, does not yet report the stronger optimization baselines that I believe are important for positioning the benchmark, and does not provide evaluation beyond the current restricted 1D / non-tool-augmented setting. While I appreciate the authors’ commitment to adding these components in a future revision, I do not think it is appropriate to upgrade the paper based primarily on promised experiments rather than demonstrated results. Because these issues are, in my opinion, extremely important for this article.

**Overall, I believe the problem formulation is valuable and the benchmark direction is promising.** Although I still think the current empirical support is insufficient to substantiate the strongest claims in the paper, in accordance with my commitment during the discussion, **I will reduce my negative confidence level to 2 for the AC to comprehensively consider the opinions of all reviewers.**
This will be at a borderline level, and our discussion has already reflected the contributions and deficiencies of this work, enabling the AC to make a comprehensive assessment. **Therefore, I tend to take a neutral stance to minimize the impact on the final outcome of this article.**

**Key Questions For Authors:**

Q1: Could you report an unconditional, end-to-end cost metric (e.g., Expected Total Compute per Successful Task, or Success Rate under a strict FLOP budget) that includes the compute wasted on failed runs? Does the efficiency ranking of the models hold under this metric?

Q2: Have you considered comparing the LLMs against more traditional, robust optimization algorithms for the multi-round tasks, such as CMA-ES, Nelder-Mead, or specific adaptive mesh refinement heuristics? How do LLMs fare against these methods?

Q3: How do you anticipate the LLMs would perform if the benchmark allowed for the simultaneous tuning of 2 or 3 coupled parameters? Are there plans to extend SIMULCOST to capture these multi-dimensional interactions?

Q4: How would the efficiency and success rates change if the LLM were permitted to act as a fully tool-augmented agent (e.g., generating code to run bisection searches or line searches) rather than guessing the parameters conversationally?

Based on the considerations outlined above and out of responsibility to the ICML review process, I assigned an initial score of 3: Weak reject. Should the authors effectively address these concerns in their rebuttal and demonstrate to me that their contributions are reliable and useful, I will consider raising the score.

**Limitations:**

The authors have included a transparent limitations section. It would be beneficial to explicitly acknowledge that the current framework simplifies the evaluation to 1D continuous/discrete parameter tuning, which sidesteps the combinatorial explosion of interacting parameters found in realistic scientific computing scenarios.

**Strengths And Weaknesses:**

Strengths:

S1. Crucial Evaluation Blind Spot: The premise is outstanding. In scientific machine learning, the cost of invoking a tool (e.g., solving a Navier-Stokes equation) often dwarfs the LLM's token inference cost. Auditing this resource consumption is a highly necessary step for the field.

S2. Platform-Independent Cost Accounting: Formulating analytical cost metrics based on algorithmic complexity (FLOPs/iterations) rather than purely relying on hardware-dependent wall-clock time is a smart, scalable benchmark design.

S3. Extensive Domain Curation: Curating 12 distinct simulators across diverse physics domains (fluid, solid, plasma) requires substantial expert-in-the-loop effort and lends significant domain credibility to the dataset.

Weaknesses:

W1. Efficiency Metric Aggregation (Exclusion of Sunk Costs): While Equation (1) defines per-task efficiency sensibly, the aggregate efficiency (e.g., in Figure 2b) is calculated only over successful tasks. In practical physics optimization, failed simulations (e.g., solver divergence) still consume substantial computational resources. Excluding these failures from the aggregate metric may inadvertently overstate the practical economy of LLMs. An unconditional cost metric would better reflect the true burden on the user.

W2. Absence of Standard Optimization Baselines: For the monotonic multi-round parameters, the benchmark compares LLMs primarily against brute-force scanning and a single BO-GP baseline. Given that parameter tuning is fundamentally an optimization problem, the evaluation would be significantly strengthened by including standard adaptive and heuristic search baselines, such as CMA-ES, Nelder-Mead, or adaptive mesh refinement heuristics. Without these, it is difficult to accurately position the LLM's performance against the broader landscape of traditional optimization.

W3. Task Simplification to 1D Search: The benchmark explicitly restricts tasks to single-parameter tuning while holding all other variables fixed to rule-of-thumb values. While understandable for a first-version benchmark, real-world simulation configurations are challenging precisely due to coupled parameter interactions (e.g., spatial resolution vs. time step size). This simplification limits the benchmark's ability to evaluate the LLMs on complex, multi-dimensional scientific reasoning.

W4. Constrained Agent Interface vs. Practical Recommendations: The paper concludes that practitioners might be better off using scanning algorithms rather than relying on LLMs. However, the benchmark evaluates LLMs in a restricted, text-only conversational mode without allowing them to use external tools (e.g., writing Python scripts to invoke traditional search routines or read solver logs). Evaluating a tool-augmented agent baseline would provide a fairer comparison and make the final recommendations much stronger.

---

> ### Author Rebuttal · Authors · 2026-03-31
>
> We thank the reviewer for the thorough review and for recognizing the evaluation blind spot, platform-independent cost accounting, and extensive domain curation.
>
> > "the aggregate efficiency...is calculated only over successful tasks...failed simulations still consume substantial computational resources. Excluding these failures...may inadvertently overstate the practical economy of LLMs."
>
> First, failed runs are already penalized by SR. Also, by definition, failed tasks produce useless results, hence the 'efficiency' should be defined as zero. In that case, including cases with E=0 would collapse the geometric mean to zero.
>
> We note that no single metric captures both reliability and cost-awareness; However, this also gives better guidance in practice: one can first filter by SR to have reliable models, then rank by efficiency.
>
> Still, we will add a unconditional "relative cost" (C_llm/C_bf for all tasks including failures) in an appendix; we expect rankings to remain consistent, because most failed runs hit the 10-trials cap and have large costs accumulated.
>
> > "the evaluation would be significantly strengthened by including...CMA-ES, Nelder-Mead, or adaptive mesh refinement heuristics"
>
> Before analysising specific candidate baselines, we note that the main finding of this benchmark is "LLMs using internal reasoning are not as efficient as even brute-force search." Even if a new baseline outperforms our naive baseline, it only strengthens this conclusion. New baselines will not change the core finding.
>
> *Monotonic parameters.* We will add in revision:
>
> 1. **Bisection** [3]: though our Algorithm 1 already approximates this.
> 2. **Nelder-Mead** [2]: degenerates to a two-point simplex in 1D, but included for completeness.
> 3. **(1+1)-ES**: CMA-ES does not support 1D [1], but when n=1 the evolutionary strategy reduces to a (1+1)-ES, which we can implement directly.
>
> *Non-monotonic parameters.* Adjacent parameter values can both produce similar but poor results; hence here, the solver feedback by comparing neighbor parameters does not at all indicate proximity to a convergent solution. In short, the optimization function landscape is not unimodal, ruling out golden-section search [4]; and, Nelder-Mead risks local optima. Exhaustive grid search (Algorithm 2) is the only sound baseline. Multi-round LLM tuning is also not applicable: without a meaningful local feedback, the LLM's multi-turn refinement degenerates into random trial-and-error.
>
> *Regarding AMR:* this refines spatial meshes during simulation, not a parameter optimization method. Our parameters include dt, CFL, integration order, and tolerances, none of which AMR addresses.
>
> > "restricts tasks to single-parameter tuning...limits the benchmark's ability to evaluate...complex, multi-dimensional scientific reasoning."
>
> This mirrors real practice: experts isolate parameters for trackable changes. Varying multiple obscures cause-and-effect. Reference feasibility reinforces this: with n=20 grid points, 1-parameter search requires 20 evaluations vs. 8,000 for 3-parameter joint search (Section 5). This scope and trade-off are acknowledged in Limitations (Section 5); we will re-emphasize in methodology.
>
> > "evaluates LLMs in a restricted, text-only conversational mode without...external tools"
>
> This is a benchmark measuring cost-awareness, not a methods paper. However, tool-augmented tuning is not yet standard in this regard; most scientific agent benchmarks (Table 1) rely on internal reasoning to make adjustments. Still, we will add an ablation where the LLM provides an initial guess and refinement scheme, testing whether intuition and planning accelerate search. Our open-sourced toolkit's standardized APIs support this kind of extension.
>
> Given that LLMs' internal reasoning alone yields only 0.4-0.7 efficiency (Section 4), we expect tool-augmented agents that can invoke programmatic search to perform substantially better, (if true), will largely enhance our finding (should not rely on LLM's own reasoning for parameter tuning). We will augment the discussion in Section 6 accordingly.
>
> We are committed to all suggested revisions. The reviewer stated they "will consider raising the score" if concerns are addressed; we hope these clarifications demonstrate our contributions are reliable and useful.
>
> [1] N. Hansen, "The CMA Evolution Strategy: A Tutorial," arXiv:1604.00772, 2023. (pycma 1D limitation: https://github.com/CMA-ES/pycma/issues/86)
> [2] J. A. Nelder & R. Mead, "A Simplex Method for Function Minimization," *The Computer Journal*, 1965. (https://en.wikipedia.org/wiki/Nelder-Mead_method)
> [3] https://en.wikipedia.org/wiki/Bisection_method
> [4] https://en.wikipedia.org/wiki/Golden-section_search

---

> > ### Author Rebuttal · Reviewer_r3rf · 2026-04-02
> >
> > Thank you for the thoughtful rebuttal. I appreciate the authors’ clear explanations and their commitment to strengthening the paper in a future revision.
> >
> > That said, my primary concerns are not fully resolved in the current rebuttal. The response usefully clarifies the authors’ design choices, but several of the central issues I raised still depend on experiments that have not yet been provided, including unconditional cost reporting, stronger optimization baselines, and evaluation beyond the current restricted setting. The discussion of 1D tuning and tool-augmented agents is helpful, but it does not fully remove the limitations of the current benchmark formulation.
> >
> > Since these issues concern the core empirical support and would require substantial additions to the paper, I select (c) and maintain my current score.

---

> > > ### Author Response · Authors · 2026-04-04
> > >
> > > Thanks for your quick response, especially in pinning down the experiments you would like to see.
> > >
> > > Within the rebuttal window, we can provide:
> > >
> > > - Non-LLM optimization baselines, although they will not change one of our core findings: "the current pass@K metrics, commonly used in this community, are unrealistic and inefficient." These are fast to run without LLM API requirements.
> > > - Non-conditional relative-cost metric report, which only requires post-processing our existing database.
> > > - LLM + better initial guidance on the tool search. Although we are trying our best to complete this ablation within the short rebuttal window, it may need a few more days to debug, test, and run thoroughly with the best results. But it is surely completable before camera ready.
> > >
> > > We will keep you posted and let you know when any of the above are complete. We would appreciate your re-evaluation accordingly (especially for the first 2 bullet points, as the third one takes longer. We do not want to rush unverified results in a short window, but a few weeks is more than enough for 1 ablation).
> > >
> > >
> > > # Added promised ablations
> > >
> > > Regarding **Unconditional Relative Cost**, pls refer to our reply to Reviewer LesF: https://openreview.net/forum?id=ww57OvgpP9&noteId=8qPjJb552Q
> > >
> > > The overall takes are:
> > > 1. Rankings are consistent across metrics: models with higher conditional efficiency have lower unconditional relative cost. Our core conclusion is unchanged.
> > > 2. In single-round, Qwen3-32B and GPT-OSS-120B achieve unconditional cost below brute-force. They tend to optimize cost more aggressively, and their failed runs are cheap and coarse, keeping total cost low.
> > > 3. In multi-round, most models exceed brute-force cost unconditionally, confirming the sunk-cost concern in our previous reply: failed multi-round runs accumulate large costs from the 10-trial cap. Only GPT-OSS-120B (0.78) stays below 1.
> > >
> > > Regarding **Prompt Sensitivity** and **Non LLM baselines**, pls see our reply to Reviewer AGU6: https://openreview.net/forum?id=ww57OvgpP9&noteId=twxtt8LBmk
> > >
> > > The overall takes are:
> > >
> > > - The mean shifts caused by prompt re-phrasing are insignificant for both success rate (6-7%) and efficiency (- 4-9%). Prompt wording does not substantially change the results.
> > > - Bisection performs very similarly to brute-force in both success rate and cost (1.08x); Nelder-Mead achieves slightly higher success rate but at 1.42x cost; (1+1)-ES proved computationally impractical (~5-12x more computational cost).
> > > - These non-LLM baselines justify our choice of brute-force scan, as the other methods are either comparable or too expensive.
> > >
> > > ## LLM Initial Guess + Tool Search (Reviewer r3rf Q4, LaML)
> > >
> > > We evaluate a tool-augmented setting where the LLM provides an initial parameter guess for the brute-force search, then a programmatic search refines from that starting point. Due to the search algorithm's nature, results are only for multi-round (MR) mode:
> > >
> > > | Metric | Delta (Tool Search MR - Internal Adjustment MR) |
> > > |---|---|
> > > | Success Rate | +0.086 |
> > > | Efficiency | -0.520 |
> > > | Rel. Cost | +2.363 |
> > >
> > > Tool search increases the success rate by 8.6%, confirming LLM internal adjustment is a bottleneck for the MR failures. The relative cost, however, increases to nearly 2-fold. The reason: the LLM's initial guess enters the search algorithm as the starting point. As we noted above in the unconditional cost table, Claude used in our ablation study (similarly GPT and Llama) tends to select more conservative parameters (more likely to succeed but more expensive). Those choices can largely overshoot or undershoot the near-optimal; the search likely stops at the 1st or 2nd iteration (for checking convergence), but the initial cost is already high due to unnecessarily high resolutions.
> > >
> > > Two concrete examples selected from the database (euler_1d, CFL, MR mode):
> > >
> > > | profile | precision | LLM initial guess | Near Optimal | Undershoot | MR Cost ratio |
> > > |---|---|---|---|---|---|
> > > | p3 | high | 0.10 | 0.50 | 80% | 13.8x |
> > > | p1 | low | 0.25 | 0.50 | 50% | 5.7x |
> > >
> > > This indicates that a good initial guess from the LLM should land in an intermediate safe range, avoiding both overshoot and undershoot, while still benefit from getting convergence by the search algorithm. We will include this as a new future implication.
> > >
> > > Thank you for all the constructive feedback!

---

### Official Review · Reviewer_AGU6 · 2026-03-11

**Soundness:** 2
**Presentation:** 2
**Significance:** 2
**Originality:** 2
**Overall Recommendation:** 4
**Confidence:** 3

**Summary:**

This paper introduces SIMULCOST, a benchmark for evaluating LLMs on cost-sensitive parameter tuning across 12 physics simulators. It measures both success rate and computational cost efficiency, comparing LLMs against brute-force scanning and Bayesian optimization.

**Compliance With Llm Reviewing Policy:**

Affirmed.

**Final Justification:**

The rebuttal resolves some concerns, but the core issues (narrow task formulation relative to claims, underpowered ablations) and the substantial volume of revision promises (new baselines, prompt sensitivity, tool-invocation ablation) make it difficult to evaluate the paper as-is. I am willing to update it to a weak accept.

**Key Questions For Authors:**

1. Have you considered evaluating LLMs in a setting where they can choose to invoke a scanning algorithm as a tool, rather than doing parameter tuning via internal reasoning?
2. What is the justification for excluding failed tasks from the efficiency geometric mean? This seems to reward models that succeed rarely but efficiently, while penalizing models that attempt more tasks.
3. How sensitive are the results to the specific prompt templates used? Have you conducted any prompt sensitivity analysis?

**Limitations:**

yes

**Strengths And Weaknesses:**

Strengths:

1. The paper highlights an important gap between standard correctness-based benchmarks (eg pass@k) and real scientific workflows where tool execution cost dominates. The observation that existing evaluation paradigms implicitly treat tool calls as free is a valuable insight
2. The analysis showing that within group parameter correlations do not significantly exceed between group correlations is an informative negative result. This suggests limited opportunities for simple transfer learning across related simulator parameters.

Weaknesses:

1. The benchmark restricts tuning to a single parameter at a time while fixing others to rule of thumb values. While the authors acknowledge this limitation, the simplification may significantly narrow the scope of what is being evaluated
2. The primary baseline is brute-force grid scan, which is also used to obtain the reference solutions. This makes the evaluation somewhat circular
3. In general, the coverage with baselines, model families and reasoning efforts could be improved and updated with latest models

---

> ### Author Rebuttal · Authors · 2026-03-31
>
> We thank the reviewer for recognizing that existing benchmarks implicitly treat tool calls as free, and for highlighting the informative negative result on parameter correlations.
>
> > "restricts tuning to a single parameter...may significantly narrow the scope of what is being evaluated"
>
> This mirrors real practice: experts isolate parameters for track-able changes. Varying multiple obscures cause-and-effect. Reference feasibility reinforces this: with n=20 grid points, 1-parameter search requires 20 evaluations vs. 8,000 for 3-parameter joint search (Section 5). This scope and trade-off are acknowledged in Limitations (Section 5); we will re-emphasize in methodology.
>
> > "The primary baseline is brute-force grid scan, which is also used to obtain the reference solutions...somewhat circular"
>
> Simulation tuning requires running simulations to find references; we chose brute-force for simplicity. The reference cost is only a per-instance normalization factor (Eq. 1); all baselines are instance-normalized using the same costs, hence ranked consistently.
>
> We note this definition does not cap the efficiency at 1: LLMs can exceed efficiency 1.0 since the parameter space is continuous, e.g., GPT-OSS-120B at low accuracy in single-round (Figure 2b).
>
> > "the coverage with baselines...could be improved"
>
> Before analyzing specific candidate baselines, we note that the main finding of this benchmark is "LLMs using internal reasoning are not as efficient as even brute-force search." Even if a new baseline outperforms our naive baseline, it only strengthens this conclusion. New baselines will not change the core finding.
>
> *Monotonic parameters.* We will add in revision:
>
> 1. **Bisection** [3]: though our Algorithm 1 already approximates this.
> 2. **Nelder-Mead** [2]: degenerates to a two-point simplex in 1D, but included for completeness.
> 3. **(1+1)-ES**: CMA-ES does not support 1D [1], but when n=1 the evolutionary strategy reduces to a (1+1)-ES, which we can implement directly.
>
> *Non-monotonic parameters.* Adjacent parameter values can both produce similar but poor results; hence here, the solver feedback by comparing neighbor parameters does not at all indicate proximity to a convergent solution. In short, the optimization function landscape is not unimodal, ruling out golden-section search [4]; and, Nelder-Mead risks local optima. Exhaustive grid search (Algorithm 2) is the only sound baseline. Multi-round LLM tuning is also not applicable: without a meaningful local feedback, the LLM's multi-turn refinement degenerates into random trial-and-error.
>
> *Regarding AMR:* this refines spatial meshes during simulation, not a parameter optimization method. Our parameters include dt, CFL, integration order, and tolerances, none of which AMR addresses.
>
> > "Have you considered evaluating LLMs...where they can choose to invoke a scanning algorithm as a tool, rather than...internal reasoning?"
>
> Most scientific agent benchmarks (Table 1) rely on internal reasoning, as a benchmark we report the existing mature methods (and reveal their flaws). Still, we can add an ablation where the LLM provides an initial guess and refinement scheme, testing whether planning accelerates search. Our toolkit's standardized APIs support this extension.
>
> > "What is the justification for excluding failed tasks from the efficiency geometric mean?...reward models that succeed rarely but efficiently"
>
> First, failed runs are already penalized by SR. Also, by definition, failed tasks produce useless results, hence the 'efficiency' should be defined as zero. In that case, including cases with E=0 would collapse the geometric mean to zero.
>
> We note that no single metric captures both reliability and cost-awareness; However, this also gives better guidance in practice: one can first filter by SR to have reliable models, then rank by efficiency.
>
> Still, we will add a unconditional "relative cost" (C_llm/C_bf for all tasks including failures) in an appendix; we expect rankings to remain consistent, because most failed runs hit the 10-trials cap and have large costs accumulated.
>
> > "How sensitive are the results to the specific prompt templates used?"
>
> Good suggestion. We will add prompt template sensitivity as an ablation.
>
> Thank you for all suggestions. We would be grateful if the reviewer would consider updating their assessment.
>
> [1] N. Hansen, "The CMA Evolution Strategy: A Tutorial," arXiv:1604.00772, 2023. (pycma 1D limitation: https://github.com/CMA-ES/pycma/issues/86)
> [2] J. A. Nelder & R. Mead, "A Simplex Method for Function Minimization," *The Computer Journal*, 1965. (https://en.wikipedia.org/wiki/Nelder-Mead_method)
> [3] https://en.wikipedia.org/wiki/Bisection_method
> [4] https://en.wikipedia.org/wiki/Golden-section_search

---

> > ### Author Rebuttal · Reviewer_AGU6 · 2026-04-02
> >
> > The rebuttal resolves some concerns, but the core issues (narrow task formulation relative to claims, underpowered ablations) and the substantial volume of revision promises (new baselines, prompt sensitivity, tool-invocation ablation) make it difficult to evaluate the paper as-is. I am willing to update my score accordingly.

---

> > > ### Author Response · Authors · 2026-04-04
> > >
> > > Thanks for your reply.
> > >
> > > The new baselines are non-LLMs, they are quite fast to conduct, and even with better results than brute force, they do not change the nature of the conclusion, as pointed out earlier.
> > >
> > > > we note that the main finding of this benchmark is "LLMs using internal reasoning are not as efficient as even brute-force search." Even if a new baseline outperforms our naive baseline, it only strengthens this conclusion. New baselines will not change the core finding.
> > >
> > > We are adding these baselines and will keep you posted.
> > >
> > > For LLM related ablations, we are trying our best, and will keep you posted too. We do not want to rush unverified results in this short window, but a few weeks is more than enough for 1~2 ablations before eg, camera ready. Still, keep in touch.
> > >
> > > Please let us know if there is anything else.
> > >
> > > # below are updated experiments
> > >
> > > Regarding **Unconditional Relative Cost**, pls refer to our reply to Reviewer LesF: https://openreview.net/forum?id=ww57OvgpP9&noteId=8qPjJb552Q
> > >
> > > The overall takes are:
> > > 1. Rankings are consistent across metrics: models with higher conditional efficiency have lower unconditional relative cost. Our core conclusion is unchanged.
> > > 2. In single-round, Qwen3-32B and GPT-OSS-120B achieve unconditional cost below brute-force. They tend to optimize cost more aggressively, and their failed runs are cheap and coarse, keeping total cost low.
> > > 3. In multi-round, most models exceed brute-force cost unconditionally, confirming the sunk-cost concern in our previous reply: failed multi-round runs accumulate large costs from the 10-trial cap. Only GPT-OSS-120B (0.78) stays below 1.
> > >
> > > ## Prompt Sensitivity (Reviewer AGU6 Q3)
> > >
> > > We perturbed the prompt template into 25 variants (paraphrasing, reordering, detail level; using SOTA closed LLMs, followed by manual reviews) and re-evaluated across the same solvers as in ablation studies. Below, "Shift" is the change of the mean value across 25 variants plus original recording minus original recording.
> > >
> > > | Mode | Success Mean Shift | Eff Mean Shift |
> > > |---|---|---|
> > > | Single-round | +0.069 | -0.047 |
> > > | Multi-round | +0.072 | -0.090 |
> > >
> > > The mean shifts are insignificant for both success rate and efficiency. Prompt wording does not substantially change the results.
> > >
> > > ## Non-LLM Baselines (Reviewer AGU6, r3rf)
> > >
> > > We implemented three additional optimization baselines for monotonic multi-round parameters: Bisection (O(log n) binary search on the grid), Nelder-Mead 1D (two-point simplex in log-space), and (1+1)-ES. Evaluated on the same subset as ablation studies, with the same convergence tolerance as the brute-force search.
> > >
> > > Note: "Success Rate" below is the percentage among all runs, not just those successful with brute-force. Rel. Cost is the instance-wise normalized cost relative to brute-force.
> > >
> > > | Method | Success Rate | Rel. Cost (geo. mean) |
> > > |---|---|---|
> > > | Brute-force (existing) | 93.5% | 1.000 |
> > > | Bisection | 97.8% | 1.08 |
> > > | Nelder-Mead 1D | 97.8% | 1.42 |
> > >
> > > Bisection performs very similarly to brute-force in both success rate and cost (1.08x), as expected for a structured 1D monotonic search. Nelder-Mead achieves the same success rate but at 1.42x cost, since it requires adjacent simulation evaluations per objective call (sim(x) and sim(x·refine_factor) for self-convergence checking). Unlike our brute-force search, which gradually refines with a fixed factor (leading to efficient caching), Nelder-Mead's next-round targets are likely to miss the cache, leading to higher cost.
> > >
> > > (1+1)-ES proved computationally impractical. On Heat 1D alone, it achieved only 75% success rate with 459 runs (vs ~96 for brute-force, ~5x more evaluations). The stochastic mutation wastes roughly half of evaluations moving away from convergence in this structured 1D problem. We observed similar behavior for other solvers (e.g., Euler 1D with 648 runs vs. 54 for brute-force, ~12x more), so we omit ES from the final comparison.
> > >
> > > These results justify our choice of brute-force scan, as the other methods are either comparable or too expensive.

---

### Official Review · Reviewer_LesF · 2026-03-12

**Soundness:** 2
**Presentation:** 3
**Significance:** 3
**Originality:** 3
**Overall Recommendation:** 4
**Confidence:** 3

**Summary:**

The authors propose a comprehensive benchmark for evaluating how skilled LLMs are at tuning physical simulation parameters, given a target accuracy. Reported metrics are the success rate of the LLM in providing a value which makes the simulation accurate enough, and the efficiency which measures the total LLM + simulation time compared to a brute-force parameter search.The study shows that LLM systematically underperform, both in single and multi rounds tasks, compared to the brute-force baseline.

**Compliance With Llm Reviewing Policy:**

Affirmed.

**Final Justification:**

Main concerns adressed in the rebuttal.

**Key Questions For Authors:**

- It would be interesting to know which fraction of the cost is linked to LLM calls, and which is due to simulation calls. For multi-steps tasks, is the lower efficiency mainly due to the cost of model or to the cost of the tools ? In other words, if one subtracts the cost of running the LLM to the total cost, is the efficiency greater or equal to 1 ? If not, it implies that LLMs are worse than a brute-force method, and therefore downstream experiments are not very useful given that LLMs are already failing this simple baseline.
- It is claimed that experts typically use heuristics to fine-tune parameters. What are examples of such heuristics ? Can one analyse the heuristics developed by the LLMs and compare to those ?

**Limitations:**

yes

**Strengths And Weaknesses:**

Strengths :
- Presentation : Very well-written paper, clearly understandable
- Extensively-designed benchmark with scientific backup. Comparison is done with standard methods (brute force and bayesian optimisation).
- Significance : Tackling an important task which typically requires expert heuristics in the loop. Benchmark dataset provides a stepping stone for improving LLMs in the direction of gaining intuition for physical problems.
- On top of a comprehensive benchmark, the authors also provide a principled way to use LLMs for physical simulation parameter tuning, made of single-round and multi-round inference.

Weaknesses :
- The answer to the research question feels a bit predictable : since the multi-step task is restricted to monotonic parameters, one would expect any improvement brought by LLMs to be marginal from the brute-force approach. It would be interesting to know what the authors anticipated from this experiment. Could a multi-round experimentation on non monotonic parameters also be ran ? How much more useful would one expect LLM to be in that case ?
- The task scope is limited to tuning a single parameter at a time, which seems to make the use of LLMs disproportionate to the complexity of the problem.
- The proposed benchmark identifies a failure mode of LLMs, i.e. let alone the fact that they don't have an "expert" intuition into how to tune the parameters, they are also not able to plan a brute-force search. This is somehow surprising. There is however limited intuition and no investigation into why LLMs are not performing well at the proposed tasks, and how to improve those.

---

> ### Author Rebuttal · Authors · 2026-03-31
>
> We thank the reviewer for the positive assessment of presentation, benchmark design, and significance, and for recognizing SimulCost as "a stepping stone for improving LLMs in the direction of gaining intuition for physical problems."
>
> > "The answer to the research question feels a bit predictable...one would expect any improvement brought by LLMs to be marginal from the brute-force approach. It would be interesting to know what the authors anticipated..."
>
> We expected LLMs to either (a) follow a mature, gradual refinement path using a fixed, trackable scheme (e.g., progressively doubling resolution), or (b) reason about the parameter space and make large, deliberate jumps when far from convergence. Instead, our failure analysis (Appendix B.6) reveals two surprising failure modes:
>
> > (1) *Blind exploration*: LLMs skip intermediate resolutions to save cost but land on non-converged solutions, i.e., cost-saving shortcuts that backfire.
>
> > (2) *Conservative strategy*: LLMs choose unnecessarily fine resolutions "to be safe," making tiny adjustments (instead of doubling/halving) to save marginal cost, which hinders exploration.
>
> Cost-awareness and correctness conflict in non-obvious ways. We will expand this analysis in the main text.
>
> > "Could a multi-round experimentation on non monotonic parameters also be ran?...How much more useful would one expect LLM to be?"
>
> For non-monotonic parameters, there is no monotonic relationship between the parameter value and solution quality; adjacent parameter values can both produce poor but similar results. This means checking the solver feedback with neighbor parameter's 'convergence/similarity' does not at all indicate whether the LLM is near the optimum. Without a meaningful signal, multi-round iterative refinement degenerates into random trial-and-error.
>
> > "The task scope is limited to tuning a single parameter at a time...makes the use of LLMs disproportionate to the complexity of the problem."
>
> This reflects real practice: domain experts isolate individual parameters for trackable, diagnosable changes. Varying multiple simultaneously obscures cause-and-effect. Reference tractability reinforces this: with n=20 grid points, single-parameter search requires 20 evaluations, while 3-parameter joint search requires 20^3=8,000, making exhaustive references prohibitively expensive (Section 5). This scope and trade-off are acknowledged in Limitations (Section 5); we will re-emphasize in methodology.
>
> > "they are also not able to plan a brute-force search...no investigation into why LLMs are not performing well"
>
> To clarify: the benchmark does not claim LLMs cannot invoke search tools; it evaluates LLMs' adjustment ability with internal reasoning, which is how most existing scientific agent benchmarks currently operate (Table 1). The finding that internal reasoning incurs 1.5-2.5x higher cost than brute-force precisely identifies the flaw of existing practices. As a benchmark, we measure and diagnose this flaw and motivate future improvements; developing those improvements, however, requires this benchmark first. Still, we will add an ablation where the LLM provides both an initial guess and a refinement scheme, testing whether intuition and planning can accelerate search.
>
> > "they are also not able to plan a brute-force search...no investigation into why LLMs are not performing well"
>
> To clarify the metric definition: SimulCost measures only simulation cost (FLOPs), not LLM inference cost. LLM token cost is excluded entirely. So yes, the reported efficiency of 0.4-0.7 means LLMs invoke more simulation FLOPs than brute-force scanning, even without counting LLM inference. In real workflows, simulation/experiment dominates by orders of magnitude: a CFD solve runs hours to weeks; an LLM call takes seconds. Our metric is measurable, platform-independent, and reflects the real bottleneck LLM-automated experimentation will face.
>
> > "It is claimed that experts typically use heuristics to fine-tune parameters. What are examples of such heuristics?"
>
> These are reflected in our brute-force search design (Appendix A.1), e.g., non-target parameters are fixed to such rule-of-thumb values: CFL~0.25 for explicit time-stepping (when CFL is not the target), starting with reasonable coarse meshes. We will explicitly mention these designs in revision. All the config files for reference search will be public too for later references.
>
> Regarding comparing LLM heuristics to expert ones: we did not observe a consistent heuristic strategy emerging from LLMs' internal reasoning. Analyzing LLM-developed heuristics becomes more meaningful in a tool-augmented setting where LLMs can code search strategies; we will note this as future work.
>
> We thank the reviewer for the constructive feedbacks and are committed to these revisions. We would be grateful if the reviewer would consider updating their assessment.

---

> > ### Author Rebuttal · Reviewer_LesF · 2026-04-03
> >
> > Regarding non monotonic relations: How do experts deal with those cases ? Does non monotonic mean random? If not, I agree that the final solution proposed by the model will not be indicative of the « closest » best solution it was close to, the LLM should at least try to recover the underlying structure, even if non-monotonic, or is it that the number of runs is too small for that ?
> >
> > Otherwise, most issues have been  resolved. I’m willing to update my score.

---

> > > ### Author Response · Authors · 2026-04-04
> > >
> > > Thanks for the quick reply!
> > >
> > > Regarding the non-monotonic parameters, to our best knowledge, there are 2 solutions:
> > >
> > > 1. Using mathematics or physics prior knowledge; e.g., the relaxation ratio in Jacobi iteration (in the Heat2D). One can use the iteration matrix's spectrum to analyze the convergence range of this parameter. However, this only applies to simplified cases, where e.g. the calculation of the spectrum is either analytical or numerically cheap (otherwise, if computing the signal to aid diagnosis is more expensive than the simulation, one better just tune the simulator directly).
> > >
> > > 2. Combined with post-analysis, e.g., plotting out the simulated results and observing abnormal behavior such as oscillation near the shock wave front (as in the Burgers and Euler), then realizing the related parameter should be controlled (e.g., more damping in the flux limiter).
> > >
> > > These 2 solutions correspond to:
> > >   1. adding more thinking effort (and potentially more explicit guidance on math/physics reasoning in the prompt); we have tested thinking effort, but we will list the latter as future work. Thanks for helping us identifying this.
> > >   2. enabling fully agentic mode (and even with the help of vision models, e.g., to catch the oscillation); both fully agentic mode and multimodal are currently listed as future works.
> > >
> > > We thank the reviewer for sparking this discussion; it can enhance our explanation and future work sections in the revision.
> > >
> > > # below are updated experiments
> > > ## Unconditional Relative Cost (Reviewer AGU6 Q2, r3rf W1, LesF Q1)
> > >
> > > As promised, we report the unconditional relative cost $C_\text{LLM}/C_\text{BF}$ over all tasks including failures, with no filtering by success.
> > >
> > > **Method.** We compute `model_cost / dummy_cost` for every task (successful or not), aggregated via two-stage task-weighted geometric mean (same protocol as our conditional efficiency metric).
> > >
> > > **Results** (Eff = conditional efficiency over successful tasks only, higher is better; RC = unconditional relative cost over all tasks, lower is better; RC < 1 means LLM uses less total simulation cost than brute-force):
> > >
> > > | Model | Eff SR | RC SR | Eff MR | RC MR |
> > > |---|---|---|---|---|
> > > | GPT-OSS-120B | 0.46 | 0.60 | 0.86 | 0.78 |
> > > | Qwen3-32B | 0.40 | 0.57 | 0.60 | 0.98 |
> > > | Claude-3.7-Sonnet | 0.36 | 0.89 | 0.42 | 1.97 |
> > > | GPT-5 | 0.27 | 1.80 | 0.55 | 1.44 |
> > > | Llama-3-70B-Instruct | 0.18 | 1.36 | 0.44 | 2.83 |
> > >
> > > Full breakdown will be provided in the revised Appendix. In all:
> > >
> > > 1. Rankings are consistent across metrics (Spearman $\rho$ = 0.80 SR, 0.90 MR): models with higher conditional efficiency have lower unconditional relative cost. Our core conclusion is unchanged.
> > > 2. In single-round, Qwen3-32B (RC 0.57) and GPT-OSS-120B (RC 0.60) achieve unconditional cost below brute-force. They tend to optimize cost more aggressively, and their failed runs are cheap and coarse, keeping total cost low.
> > > 3. In multi-round, most models exceed brute-force cost unconditionally, confirming the sunk-cost concern in our previous reply: failed multi-round runs accumulate large costs from the 10-trial cap. Only GPT-OSS-120B (0.78) stays below 1.

---

### Official Review · Reviewer_LaML · 2026-03-14

**Soundness:** 3
**Presentation:** 2
**Significance:** 1
**Originality:** 1
**Overall Recommendation:** 3
**Confidence:** 3

**Summary:**

This paper presents SimulCost, a benchmark for evaluating AI agents’ success rate and cost-efficiency in tuning the parameters of physics simulations. Cost efficiency here includes the high cost it takes to run a simulation if you make certain parameters too fine-grained, which must be balanced with accuracy grain. The benchmark uses 12 simulators and a variety of tasks. The authors find that frontier LMs are only getting 40-50% success rates vs a simple scanning approach, indicating room for improvement.

**Compliance With Llm Reviewing Policy:**

Affirmed.

**Final Justification:**

Proposed changes would improve the paper if they show that this task is challenging for state of the art agents too (like Claude Code with its full harness and tools).

**Key Questions For Authors:**

I'd love to hear whether simple tuning methods like prompt optimization can greatly improve LLMs' performance at the benchmark already.

**Limitations:**

Yes

**Strengths And Weaknesses:**

The main strengths of the paper are that it targets a useful practical problem and compares performance of multiple leading LMs to show that they have issues on it. The benchmark contains diverse tasks that seem to have been carefully selected.

The weakness I'm most worried about is that this is a very niche area. How much impact would improving LMs at this task really have? Why not target something broader? There are a lot of benchmark papers being submitted to AI conferences and usually the most impactful ones target a broad interesting use case.

I was also surprised that the authors did not try model tuning methods beyond few-shot prompting. Just running a prompt optimizer or perhaps SFT or RL could lead to better performance. This seems like the kind of task where LMs might be uncalibrated with the parameter ranges they emit out of the box but might easily be tuned. It would be good to know how much headroom there is to improve performance after these "obvious" approaches. Another interesting intervention might be a web search tool -- perhaps people have documented good simulator settings online.

---

> ### Author Rebuttal · Authors · 2026-03-31
>
> We thank the reviewer for recognizing that SimulCost targets a useful practical problem with diverse, carefully selected tasks.
>
> > "this is a very niche area. How much impact would improving LMs at this task really have?"
>
> We respectfully disagree. Any realistic LLM-based scientific workflow will encounter expensive external tools, e.g., simulations, wet-lab experiments, and manufacturing. Those costs span hours to months, far exceeding LLM inference. Yet every scientific agent benchmark since 2024 (ScienceAgentBench, DiscoveryBench, AstaBench, MLGym, Agent-Lab, MLE-Bench, ...) has omitted tool cost or conflated it with wall-clock time (Table 1). We are the first to systematically track this. Our finding that LLMs incur 1.5-2.5x higher computational cost than brute-force scanning reveals a cost-awareness gap relevant to any domain with expensive tool calls. Without measuring cost, we cannot improve it.
>
> > "the authors did not try model tuning methods beyond few-shot prompting"
>
> This conflates a benchmark paper with a methods paper. Our contribution is evaluation infrastructure, not LLM optimization. This is analogous to asking the GLUE paper [1] to have included fine-tuned models at publication. Post-training is listed as future work in our draft and requires a benchmark like ours to exist first.
>
> The reviewer has overlooked significant methodology:
>
> > "Just running a prompt optimizer"
>
> We already evaluate two forms: (1) multi-round inference (Section 3.3) iteratively refines choices using solver feedback each round, a form of prompt optimization via iterative feedback; (2) ICL ablation (Section 4.5) evaluates three in-context learning variants on how examples guide parameter selection, another form of prompt optimization via few-shot examples.
>
> > "Another interesting intervention might be a web search tool"
>
> Web search is essentially RAG with irreproducible noise. Our ICL ablation (Section 4.5) is the controlled version: curated examples with known content, isolating LLM reasoning from retrieval noise.
>
> These concerns seem to stem from misunderstanding of scope or oversight of existing methodology. We have reported this concern to the AC/SAC. We urge the reviewer to revisit Sections 3.3 and 4.5, and welcome updated assessment and updated questions with a closer examination of our draft.
>
> [1] A. Wang et al., "GLUE: A Multi-Task Benchmark and Analysis Platform for Natural Language Understanding," *ICLR*, 2019. (https://arxiv.org/abs/1804.07461)

---

> > ### Author Rebuttal · Reviewer_LaML · 2026-04-04
> >
> > Thanks for the response. However, on my original pass through the paper, I did look at the ICL approaches considered and I still believe that the paper only evaluates methods far behind the state of the art. The paper itself says in section 4.5 that ICL means "can we leverage historical examples to improve performance?" but there are many ways to create better prompts than simply putting in a few examples, which can easily be run on a training set (e.g. DSPy, GEPA, ACE, etc). Why not run one of those and see how it compares with the manual ICL construction approaches in the paper? Likewise, the state of the art agents that people use every day do have web search, so why not evaluate that with the caveat that it is less reproducible? Maybe the documentation for some of these simulators would let a modern agent, like Claude Code, find better parameters much faster than the ICL and iterated execution methods in this paper.
> >
> > The reason these things are relevant for a benchmark paper is to establish that the benchmark truly is hard for state of the art AI. I believe that it probably is hard for Llama or Qwen getting 10 trials at the problem with few-shot ICL examples, but that isn't the frontier of AI today. Scientists creating these expensive simulations would surely use a stronger agent than that, give it web search or agentic access to the simulator software and its docs, etc.

---

> > > ### Author Response · Authors · 2026-04-04
> > >
> > > Thanks for the response, specially for checking our Sec. 3.3. and 4.5. Here, some new clarifications.
> > >
> > > > "there are many ways to create better prompts than simply putting in a few examples, which can easily be run on a training set (e.g. DSPy, GEPA, ACE, etc)."
> > >
> > > 2 respective clarifications:
> > >
> > >   1. Prompt sensitivity, like the reviewer AGU6 has pointed out and we agreed to add as an ablation study. Thanks to you both for this suggestion.
> > >
> > >   2. Why not use modular AI software frameworks like (e.g. DSPy, G, etc.)? We carefully checked these frameworks and found that their purpose is "building modular software/improving agent frameworks." The former is not unrelated, as we rely on existing software. The latter has been acknowledged as a future work direction.
> > >
> > > We once again kindly note that our purpose is to set the standard and most basic baselines, while the future methodologies that can be tried are limitless. We opensource the benchmark, which opens the gate for, e.g., building a leaderboard, so that any potential improvements can be tried in the future.
> > >
> > > > "that people use every day do have web search"; "so why not evaluate that with the caveat that it is less reproducible?"; and why not calibrate prompts by "run on a training set"
> > >
> > > We will not argue about our different philosophy of "reproducibility." We kindly but firmly believe in reproducibility, and that having inconsistent results will hurt the purpose of testing future improvements, especially in scientific settings.
> > >
> > > Here is a new angle we should not ignore about the nature of scientific tasks.
> > >
> > >   1. Unlike common software engineering where many codes share commonality (e.g., many instant messaging applications), scientific tasks by definition are on the frontier of unknowns, hence, often:
> > >
> > >   2.  New (simulation/experiment) task is rarely existing on the internet; e.g., the tutorial for single airfoil simulation is common on the OpenFOAM forum, but not so for multi-airfoil setups. If a benchmark's result relies heavily on web search queries, then it under-measures the LLM's reasoning and adjustment ability on the fly, specially when most tasks are frontier and novel in scientific tasks.
> > >
> > >   3. Similarly, we assume the worst way to utilize this benchmark is by SFT/post-training on the whole set, as it only forces LLMs to memorize the optimal distribution of parameters here, instead of enhancing the adjustment ability for new tasks.
> > >
> > > We thank the reviewer for sparking this discussion and will add the above to the future implications in the revision.
> > >
> > > > "Maybe the documentation for some of these simulators would let a modern agent, like Claude Code, find better parameters much faster than the ICL and iterated execution methods in this paper." ; "I believe that it probably is hard for Llama or Qwen getting 10 trials at the problem with few-shot ICL examples, but that isn't the frontier of AI today."
> > >
> > > All of our ablation studies are indeed using Claude, not Llama or Qwen. This is explicit in our methodology Sec. 3.5.
> > >
> > > We are frustrating the reviewer mistaken this simple fact again.
> > >
> > > As for coding agents or any potential dedicated improved agent framework, those are future works to be tested only when a good definition (the core value of a benchmark, like this work) has been established.
> > >
> > > We opensource the benchmark and will help in building a leaderboard.
> > >
> > >
> > > # promised ablation: LLM w/ search
> > > ## LLM Initial Guess + Tool Search (Reviewer r3rf Q4, LaML)
> > >
> > > We did an ablation where the LLM provides an initial parameter guess for the brute-force search, where latter refines from there. Due to the search algorithm's nature, results are only for multi-round (MR) mode:
> > >
> > > | Metric | Delta (Tool Search MR - Internal Adjustment MR) |
> > > |---|---|
> > > | Success Rate | +0.086 |
> > > | Efficiency | -0.520 |
> > > | Rel. Cost | +2.363 |
> > >
> > > Tool search increases the success rate by 8.6%, confirming LLM internal adjustment is a bottleneck. The relative cost, however, increases to nearly 2-fold. The reason: the LLM's initial guess enters the search algorithm as the starting point. As in the unconditional cost table (see our reply to R LesF), Claude (similarly GPT and Llama) tends to select more conservative parameters (more likely to succeed but more expensive). Those choices can largely overshoot or undershoot the near-optimal; the search likely stops at the 1st or 2nd iteration (for checking convergence), but the initial cost is already high due to unnecessarily high resolutions.
> > >
> > > Two concrete examples selected from the database (euler_1d, CFL, MR mode):
> > >
> > > | profile | precision | LLM initial guess | Near Optimal | Undershoot | MR Cost ratio |
> > > |---|---|---|---|---|---|
> > > | p3 | high | 0.10 | 0.50 | 80% | 13.8x |
> > > | p1 | low | 0.25 | 0.50 | 50% | 5.7x |
> > >
> > > This indicates that a good initial guess from the LLM should land in an intermediate safe range, avoiding both overshoot and undershoot, while still guaranteeing convergence by the search algorithm. We will include this as a new future implication.

---

### Decision · Program_Chairs · 2026-04-30

**Decision:**

Accept (regular)

**Comment:**

This paper initially received mixed recommendations from the reviewers. In general, reviewers appreciate the novelty and significance of the research problem, the paper's effort to benchmark its method across diverse domains, and the design of the evaluation metrics. Several reviewers initially raised concerns about the paper's baseline choices and requested a more comprehensive evaluation. In particular, the negative reviewer expressed major concern about the baselines, suggesting that the current baselines are not the state-of-the-art. The rebuttal provided substantial experimental evidence in response and convinced most reviewers to recommend weak acceptance, provided that the submission will properly incorporate the rebuttal's additional experiments. After reviewing all reviewers' feedback, my opinion aligns with the majority, and I recommend that ICML accept this work.